# APOBEC3B regulates R-loops and promotes transcription-associated mutagenesis in cancer

Jennifer L. McCann [1,2,3,4,16], Agnese Cristini [5,16], Emily K. Law [1,2,3,4], Seo Yun Lee[6,7], Michael Tellier [5,8], Michael A. Carpenter [1,2,3,4,9,10], Chiara Beghè[5], Jae Jin Kim[6,7], Anthony Sanchez [6], Matthew C. Jarvis[2,3,4], Bojana Stefanovska [1,2,3,4,9,10], Nuri A. Temiz [2,11], Erik N. Bergstrom[12,13,14], Daniel J. Salamango[2,3,4], Margaret R. Brown[2,3,4], Shona Murphy [5], Ludmil B. Alexandrov [12,13,14], Kyle M. Miller [6,15,17] ✉, Natalia Gromak [5,17] ✉ & Reuben S. Harris [1,2,3,4,9,10,17] ✉

The single-stranded DNA cytosine-to-uracil deaminase APOBEC3B is an antiviral protein implicated in cancer. However, its substrates in cells are not fully delineated. Here APOBEC3B proteomics reveal interactions with a surprising number of R-loop factors. Biochemical experiments show APOBEC3B binding to R-loops in cells and in vitro. Genetic experiments demonstrate R-loop increases in cells lacking APOBEC3B and decreases in cells overexpressing APOBEC3B. Genome-wide analyses show major changes in the overall landscape of physiological and stimulus-induced R-loops with thousands of differentially altered regions, as well as binding of APOBEC3B to many of these sites. APOBEC3 mutagenesis impacts genes overexpressed in tumors and splice factor mutant tumors preferentially, and APOBEC3-attributed kataegis are enriched in RTCW motifs consistent with APOBEC3B deamination. Taken together with the fact that APOBEC3B binds single-stranded DNA and RNA and preferentially deaminates DNA, these results support a mechanism in which APOBEC3B regulates R-loops and contributes to R-loop mutagenesis in cancer.

The APOBEC3 family of single-stranded (ss)DNA cytosine deaminases function in the overall innate immune response to viral infection[1,2]. Popularized initially by HIV-1 restriction activity, the seven human APOBEC3 enzymes collectively exhibit activity against a broad number of DNA-based viruses including retroviruses, hepadnaviruses, papillomaviruses, parvoviruses, polyomaviruses and herpesviruses. An important biochemical feature of this family of enzymes is an intrinsic preference for different nucleobases immediately 5′ of target cytosines. For example, APOBEC3B (A3B) and APOBEC3A (A3A) deaminate cytosines in 5′-TC motifs, and the antibody gene diversification enzyme activation-induced cytidine deaminase (AID) prefers 5′AC/GC motifs[3–5].

In addition to beneficial functions in innate and adaptive immunity, multiple DNA cytosine deaminases have detrimental roles in cancer mutagenesis[1,6,7]. Misprocessing of AID-catalyzed deamination events in antibody gene variable and switch regions can result in DNA breaks and chromosomal translocations in B-cell malignancies[7]. Off-target deamination of other genes also occurs at lower frequencies, and the resulting mutations can also contribute to B-cell cancers[7]. In comparison, cancer genomics projects have reported an APOBEC mutation signature in a variety of tumor types (ref. 8 and reviews above). In cancer, the APOBEC3 mutation signature is defined as C-to-T transitions and C-to-G transversions in 5′-TCA and 5′-TCT motifs (single base

substitution (SBS)2 and SBS13, respectively). APOBEC3 enzymes are estimated to be the second largest mutation-generating process in cancer following spontaneous deamination by water, which associates with aging (SBS1) (ref. [8]).

Despite extensive documentation of the APOBEC3 mutation signature in cancer, the precise molecular mechanisms governing this mutational process are unclear. One challenge is the likelihood that at least two enzymes, A3B and A3A, combine in different ways to generate the overall signature (for example, recent studies[9–11] and references therein). However, insights have been gleaned from the physical characteristics of genomes with, for instance, APOBEC3 signature association with chromosomal DNA replication[12–16]. Other genomic structures with exposed ssDNA may be similarly prone to APOBEC3 mutagenesis such as ssDNA loop regions of hairpins[17,18] and ssDNA tracts in recombination and repair reactions, which can manifest as clusters of strand-coordinated mutations (aka. kataegis; for example, refs. [17,19–22]). Together, these studies have indicated a mechanism in which expression of A3B and/or A3A leads to mutagenic encounters with exposed ssDNA followed in some instances by processive local deamination.

Another potential substrate for APOBEC3 enzymes is an R-loop, which occurs when nascent RNA re-anneals to the transcribed DNA strand, creating a three-stranded structure containing an RNA/DNA hybrid and a displaced nontranscribed ssDNA strand[23–25]. R-loops are substrates in AID-catalyzed antibody diversification[7] and represent a prominent source of genome instability in cancer[23–25]. However, evidence linking APOBEC3 enzymes to R-loop-associated mutation and genome instability is lacking apart from a report postulating that U/G mismatches, which can be created by C-to-U deamination of R-loop ssDNA followed by R-loop dissolution and DNA reannealing, may be responsible for a synthetic lethal interaction between A3B activity and uracil excision repair disruption[26].

A3B is strongly implicated in cancer mutagenesis based on constitutive nuclear localization, overexpression in tumors, upregulation by cancer-causing viruses such as human papillomavirus and associations with clinical outcomes[1,27,28]. A3B is also capable of directly inflicting APOBEC signature mutations in human genomic DNA[9–11]. To further investigate A3B in cancer, an unbiased affinity purification and mass spectrometry (AP–MS) approach was used to identify A3B-interacting proteins. Two dozen proteins were recovered in biologically independent experiments, and 60% of the resulting high-confidence interactors had been reported previously as R-loop-associated factors in RNA/DNA hybrid AP–MS experiments[29]. A comprehensive series of genetic, cell biology, biochemistry, genomic and bioinformatic studies showed that A3B functions in R-loop homeostasis, and moreover, R-loop regions impacted by A3B are enriched for APOBEC3 signature mutations including kataegis. Altogether, these results reveal an unanticipated role for A3B in R-loop biology and a distinct mechanism of transcription-associated mutation in cancer.

## Results

### A3B interacts with R-loops and R-loop-associated proteins

To identify A3B regulatory factors, a functional A3B-2xStrep-3xFlag construct (hereafter A3B-SF) was expressed in 293T cells, anti-Strep affinity-purified, and subjected to MS to identify interacting proteins (workflow in Extended Data Fig. 1a). This procedure included RNase A and high salt concentrations to enrich for direct and strong interactions, respectively. Immunoblots, Coomassie gels and DNA deaminase activity assays validated the presence, enrichment and activity of affinity-purified A3B (Extended Data Fig. 1b–d). An enhanced green fluorescent protein (eGFP)-SF construct and an empty 2xStrep-3xFlag vector were negative controls.

Six independent AP–MS experiments yielded 24 specific A3B-interacting proteins (Supplementary Table 1 and Extended Data Fig. 1e,f). These proteins were abundant in all six A3B-SF datasets

and absent in GFP-SF or empty vector datasets. A total of 60% of these A3B interactors had been found independently in S9.6 AP–MS experiments[29] (Fig. 1a,b). As the S9.6 mAb binds RNA/DNA hybrid with high affinity (Methods), this interaction overlap suggested that A3B may also interact with R-loops. To test this hypothesis, interactions between A3B and multiple R-loop-associated factors were confirmed by co-immunoprecipitation (co-IP; Fig. 1c,d and Extended Data Fig. 1e,f). For example, doxycycline (Dox)-inducible A3B-eGFP was immunoprecipitated from MCF10A cells with an anti-eGFP antibody and the R-loop-associated protein hnRNPUL1 was detected by immunoblotting (Fig. 1c,d). Parallel slot blots showed that R-loops also copurified with A3B-eGFP in an RNase H-sensitive manner demonstrating specificity (Fig. 1d).

The S9.6 mAb was then used to IP RNA/DNA hybrids from MCF10A cells treated with phorbol 12-myristate 13-acetate (PMA) to induce endogenous A3B expression[30]. Immunoblotting confirmed the enrichment of an established R-loop interacting protein, TOP1 (ref. [31]), and a shared R-loop and A3B interactor, hnRNPUL1, in all S9.6 IP reactions except those saturated with a synthetic RNA/DNA hybrid competitor (Fig. 1e). Lamin B1 served as a negative control. Endogenous A3B copurified with R-loops in basal noninduced conditions, and this interaction increased following PMA treatment (Fig. 1e). Notably, no A3B signal was detected in S9.6 pull-downs from A3B knockout (KO) MCF10A cells (Fig. 1e and Extended Data Fig. 2a–d).

### A3B depletion triggers increased nuclear R-loop levels

To investigate a potential role for A3B in R-loop biology, R-loop levels were quantified in wild-type (WT) MCF10A and its A3B KO derivative. First, nucleoplasmic S9.6 staining intensity was measured by immunofluorescence (IF) confocal microscopy. These experiments revealed a strong increase in nucleoplasmic S9.6 fluorescence in A3B KO compared to WT cells (Fig. 2a,b). Second, S9.6 dot blots confirmed elevated R-loop levels in the A3B KO cells in comparison to WT (Fig. 2c,d). In both experiments, RNase H treatment eliminated the increase in nucleoplasmic R-loop signals observed in the absence of endogenous A3B. In comparison, nucleolar S9.6 signal was mostly insensitive to RNase H treatment, likely due to rRNA being detected by S9.6 Ab[32].

To further investigate A3B and R-loops, analogous experiments were done using U2OS cells. A3B knockdown caused a strong increase in nucleoplasmic S9.6 staining by IF compared to control cells (Fig. 2e,f and Extended Data Fig. 2e–g). An increase in RNA/DNA hybrid signal was also obtained in S9.6 dot blots from A3B-depleted versus control cells (Fig. 2g,h). As mentioned above, specificity in these experiments was confirmed by RNase H treatment. R-loop imbalances are known sources of DNA damage[23–25], and elevated R-loop levels in A3B KO MCF10A and A3B-depleted U2OS cells triggered concomitant increases in DNA damage as evidenced by staining of the DNA damage marker γ-H2AX (Fig. 2i–l). However, these elevated levels of R-loops and DNA damage did not alter overall rates of DNA replication or cell cycle progression (Extended Data Fig. 2h–k).

### A3B deamination is required to reduce nuclear R-loop levels

Given that A3B loss increased R-loop accumulation, we next asked whether A3B overexpression has the opposite effect. These experiments used JQ1, a bromodomain and extra-terminal protein family inhibitor, to enhance global R-loop levels as shown previously[33,34]. As expected, JQ1-treated but not untreated or DMSO-treated cells exhibited increased R-loops, as measured by S9.6 IF staining (Fig. 3a,b). Similar results were obtained with cells expressing a bacterial mCherry-RNaseH1 D10R-E48R mutant, which binds but does not process R-loops (Fig. 3c,d; Methods). Next, U2OS cells were transfected with A3B-eGFP or eGFP plasmids, incubated for 24 h to allow for protein expression, treated for 4 h with JQ1 and then analyzed by IF for S9.6 staining. We observed that A3B-eGFP caused a substantial decrease in nucleoplasmic S9.6 levels compared to eGFP control

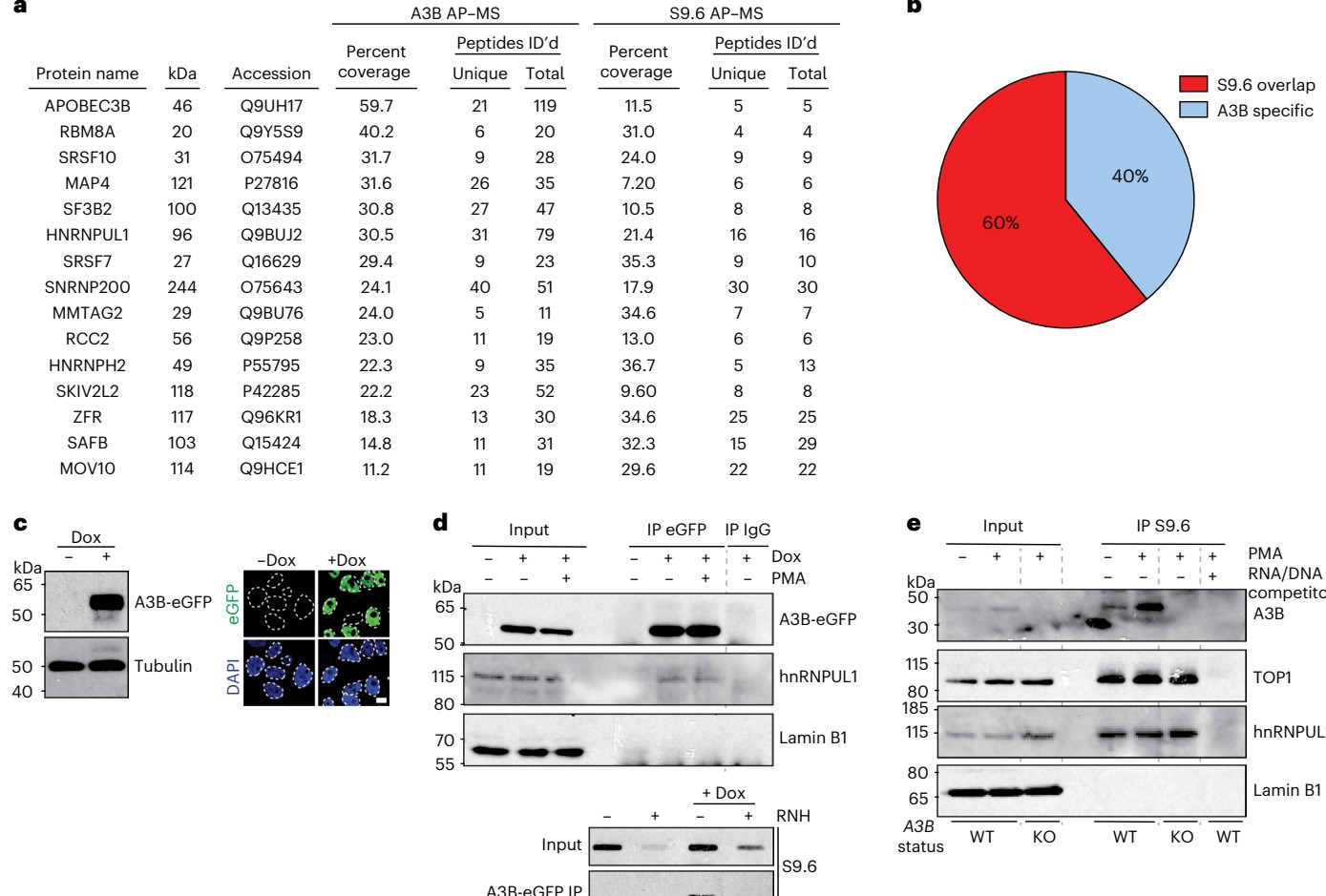

**Fig. 1 | APOBEC3B (A3B) interacts with R-loop-associated proteins.**
**a,b,** Shared proteins in A3B and S9.6 AP–MS datasets. **c,** Immunoblot and IF microscopy analysis of MCF10A-TREx-A3B-eGFP cells treated with vehicle or Dox (1 μg ml⁻¹, 24 h). A3B-eGFP (green) is predominantly nuclear (DAPI, blue). Ten-micrometer scale bar; n = 2 (left); n = 1 (right) biologically independent experiments. **d,** Immunoblots of indicated proteins in A3B-eGFP or IgG IP from TREx-A3B-eGFP MCF10A cells ± Dox (1 μg ml⁻¹, 24 h), treated with PMA

(25 ng ml⁻¹, 2 h) and probed with indicated antibodies (top). Slot blot of A3B-eGFP IP from TREx-A3B-eGFP MCF10A cells ± Dox (1 μg ml⁻¹, 24 h) ± exogenous RNase H (RNH) probed with S9.6 antibody (bottom). n = 2 biologically independent experiments. **e,** Immunoblots of indicated proteins in S9.6 IP reactions from MCF10A WT or *A3B* KO cells treated with PMA (25 ng ml⁻¹, 5 h). n = 2 biologically independent experiments.

(Fig. 3e,f). Interestingly, expression of A3A, which is more active than A3B biochemically[35], had no effect on the S9.6 signal, suggesting a specific R-loop role for A3B (Fig. 3e,f).

The hallmark biochemical activity of A3B is ssDNA C-to-U deamination. To determine whether this activity is required for R-loop regulation, U2OS cells were transfected with constructs expressing A3B-eGFP or the catalytic mutant E255A. Notably, WT A3B caused a substantial reduction in nucleoplasmic R-loop levels as quantified by IF, whereas the catalytic mutant had a less pronounced effect despite similar expression levels (Fig. 3g,h). However, as U2OS cells already express high levels of endogenous A3B, and APOBEC3 enzymes including A3B are reported to oligomerize[36,37], this intermediate phenotype could potentially arise from the oligomerization between the overexpressed mutant A3B and the endogenous A3B.

Therefore, a series of genetic complementation experiments was performed to compare the activities of WT A3B and the E255A catalytic mutant in cells lacking endogenous A3B. First, endogenous *A3B* was ablated from U2OS cells as described above, which resulted in lower A3B protein and activity levels (Fig. 3i). Second, *A3B*-depleted U2OS cells were stably transfected with shRNA-resistant constructs expressing HA-tagged WT A3B, A3B-E255A or an empty vector control (Fig. 3i).

The WT A3B enzyme, but not A3B-E255A, restored ssDNA deaminase activity as expected (Fig. 3i, bottom). Third, R-loop levels were analyzed by S9.6 dot-blot assays. Notably, complementation with WT A3B rescued the effect of *A3B* depletion and caused a significant reduction in R-loops (Fig. 3j,k). In contrast, cells complemented with similar levels of A3B-E255A showed no significant change in R-loop levels (Fig. 3j,k). Finally, these results were confirmed with quantification of the nucleoplasmic S9.6 and mCherry-RNaseH1 mutant IF signals of U2OS parental and *A3B* KO cells complemented with WT or E255A A3B (Fig. 3l–o and Extended Data Fig. 2l–n). Taken together, these results showed that A3B-dependent suppression of R-loops requires catalytic activity.

### A3B alters the genome-wide distribution of R-loops

Increased R-loops in the nucleoplasmic compartment of *A3B*-depleted cells suggested a role for A3B in regulating R-loop levels genome-wide. In support, this elevated signal required transcription as evidenced by treatment of *A3B*-depleted cells with the global transcription inhibitor triptolide (TRP)[38] and the transcription elongation inhibitor, flavopiridol (FLV)[39] (Fig. 4). Next, we investigated the role of A3B in regulating R-loop levels genome-wide using DNA/RNA immunoprecipitation sequencing (DRIP)–seq experiments. DRIP–seq peaks in WT and *A3B*

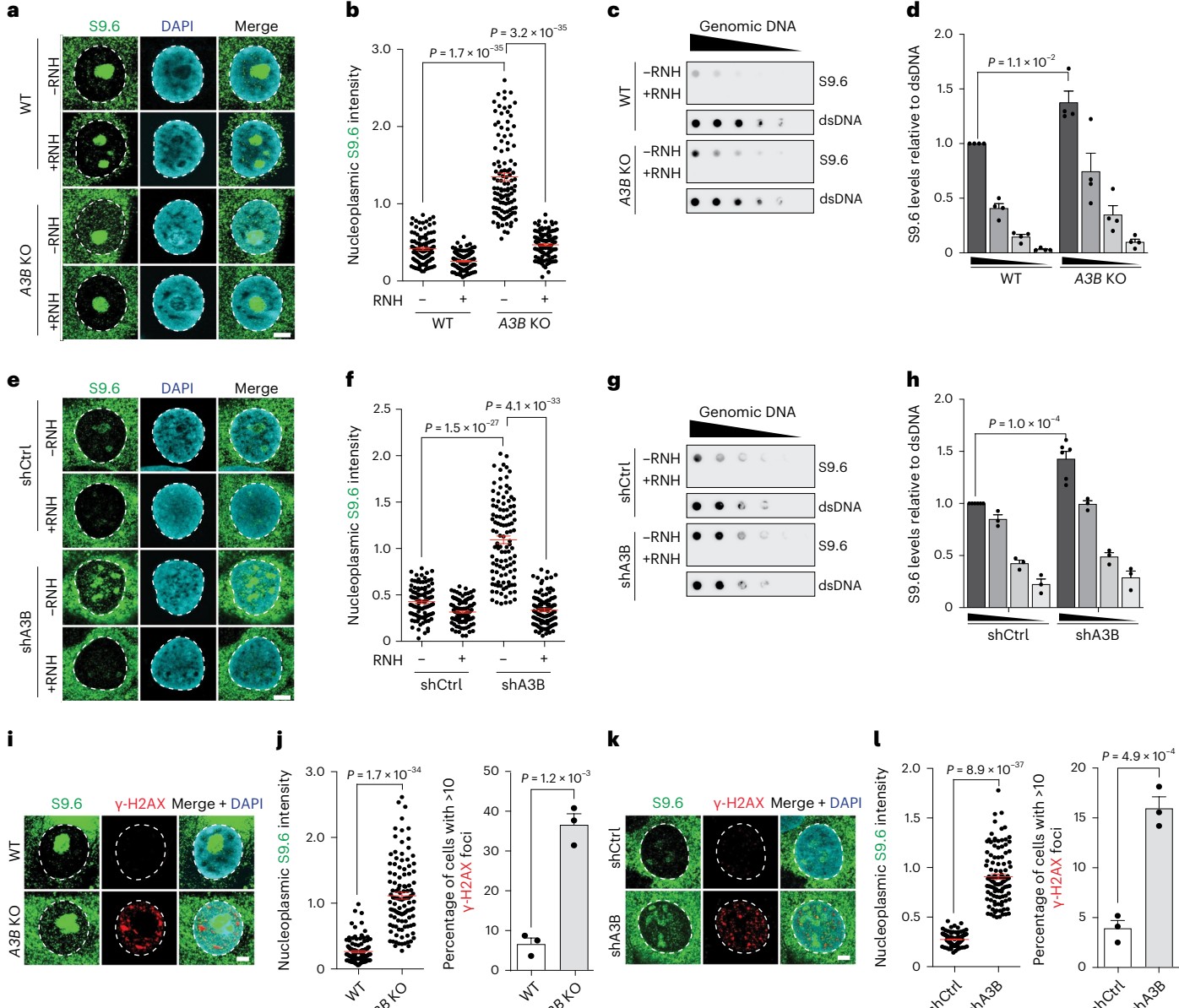

**Fig. 2 | Elevated nuclear R-loop levels in *A3B* knockout and *A3B* depleted cells. a,b,** IF images (**a**) and quantification (**b**) of MCF10A WT and *A3B* KO cells stained with S9.6 (green) and DAPI (blue) (representative images; 5 μm scale; *n* = 3 independent experiments with >100 nuclei per condition; red bars represent mean ± s.e.m.; *P* value by Mann–Whitney test). **c,d,** S9.6 dot-blot analysis of MCF10A WT and *A3B* KO genomic DNA dilution series ± exogenous RNase H (RNH; representative images); parallel dsDNA dot blots provided a loading control (**c**). Quantification normalized to the most concentrated WT signal (representative experiment shown from four independent experiments; mean ± s.e.m.; *P* value by two-tailed unpaired *t*-test) (**d**). **e,f,** IF images (**e**) and quantification (**f**) of U2OS shCtrl and shA3B cells stained with S9.6 (green) and DAPI (blue; representative images; 5 μm scale; *n* = 3 independent experiments with >100 nuclei per condition; red bars represent mean ± s.e.m.; *P* value by Mann–Whitney test). **g,h,** S9.6 dot-blot analysis of a U2OS shCtrl and shA3B

genomic DNA dilution series ± exogenous RNase H (RNH; representative images); parallel dsDNA dot blots provided a loading control (**g**). Quantification normalized to the most concentrated shCtrl signal (representative experiment shown from three independent experiments; mean ± s.e.m.; *P* value by two-tailed unpaired *t*-test) (**h**). **i,j,** IF images (**i**) and quantification (**j**) of MCF10A WT and *A3B* KO cells stained with S9.6 (green), DAPI (blue) and γ-H2AX (representative images; 5 μm scale; *n* = 3 independent experiments with >100 nuclei per condition; red bars represent mean ± s.e.m.; *P* value by Mann–Whitney test (left); *P* value by two-tailed unpaired *t*-test (right). **k,l,** IF images (**k**) and quantification (**l**) of U2OS shCtrl and shA3B cells stained with S9.6 (green), DAPI (blue) and γ-H2AX (representative images; 5 μm scale; *n* = 3 independent experiments with >100 nuclei per condition; red bars represent mean ± s.e.m.; *P* value by Mann–Whitney test (left); *P* value by two-tailed unpaired *t*-test (right).

KO MCF10A cells were mainly intragenic and distributed between protein-coding, long noncoding RNA and enhancer RNA genes (Fig. 5a,b), similar to R-loop distributions observed previously[39–41]. As anticipated by transcription dependence and DRIP–seq peak distributions, the vast majority of DRIP–seq positive regions occurred in expressed genes (Extended Data Fig. 3a).

A global comparison of DRIP–seq peaks between *A3B* KO and WT MCF10A revealed changes in the overall R-loop landscape with 8,296 peaks 'increased', 13,761 peaks 'decreased' and 154,036 peaks 'unchanged' (red versus blue traces in Fig. 5c–e). Representative individual gene results are shown for *GADD45A* and *PHLDA1*, *HIST1H1B* and *SYT8* and *HIST1H1E* and *DDX1* that show increased, decreased and

unchanged R-loop levels, respectively, in KO compared to WT cells (Fig. 5f,h,j). These DRIP–seq results were confirmed by gene-specific DRIP–qPCR (Fig. 5g,i,k). As discussed above, DRIP–qPCR signals were reduced to background levels by RNase H treatment, confirming R-loop specificity (Fig. 5g,i,k, striped bars). Differential DRIP signals in these genes were not due to transcription differences between KO and WT cells (Extended Data Fig. 3b). As expected, negligible DRIP signals were found in nonexpressed genes and intergenic loci (for example, *TFF1* in Extended Data Fig. 3c,d). Similar DRIP results were obtained in *A3B*-depleted HeLa cells (Extended Data Fig. 3e–g).

### A3B accelerates the kinetics of R-loop resolution

Transcriptional activation by different signal transduction pathways is known to increase R-loop formation[42–44]. We therefore investigated whether A3B may also affect signal transduction-induced R-loops. *A3B* WT and KO MCF10A lines were treated with PMA to induce the protein kinase C and noncanonical nuclear factor kappa B (NF-κB) signal transduction pathways that activate the transcription of many genes including endogenous *A3B* (refs. 30,45). DRIP–seq analysis in these cells demonstrated that PMA caused perturbations in the overall R-loop landscape, resulting in increased and decreased R-loop peaks (Supplementary Note and Extended Data Fig. 4a–j). Interestingly, A3B, as detected by chromatin immunoprecipitation followed by sequencing (ChIP–seq) with A3B-eGFP, appeared to bind preferentially to genomic DNA regions overlapping with PMA-enriched R-loop peaks (Supplementary Note and Extended Data Fig. 4a–d,k). Furthermore, kinetic analysis by DRIP–seq and IF revealed that A3B contributes to timely resolution of PMA-induced R-loops (Supplementary Note and Fig. 6).

### Biochemical activities of A3B required for R-loop resolution

To investigate the biochemical activities of A3B in R-loop resolution, WT A3B was purified from 293T cells (Extended Data Fig. 5a) and used for nucleic acid-binding and DNA deamination experiments (Fig. 7 and Extended Data Fig. 5b,c). EMSAs indicated that A3B binds R-loop structures, ssDNA and ssRNA (as expected[35,36,46]), and, to lesser extents, dsDNA, dsRNA and RNA/DNA hybrid (also expected[35,36,46]; Extended Data Fig. 5b,c). These native EMSAs were hard to quantify due to accumulation of large protein/nucleic acid complexes in the wells. We therefore quantified the release of fluorescently labeled ssRNA and ssDNA from A3B by incubating with unlabeled nucleic acid competitors. These experiments demonstrated that A3B binds equally strongly to both ssRNA and ssDNA (Fig. 7b).

RNA is an inferred inhibitor of A3B based on experiments where exogenous RNase A treatment is required to detect ssDNA deaminase activity in cancer cell extracts[30,47]. We therefore wondered whether the RNA in R-loop structures might inhibit the deaminase activity of A3B on the unpaired ssDNA. Qualitative single timepoint reactions indicated clear activity on free ssDNA cytosines and potentially reduced activities on cytosines in bubble, short and long R-loop structures

(Fig. 7c). A quantitative time course comparing A3B activity on free ssDNA versus ssDNA in the R-loop structure indicated that the latter substrate is only ~2-fold less preferred (Fig. 7d). These data showed that R-loops can be substrates for A3B-catalyzed ssDNA deamination. The twofold diminution in activity may be due to ssDNA inaccessibility caused by the relatively short nature of the synthetic R-loop (21 nucleotides) and/or competition with unpaired ssDNA or ssRNA.

To gain additional insights into A3B function in R-loop biology, we analyzed the nucleoplasmic R-loop phenotypes of A3B mutants defective in either nuclear localization (Mut1) (ref. 48) or nucleic acid binding (Mut2) (ref. 46). Both of these activities are governed by the N-terminal domain of A3B and independent of the C-terminal domain, which binds ssDNA weakly but catalyzes deamination[35,46,48]. We confirmed the nuclear localization defect of Mut1 and showed that Mut2 still retains this activity (Fig. 7e). Mut2 was also purified and, in contrast to WT A3B, demonstrated defective binding to ssRNA and ssDNA (Fig. 7f and Extended Data Fig. 5a). However, Mut2 still retained high levels of ssDNA deaminase activity and was similarly active on free and short R-loop-containing ssDNA substrates (Fig. 7g). This result is consistent with the possibility that unpaired nucleic acid may interfere with the deaminase activity of the WT enzyme but not Mut2, which has reduced nucleic acid-binding activity. Key biochemical results with WT and Mut2 A3B were reproduced with independent >85% pure protein preparations (Extended Data Fig. 5d–g). Most importantly, in contrast to WT A3B, neither mutant was capable of decreasing nucleoplasmic R-loop levels following JQ1 treatment in U2OS cells (Fig. 7h,i). The separation-of-function Mut2 protein also had a diminished capacity to co-IP interactors (Extended Data Fig. 1f). These results combined to indicate that both nuclear localization and nucleic acid-binding activities are required for A3B to regulate nucleoplasmic R-loop levels.

### Evidence for R-loop mutagenesis by A3B

Our results suggested a model in which exposed ssDNA cytosines in R-loop regions are deaminated by A3B and resolved into mutagenic or nonmutagenic outcomes (Fig. 8a). Mutagenic outcomes are predicted to reflect the intrinsic structural preference of A3B for TC motifs[5] and more broadly TCW and RTCW[10,49–51]. For comparison, A3A exhibits a preference for YTCW motifs[10,11,49–51].

First, we predicted that higher rates of transcription should lead to higher rates of R-loop formation and increased exposure to A3B-mediated deamination because prior work had already correlated gene expression and R-loop formation[39]. This idea was addressed using whole-exome sequenced (WES) breast cancers and corresponding RNA-seq data from The Cancer Genome Atlas (TCGA) project as well as whole-genome sequenced (WGS) breast cancers from the International Cancer Genome Consortium (ICGC) and normal breast tissue gene expression data from the Genotype-Tissue Expression (GTEx) project (Methods). An initial association between gene expression levels and APOBEC3-attributed mutations was intriguing but became

**Fig. 3 | A3B overexpression reduces nuclear R-loop levels. a–d**, IF images (**a**) and quantification (**b**) of U2OS cells stained with S9.6 antibody (green) and treated with 0.5 μM JQ1 or 0.005% DMSO for 4 h. **c,d**, IF images (**c**) and quantification (**d**) of U2OS cells expressing catalytic inactive mCherry-RNaseH1 (mCherry-RNaseH1-mut, red) and treated with 0.5 μM JQ1 or 0.005% DMSO for 4 h (representative images; 5 μm scale; *n* = 3 independent experiments with 60 nuclei per condition; red bars represent mean ± s.e.m.; *P* value by Dunnett multiple comparison; NS, not significant). **e–h**, IF images (**e,g**) and quantification (**f,h**) of U2OS cells expressing the denoted eGFP construct (green) and stained with S9.6 (red) and DAPI (blue). Top, experimental workflow and bottom, representative images (5 μm scale; *n* = 3 independent experiments with >60 nuclei per condition; red bars represent mean ± s.e.m.; *P* value by Mann–Whitney test). **i**, Immunoblots of U2OS shCtrl or shA3B cells complemented with empty vector (EV), A3B-HA or A3B-E255A-HA. Bottom, the results of a DNA deaminase activity assay with extracts from the indicated cell lines (reaction quantification

below with purified A3A as a positive control (+) and reaction buffer as a negative control (-); *n* = 2 independent experiments). **j,k**, Dot-blot analysis of U2OS shCtrl or shA3B cells complemented with EV, A3B-HA or A3B-E255A-HA. A genomic DNA dilution series ± exogenous RNase H (RNH) was probed with either S9.6 antibody or dsDNA antibody as a loading control (representative images) (**j**). Quantification normalized to the most concentrated shCtrl signal (*n* = 3 independent experiments; mean ± s.e.m.; *P* value by two-tailed unpaired *t*-test) (**k**). **l–o**, IF images (**l**) and quantification (**m**) of U2OS WT and *A3B* KO cells expressing GFP-EV, A3B WT or A3B E255A (green). Cells were stained with S9.6 antibody (blue). IF images (**n**) and quantification (**o**) of U2OS WT and *A3B* KO cells expressing GFP-EV, A3B WT or A3B E255A (green). Cells were cotransfected with catalytic inactive mCherry-RNaseH1 mutant (mCherry-RNaseH1-mut, red; representative images; 5 μm scale; *n* = 3 independent experiments with 60 nuclei per condition; red bars represent mean ± s.e.m.; *P* value by Dunnett multiple comparison).

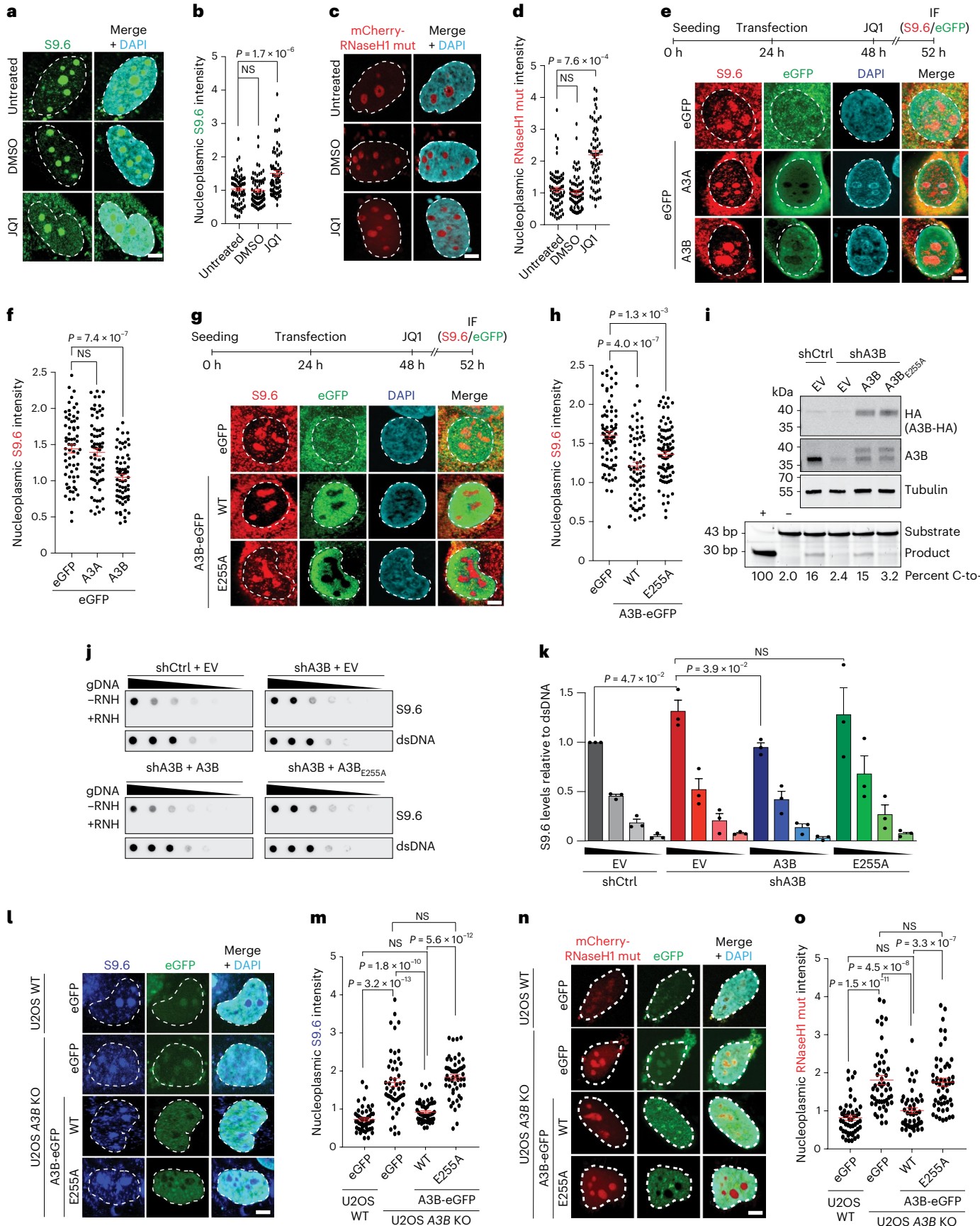

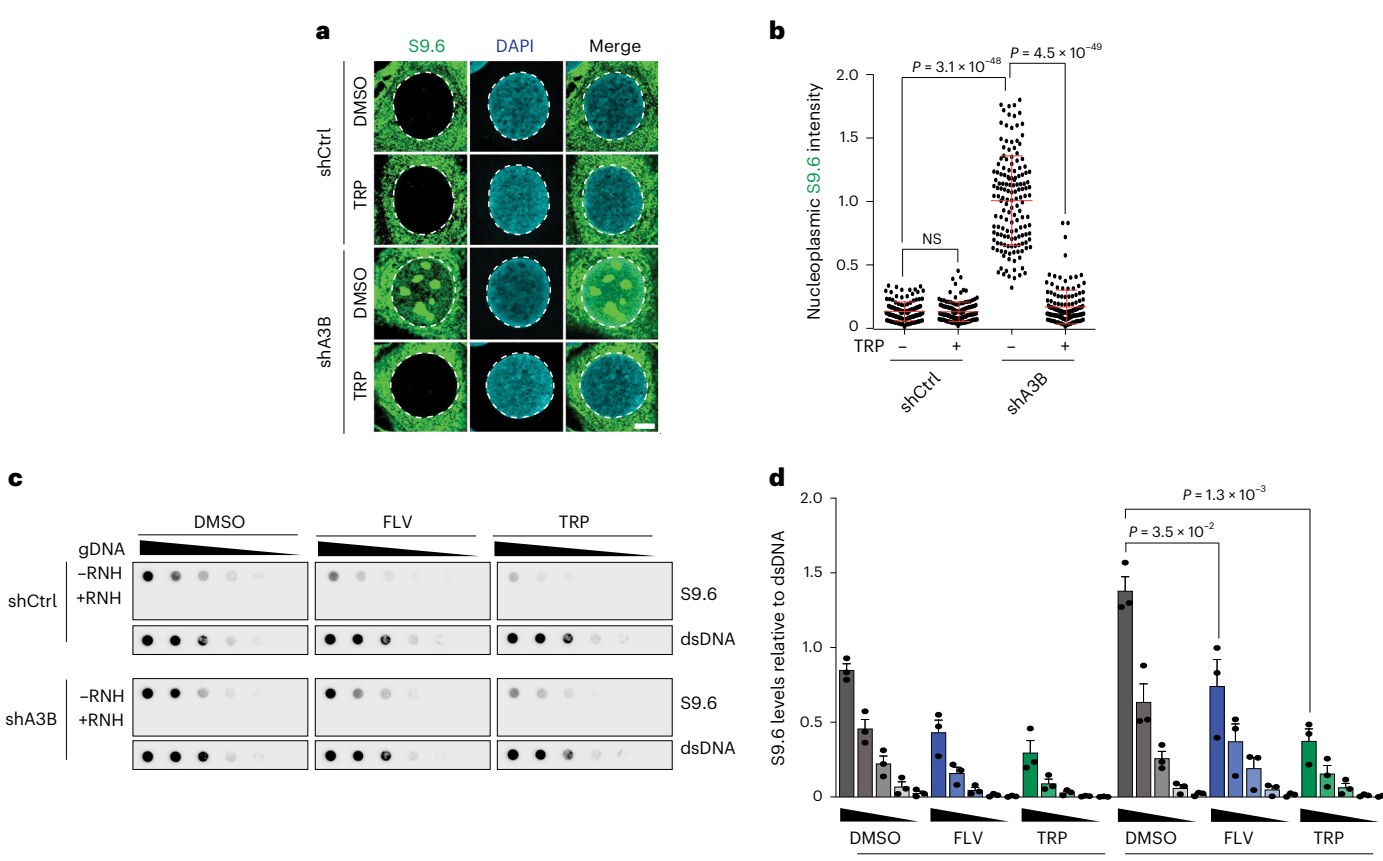

**Fig. 4 | A3B-regulated R-loops are transcription-dependent. a,b**, IF images (**a**) and quantification (**b**) of U2OS shCtrl and shA3B cells treated with TRP (1 µM, 4 h) and subsequently stained with S9.6 (green) and DAPI (blue; representative images; 5 µm scale; $n = 3$ independent experiments with >100 nuclei per condition; red bars represent mean ± s.e.m.; $P$ value by Mann–Whitney test). **c,d**, S9.6 dot-blot analysis of a genomic DNA dilution series ± RNH from U2OS shCtrl or shA3B cells treated with TRP (1 µM, 4 h) or FLV (1 µM, 1 h; representative images); parallel dsDNA dot blots provided a loading control (**c**). Quantification was normalized to the most concentrated shCtrl/DMSO signal ($n = 3$ independent experiments; mean ± s.e.m.; $P$ value by two-tailed unpaired $t$-test) (**d**). TRP, triptolide; FLV, flavopiridol.

insignificant after accounting for gene size (Extended Data Fig. 6a,b). However, a strong positive association emerged between the magnitude of gene overexpression in breast cancer compared to normal breast tissue and the proportion of mutations attributable to APOBEC3 deamination (TCW mutations in Fig. 8b; RTCW/YTCW breakdown in Extended Data Fig. 6c). Thus, the higher the degree of gene overexpression in breast cancer, the higher the proportion of mutations attributable to APOBEC3, with the highest overexpressed gene group showing an average of over 50-fold more APOBEC3 signature mutations than any of the three lowest expressed gene groups (Methods). The highest overexpressed gene group in breast cancer (>16-fold above normal breast tissue) also showed a strong bias of APOBEC3 signature mutation on the nontranscribed strand over the transcribed strand ($P < 0.038$ by Wilcoxon rank-sum test; Supplementary Table 1).

Second, we predicted that splice factor mutant tumors will show elevated levels of APOBEC3 signature mutations, as splicing defects are known to increase R-loop formation[52–54]. This idea was investigated by splitting the TCGA breast cancer WES dataset into tumors with and without mutations in splice factor genes and evaluating associations with the proportion of mutations attributable to APOBEC3 activity. Remarkably, 53% of the breast tumors with mutant splice factor genes (43/81) had substantial levels of APOBEC3 signature mutations (Fig. 8c). In contrast, only 35% of breast tumors without mutations in the same splice factor gene set (326/841) showed a detectable APOBEC3 mutation signature (Fig. 8c; $P < 0.017$ by Fisher's exact test). Interestingly, the A3B-associated RTCW motif was only absent from one of

the splice factor mutant tumors (1/43) in comparison to the nonsplice factor mutant group (52/326) (Extended Data Fig. 6d,e; $P = 0.028$ by Fisher's exact test). Splice factor mutant tumors also had a higher mean percentage of APOBEC3-attributed mutations (39% versus 31%, respectively; $P = 0.042$ by unpaired two-sample Welsh's $t$-test) as well as a higher total number of mutations on average than nonsplice factor mutated samples ($P = 0.0018$ by Welch's two-sample $t$-test). Even the top quartile of tumors with the strongest APOBEC3 signature had a higher total number of mutations in the splice factor mutant group ($P = 0.0095$ by Welch's two-sample $t$-test). Similarly sized housekeeping gene sets selected randomly were not highly mutated (Methods). Thus, the observed splice factor defects are likely contributing to the higher rates of mutation. In strong support of A3B-dependent activity on R-loops resulting from aberrant splicing, A3B overexpression suppressed the increase in R-loops caused by treating U2OS with the splicing inhibitor pladienolide B (Plad B; Fig. 8d).

Third, because APOBEC3 signature kataegic events are due to at least one APOBEC3 enzyme[17,19–22], we asked what proportion of these events occur in genes and, moreover, occur on the nontranscribed strand versus the transcribed strand. Global mapping of all kataegis events in primary breast adenocarcinomas from the Pan-Cancer Analysis of Whole Genomes (PCAWG) revealed a bimodal distribution with one peak located within 1 kbp of a structural variation (SV) breakpoint and another similarly sized peak much further away from the nearest SV breakpoint (~1 Mbp; $n = 198$ WGS datasets; blue bars in Fig. 8e). As expected[55], the SV breakpoint-proximal subset of kataegis events is

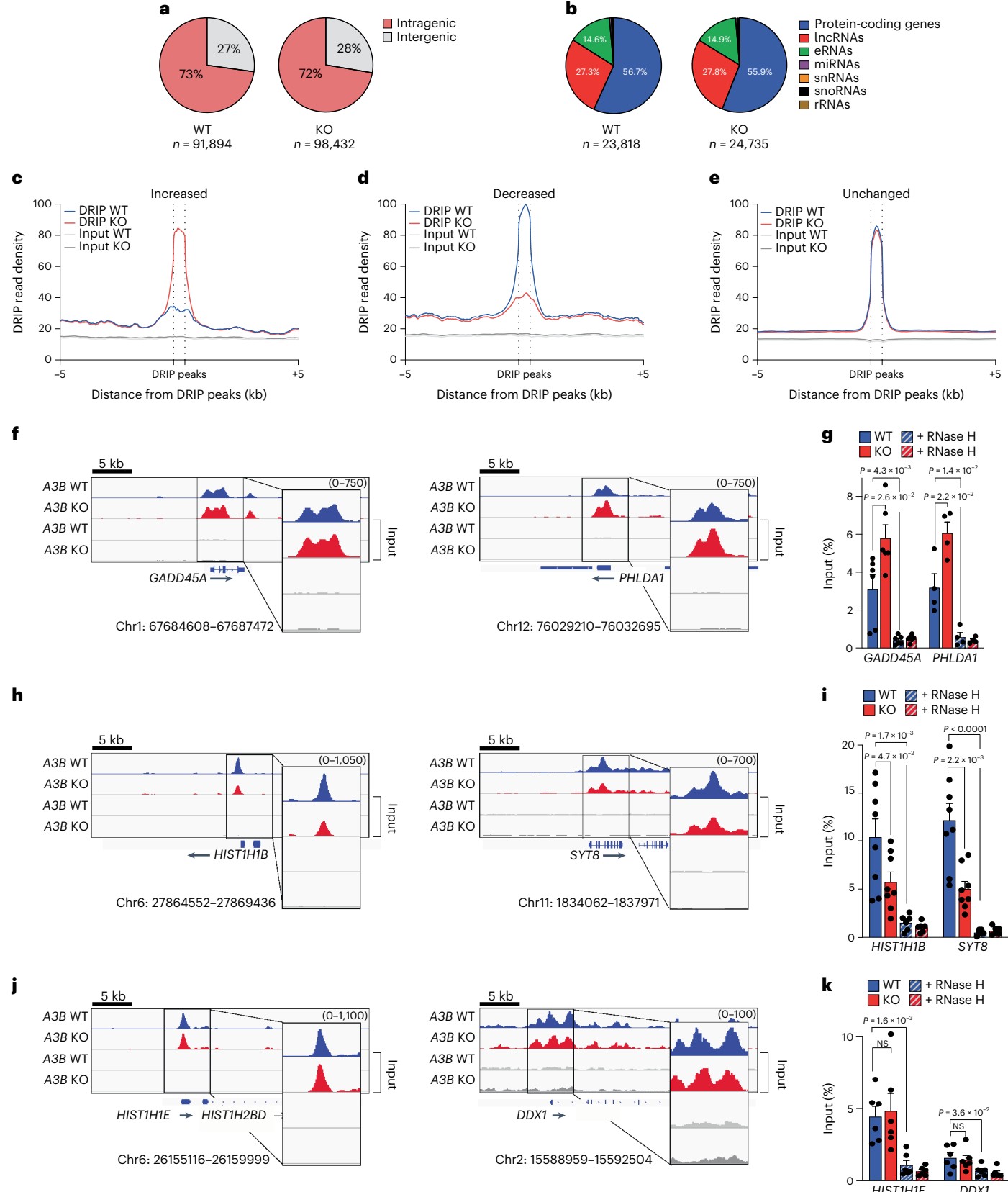

**Fig. 5 | A3B affects a large proportion of R-loops genome-wide.**
**a,b**, Pie graphs representing R-loop distributions in MCF10A WT and *A3B* KO cells. **c–e**, Meta-analysis of read density (FPKM) for DRIP–seq results from WT (blue) and *A3B* KO (red) MCF10A partitioned into three groups (**c**, increased; **d**, decreased and **e**, unchanged) as described in the text. Input read densities are indicated by overlapping gray lines. **f–k**, DRIP–seq profiles (**f,h,j**) for representative genes in each of the groups defined in **c–e** (WT, blue; KO, red). DRIP–qPCR for the indicated genes (**g,i,k**) ±exogenous RNase H (RNH; striped bars). Values are expressed as percentage of input (means ± s.e.m.). *n* = 6 (*GADD45A*, *HIST1H1E* and *DDX1*), *n* = 4 (*PHLDA1*), *n* = 8 (−RNH) and *n* = 6 (+RNH; *HISTH1B* and *SYT8*) biologically independent experiments for indicated gene. *P* value by two-tailed unpaired *t*-test.

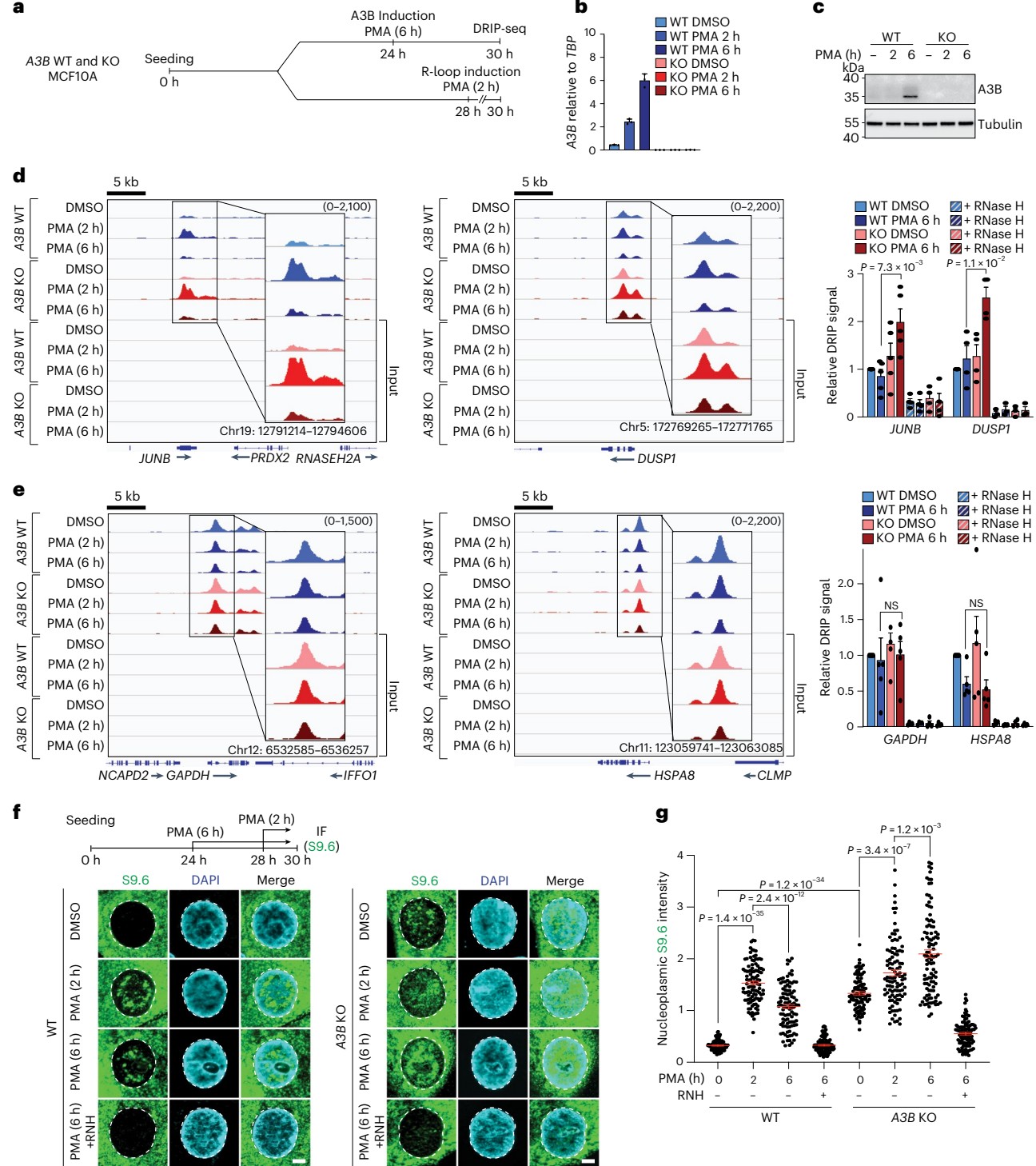

**Fig. 6 | Kinetics of R-loop induction and resolution. a**, Schematic of the DRIP–seq workflow used for **d**–**e**. **b**, RT–qPCR of *A3B* mRNA from MCF10A WT and *A3B* KO cells treated with PMA (25 ng ml⁻¹) for the indicated times. Values are expressed relative to the housekeeping gene, *TBP* (*n* = 3; mean ± s.e.m.; KO levels not detectable). **c**, Immunoblots of extracts from MCF10A WT and *A3B* KO cells treated with PMA (25 ng ml⁻¹) for the indicated times and probed with indicated antibodies (*n* = 2 independent experiments). **d**, DRIP–seq profiles for two PMA-responsive genes, *JUNB* and *DUSP1*, in DMSO or PMA-treated (25 ng ml⁻¹) MCF10A WT (top profiles, blue) and *A3B* KO (bottom profiles, red). DRIP–qPCR ± exogenous RNase H (RNH; striped bars) is shown in the histogram to the right. Values are normalized to DMSO WT (mean ± s.e.m.). *n* = 5 (−RNH) and *n* = 4 (+RNH; *JUNB*) and *n* = 4 (−RNH) and *n* = 3 (+RNH; *DUSP1*) biologically

independent experiments for indicated gene. *P* value by two-tailed unpaired *t*-test. **e**, DRIP–seq profiles for two PMA nonresponsive genes, *GAPDH* and *HSPA8*, in DMSO or PMA-treated (25 ng ml⁻¹) MCF10A WT (top profiles) and *A3B* KO (bottom profiles). DRIP–qPCR ± exogenous RNase H (RNH; striped bars) is shown in the histogram to the right. Values are normalized to DMSO WT (mean ± s.e.m.). *n* = 5 (−RNH) and *n* = 4 (+RNH; *GAPDH* and *HSPA8*) biologically independent experiments for indicated gene. *P* value by two-tailed unpaired *t*-test. **f,g**, IF images (**f**) and quantification (**g**) of MCF10A WT and *A3B* KO cells treated with PMA (25 ng ml⁻¹) for the indicated times and stained with S9.6 (green) and DAPI (blue; 5 μm scale; *n* = 2 independent experiments with >100 nuclei per condition; red bars represent mean ± s.e.m.; *P* value by Mann–Whitney test).

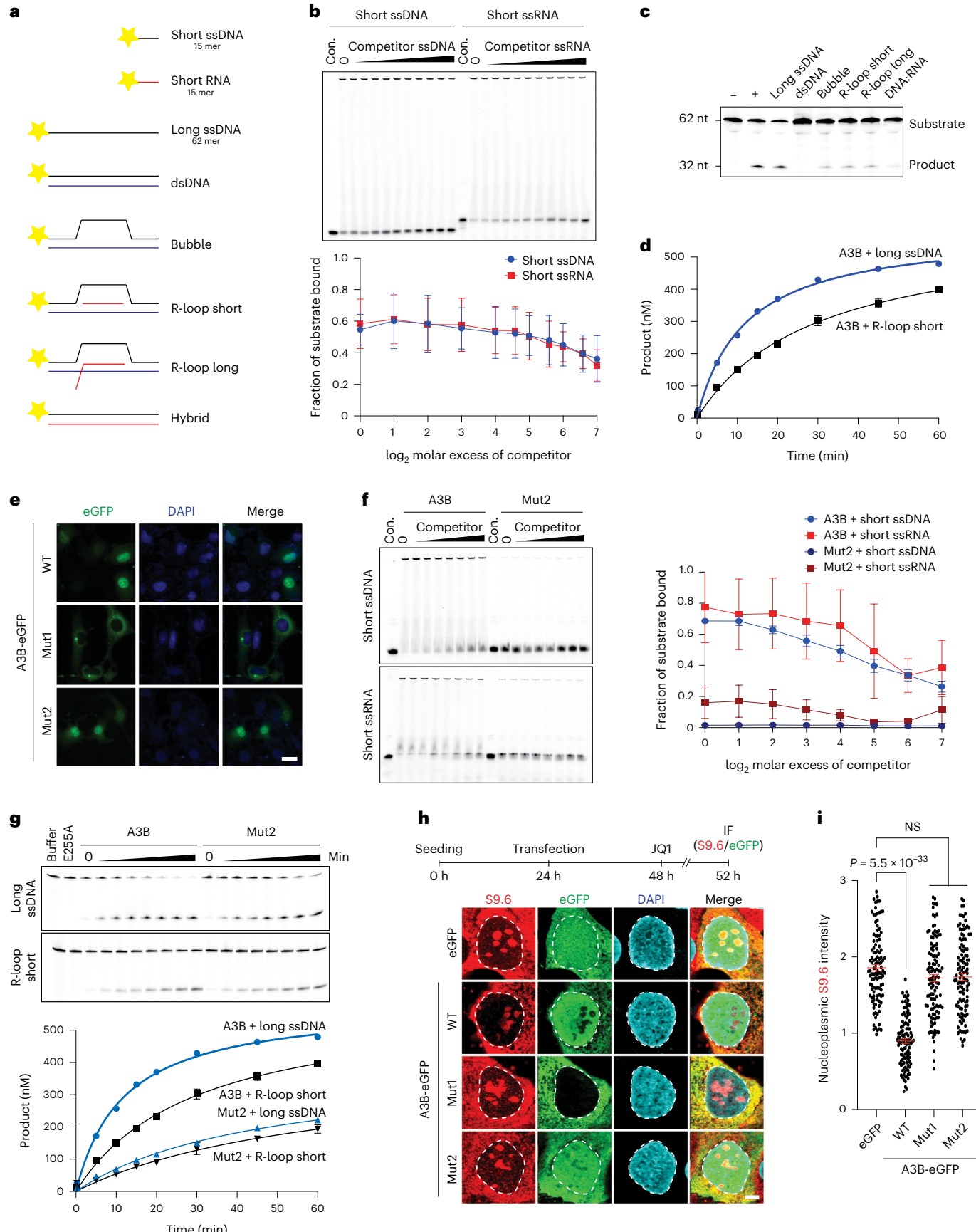

**Fig. 7 | A3B biochemical activities required for R-loop resolution.**
**a**, Schematics of the nucleic acids used in biochemical experiments
(5′ fluorescent label indicated by yellow star). The 15-mer short ssDNA and short
RNA were used in EMSAs in **b** and **f**, and the 62-mer long ssDNA was used alone or as
annealed to the indicated complementary nucleic acids (black, DNA; red, RNA) in
other experiments. **b**, Native EMSAs of A3B binding to fluorescently labeled short
15 mer ssDNA or RNA in the presence of increasing concentrations of otherwise
identical unlabeled competitor. The corresponding quantification shows the
average fraction bound to substrate ± s.d. from $n = 3$ independent experiments.
**c**, Substrates in **a** tested qualitatively for deamination by A3B ($n = 2$ independent
experiments). Negative (−) and positive (+) controls are the long ssDNA alone and
deaminated by recombinant A3A. **d**, A quantitative time course of A3B-catalyzed
deamination of the long ssDNA versus the R-loop (short) substrate (mean ± s.d. of
$n = 3$ independent experiments are shown with most error bars smaller than the

symbols). **e**, Subcellular localization of A3B-eGFP (WT), Mut1 and Mut2 in U2OS
cells (scale = 10 µM; $n = 2$ independent experiments). **f**, EMSAs comparing A3B WT
and Mut2 binding to short 15 mer ssDNA and RNA in the presence of increasing
concentrations of otherwise identical unlabeled competitor ssDNA or RNA. The
corresponding quantification shows the average fraction bound to substrate ± s.d.
from $n = 3$ independent experiments. **g**, Quantitative comparison of A3B WT
and Mut2 deamination of the long ssDNA versus an R-loop (short) substrate.
Representative gels are shown for the time-dependent accumulation of product,
along with quantitation of $n = 3$ independent experiments (mean ± s.d. with most
error bars smaller than the symbols; for comparison, the WT data are the same as
those in **d**). **h,i**, IF images (**h**) and quantification (**i**) of U2OS cells expressing the
indicated eGFP construct (green) and stained with S9.6 (red) and DAPI (blue; 5 µm
scale; $n = 2$ independent experiments with >100 nuclei per condition; red bars
represent mean ± s.e.m.; $P$ value by Mann–Whitney test).

---

likely due to deamination of resected ssDNA ends during recombina-
tion repair. Also expected, dispersed APOBEC3-attributed mutations
occur on average of >1 Mbp apart (yellow bars in Fig. 8e). In contrast, the
majority of APOBEC3-attributed kataegic events (>75%) map >10 kbp
away from SV breakpoints (teal bars >10 kbp in Fig. 8e), and ~17% of
these events occur within R-loop regions identified above by DRIP–seq
(red bars in Fig. 8e; Methods).

Finally, we investigated the sequence motifs of mutations across
individual kataegic events compared to nonclustered mutations within
R-loop regions partitioned into nontranscribed strand and transcribed
strand regions of genes and within intergenic regions. Specifically,
we investigated the overall enrichments for A3B-associated RTCA
and A3A-associated YTCA tetranucleotide motifs for each mutation
found in a sample (R = A or G; Y = C or T)[49]. This analysis indicated
that APOBEC3 kataegic mutations overlapping NTS R-loop regions
are skewed toward A3B-associated RTCA motifs, in contrast to dis-
persed APOBEC3 mutations (Fig. 8f,g and Extended Data Fig. 7). The
overall RTCW skew of kataegic (>3 mutations per cluster) ver-
sus dispersed APOBEC3 mutations is elevated for mutations occur-
ring on the nontranscribed strand and transcribed strand but not for
mutations in intergenic regions (Fig. 8f). For greater stringency, this
latter analysis was repeated for longer APOBEC3 kataegic tracts
(≥5 mutations per cluster) and a statistically significant enrichment is
only evident for RTCA events on the nontranscribed strand of genes
(Fig. 8g). Specifically, this significant enrichment was driven by
longer, R-loop-associated kataegic events occurring within the non-
transcribed strand, which was not observed for transcribed strand
or intergenic events (Extended Data Fig. 7). Furthermore, 70% of the
R-loop kataegis occurring within the nontranscribed strand were
enriched for A3B-associated RTCA motifs compared to a minority of
events associated with A3A-like YTCA motifs (Fig. 8g and Extended
Data Fig. 7b). Representative nontranscribed strand kataegic events
are shown for *PRKCA* and *LGR5* (Fig. 8h). Taken together, these
bioinformatic analyses support a model in which at least a subset of

R-loop structures is susceptible to C-to-U deamination events that
occur on the nontranscribed strand and are most likely catalyzed
by A3B.

## Discussion

Our studies reveal an unanticipated role for the antiviral enzyme A3B in
R-loop biology. We delineate a functional relationship between A3B and
R-loops with higher R-loop levels occurring upon A3B deficiency and
lower R-loop levels upon A3B overexpression. Genome-wide DRIP–seq
experiments in physiological conditions and upon activation of a signal
transduction pathway with PMA indicated that thousands of R-loops in
cells are affected by A3B. This number represents over 10% of R-loops
genome-wide, which is comparable to the impact of established R-loop
regulatory factors[40,56]. These findings are also in line with the knowl-
edge that multiple proteins contribute to R-loop regulation, includ-
ing RNase H1, RNase H2, TOP1, SETX, AQR, UAP56/DDX39B, FANCD2
and BRCA1/BRCA2 (refs. 23–25). Determining the specific subsets of
factors responsible for regulating individual R-loops remains a
challenge for future studies.

Our studies also shed light on the molecular mechanism of R-loop
resolution. A3B depletion and overexpression have opposing effects
with the former causing a net increase in R-loops and the latter a net
decrease. A3B complementation experiments revealed that this A3B
function requires an intact catalytic glutamate (E255) consistent with
a role for cytosine-to-uracil deamination. Nuclear localization is also
required, which further supports a direct model and helps rule out
indirect cytoplasmic effects. Our biochemical experiments showed
that ssRNA- and ssDNA-binding activities are comparable in strength.
Together with the fact that A3B's strong nucleic acid-binding activity
resides within the N-terminal domain and the weaker ssDNA-binding
activity required for catalysis is governed by the C-terminal domain,
we favor a working model in which direct binding of A3B to nascent
ssRNA adjacent to R-loops and/or to ssDNA exposed in R-loop struc-
tures is critical for R-loop regulation. Based on specialized mechanisms

---

**Fig. 8 | R-loop mutagenesis and kataegis by APOBEC3B. a**, Model for A3B-
mediated R-loop resolution with and without mutation. Other R-loop regulatory
factors are depicted in shades of green and blue. Transcription, splicing and
other RNA- and R-loop-associated complexes are not depicted for clarity. **b**, A dot
plot showing the fraction of APOBEC3-attributed mutations (per Mbp per tumor)
in the indicated gene expression groups (fold change (FC) in breast tumors
relative to the average observed in normal breast tissues). This analysis includes
only breast cancers with significant APOBEC3 signature enrichment ($Q < 0.05$;
$n = 154$ tumors). Pairwise comparisons are significant for all combinations of
the lowest three versus the highest four expression groups ($P$ value by Welsh's
$t$-test). **c**, Stacked bar graphs showing the proportion of each COSMIC mutation
signature in TCGA breast tumors with mutations in splice factor genes or not
($n = 81$ splice factor mutated tumors; $n = 841$ for nonsplice factor mutated
tumors; $P < 0.017$ by Fisher's exact test). The APOBEC3 signature percentage (red)
comprises COSMIC signatures SBS2 and SBS13, and other signatures are shown in

different shades of gray. **d**, Quantification of nucleoplasmic R-loop levels
in U2OS cells expressing an empty vector (EV) control or A3B and treated
with DMSO or the splicing inhibitor Plad B (4 µM, 2 h; $n = 3$ independent
experiments with >50 nuclei per condition; red bars represent mean ± s.e.m.;
$P$ value by two-tailed unpaired $t$-test). **e**, Distribution of the distances to the
nearest SV of all nonclustered APOBEC3 mutations (gold), all kataegic mutation
events (teal) and R-loop-associated APOBEC3 kataegic mutations (red). **f,g**, Box
plot representations of the fold-enrichment within R-loop regions of short (≥3)
and long (≥5) APOBEC3 kataegic tracts (RTCA/YTCA) in PCAWG breast tumor
WGS. Data are shown for NTS, TS and intergenic regions, and nonclustered
mutations within the same regions serve as controls ($Q$ values by Mann–Whitney
$U$ test). **h**, Representative NTS kataegic events in *PRKCA* (chromosome 17
64,627,540–64,628,540) and *LGR5* (chromosome 12 71,850,425–71,852,135).
WT trinucleotides and mutational outcomes are indicated.

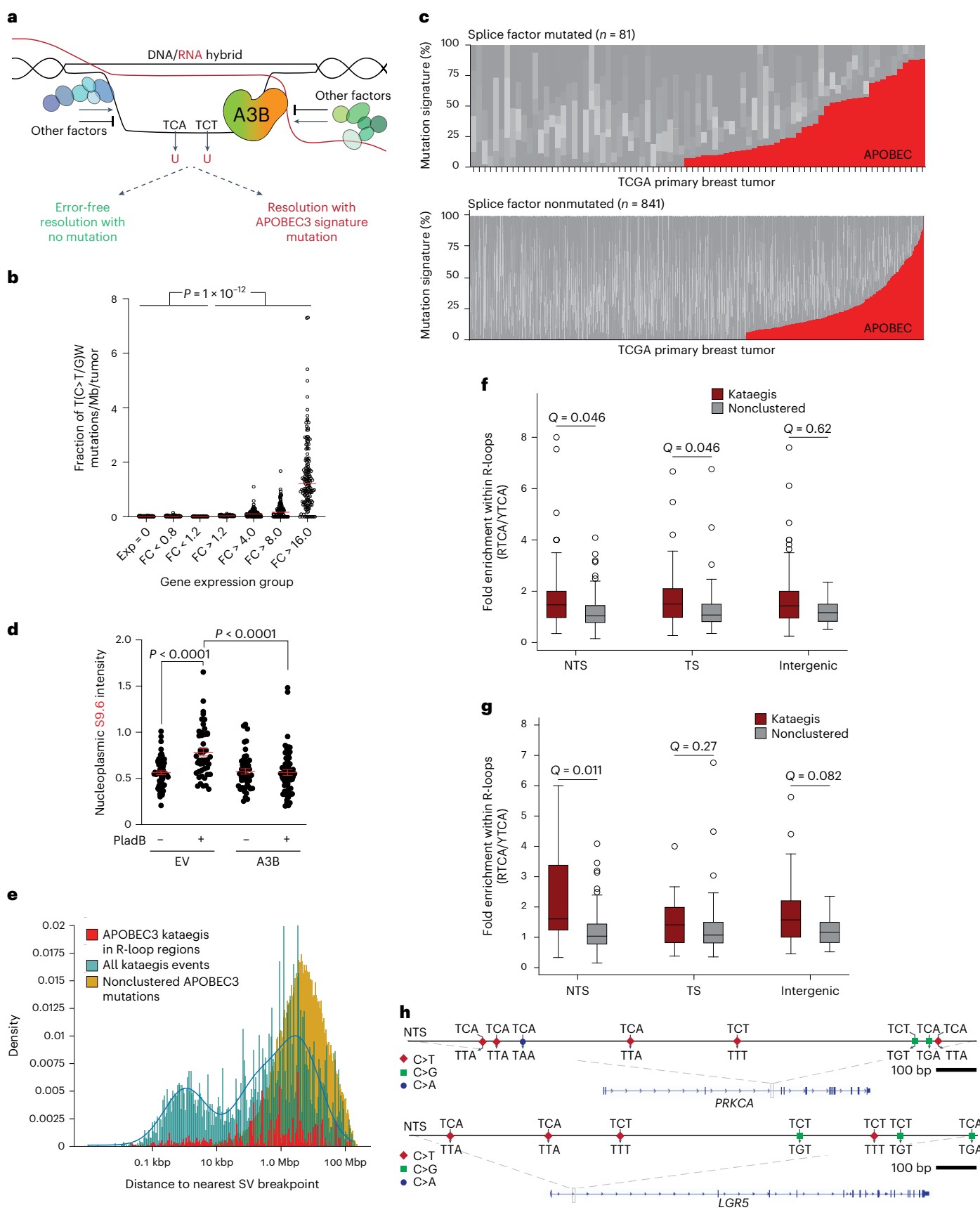

including AID-catalyzed antibody diversification[7] and Cas-mediated cytosine base editing[57], exposed ssDNA cytosines in R-loop structures can be deaminated by A3B, and then the resulting uracils become substrates for multiple competing DNA repair/replication processes. This can lead to error-free repair as well as multiple error-prone/mutagenic outcomes ranging from signature mutations to DNA breaks and larger-scale chromosome aberrations.

Although we and others[15,16] did not find a general association between APOBEC3 signature mutation and gene expression levels, a recent study reported higher APOBEC3 mutation densities on the nontranscribed strand of actively expressed genes in multiple cancer types[58]. Our studies indicate that the nontranscribed strand of R-loop regions is particularly susceptible to APOBEC3 mutagenesis including kataegis. Moreover, our studies show that transcription-associated defects in cancer such as gross overexpression and splice factor malfunction additionally increase the probability of APOBEC3 mutagenesis. These mechanistic links are further supported by data showing that A3B can suppress the increase in R-loop formation caused by treating cells with the splicing inhibitor Plad B. Despite the possibility that other APOBEC3 enzymes (most notably A3A) may also contribute to R-loop-associated mutations, a specific role for A3A in R-loop homeostasis is disfavored because its overexpression did not affect R-loop levels. In addition, most APOBEC3 kataegic events observed far away from sites of structural variation are enriched for mutations in A3B-associated 5′-RTCW motifs and not in A3A-associated 5′-YTCW motifs.

In addition to the direct mechanism discussed above, a potentially overlapping alternative is A3B-dependent recruitment of proteins known to promote R-loop resolution. Such interactions could be direct or bridged, for instance, by RNA or ssDNA. In support of this possibility, the A3B separation-of-function mutant Mut2, which is deficient in nucleic acid binding but proficient in nuclear import and DNA deamination, is less capable of interacting with several R-loop-associated factors. Moreover, although our studies here focused on strong A3B interactors, several weaker binders such as the helicase DHX9 might be relevant. This R-loop helicase was reported recently as a regulator of A3B antiviral activity[59]. Such factors may help explain the subset of genes that exhibit decreased R-loop levels in the absence of A3B. Further studies on A3B regulation of R-loop homeostasis will undoubtedly illuminate additional R-loop biology, provide insights into the normal physiological functions of A3B and define new drug-actionable nodes in A3B-overexpressing tumor types such as breast cancer.

## Online content

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

[1]Howard Hughes Medical Institute, University of Minnesota, Minneapolis, MN, USA. [2]Masonic Cancer Center, University of Minnesota, Minneapolis, MN, USA. [3]Institute for Molecular Virology, University of Minnesota, Minneapolis, MN, USA. [4]Department of Biochemistry, Molecular Biology and Biophysics, University of Minnesota, Minneapolis, MN, USA. [5]Sir William Dunn School of Pathology, University of Oxford, Oxford, UK. [6]Department of Molecular Biosciences, University of Texas at Austin, Austin, TX, USA. [7]Department of Life Science and Multidisciplinary Genome Institute, Hallym University, Chuncheon, Republic of Korea. [8]Department of Molecular and Cell Biology, University of Leicester, Leicester, UK. [9]Biochemistry and Structural Biology Department, University of Texas Health San Antonio, San Antonio, TX, USA. [10]Howard Hughes Medical Institute, University of Texas Health San Antonio, San Antonio, TX, USA. [11]Institute for Health Informatics, University of Minnesota, Minneapolis, MN, USA. [12]Department of Cellular and Molecular Medicine, UC San Diego, La Jolla, CA, USA. [13]Department of Bioengineering, UC San Diego, La Jolla, CA, USA. [14]Moores Cancer Center, UC San Diego, La Jolla, CA, USA. [15]Livestrong Cancer Institutes, Dell Medical School, University of Texas at Austin, Austin, TX, USA. [16]These authors contributed equally: Jennifer L. McCann, Agnese Cristini. [17]These authors jointly supervised this work: Kyle M. Miller, Natalia Gromak, Reuben S. Harris. ✉e-mail: kyle.miller@austin.utexas.edu; natalia.gromak@path.ox.ac.uk; rsh@uthscsa.edu

## Methods

### Cell lines and culturing

U2OS cells were obtained from ATCC (HTB-96) and were maintained in McCoy's 5A Medium (Thermo Fisher Scientific, 16600082) supplemented with 10% FBS (Gibco) and 0.5% Penicillin/Streptomycin (Pen/Strep; 50 units). U2OS shCtrl and shA3B cell lines were made using previously described shCtrl and shA3B lentiviral constructs, viral production and transduction methods and puromycin selection 1 µg ml[-1] (ref. 47). U2OS pcDNA3.1-A3-3xHA stable lines were made via linear (NruI digested) transfection and selection using 800 µg ml[-1] G418. HEK 293T cells were obtained from ATCC (CRL-3216) and were maintained in RPMI (Hyclone) supplemented with 10% FBS (Gibco) and 0.5% Pen/Strep (50 units). MCF10A cells were obtained from ATCC (CRL-10317) and were maintained in DMEM/F12 (Invitrogen, 11330-032) supplemented with 5% horse serum (Invitrogen, 16050-122), 20 ng ml[-1] EGF (Peprotech), 0.5 µg ml[-1] hydrocortisone (Sigma-Aldrich, H-0888), 100 ng ml[-1] cholera toxin (Sigma-Aldrich, C-8052), 10 µg ml[-1] insulin (Sigma-Aldrich, I-1882, I-9278) and 0.5% Pen/Strep (Invitrogen, 15070-063). MCF10A-TREx-A3B-eGFP were maintained in the same MCF10A media described above with the addition of 100 µg ml[-1] Normocin. S9.6 Hybridoma cells were obtained from ATCC (HB-8730) and were maintained in DMEM (Hyclone) supplemented with 10% FBS (Gibco) and 0.5% Pen/Strep (50 units). HeLa cells were obtained from N.J. Proudfoot (University of Oxford) and were maintained in DMEM (Sigma-Aldrich) supplemented with 10% FBS (Sigma-Aldrich) and 0.5% Pen/Strep (Invitrogen, 15070-063). MCF10A *A3B* KO cell line was engineered by transduction with pLentiCRISPR expressing a gRNA targeting both the *A3A* and *A3B* genes (Supplementary Table 1). Cells were selected with puromycin and seeded for single-cell cloning. Deletion mutant lines were identified by PCR using primers amplifying unique sequences within the *A3B* gene and/or the *A3A/B* junction (primers in ref. 60; Supplementary Table 1) and confirmed by qPCR and immunoblots. U2OS *A3B* KO cell line was engineered by transduction with pLentiCRISPR expressing a gRNA targeting exon 3 of *A3B* (Supplementary Table 1). Cells were selected with puromycin and seeded for single-cell cloning. Biallelic *A3B* KO was confirmed by PCR using primers spanning the gRNA target region and subsequent sequencing in addition to immunoblotting (Supplementary Table 1). HeLa RNAi was performed in six-well plates 24 h after seeding with 22 nM siRNA and Lipofectamine 2000, and after 6 h, the medium was changed. A second transfection was performed 48 h after seeding using the same experimental setting, and then cells were reseeded 24 h before the experiment. siRNAs were purchased from GE Healthcare targeting luciferase (D-001400-01) or *A3B* (Supplementary Table 1).

### Plasmids and cloning

C-terminal eGFP epitope-tagged plasmids used in this study were described previously[47,61–63]. Catalytic mutant A3B-E255A and shRNA-resistant derivatives were made using standard site-directed mutagenesis. C-terminal 3x-HA epitope-tagged plasmids used in this study were described previously[64], and shRNA-resistant derivatives were made using standard site-directed mutagenesis. C-terminal 2xStrep and 3xFlag-tagged eGFP and A3B constructs used for proteomics were described[65]. cDNA for some interactors constructs were ordered from Origene (RC216648, RC204785 and RC214037) while the rest were cloned from 293T cDNA. 4/TO-C-terminal 3xFlag-tagged interactor constructs used for IP were generated using standard cloning techniques. A3B Mut1 (E22Y/E24R/Y28S/G29R/S31N/Y32T) (ref. 48) and Mut2 (Y13D/Y28S/Y83D/W127S/Y162D/Y191H) (ref. 46) were subcloned into 5/TO-A3B-GFP as a *Hind*III and *Kpn*I fragment from a reported construct or gBlock (IDT), respectively. The pcDNA5/FRT/TO-mCherry-RNaseHI-D10R-E48R plasmid was reported previously[33,66]. All oligonucleotide sequences used to generate new constructs are listed in Supplementary Table 1.

### AP–MS

The 293T cells were transfected with pcDNA4/TO-A3B-2xStrep-3xFlag or eGFP-2xStrep-3xFlag using Transit LT1 (Mirus). Cells were collected in 1× PBS 48 h post-transfection. Cells were washed two times in 1× PBS followed by lysis (50 mM Tris–HCl (pH 8.0), 1% Tergitol NP-40, 150 mM NaCl, 0.5% sodium deoxycholate, 0.1% SDS, 1 mM DTT, 1× protease inhibitor (Roche), RNase A and DNase). Lysates were subjected to sonication before clearing by centrifugation. Cleared lysates were then added to Strep-Tactin Superflow resin (IBA) followed by end-over-end rotation for 2 h at 4 °C. Following IP, the anti-Strep resin was washed three times in high-salt wash buffer (20 mM Tris–HCl (pH 8.0), 1.5 mM MgCl$_2$, 1 M NaCl, 0.2% Tergitol NP-40, 0.5 mM DTT and 5% glycerol) followed by three washes in low-salt wash buffer (same as high salt but with 150 mM NaCl). To remove detergents for proteomics submission, samples were subjected to three washes of no-detergent wash buffer (20 mM Tris–HCl (pH 8.0), 1.5 mM MgCl$_2$, 150 mM NaCl, 0.5 mM DTT and 5% glycerol). Protein was eluted from the resin in elution buffer (100 mM Tris–HCl (pH 8.0), 150 mM NaCl and 2.5 mM desthiobiotin). Samples were validated using immunoblotting, DNA deaminase activity assays (discussed below) and Coomassie staining. In-solution samples were analyzed by liquid chromatography–mass spectrometry/mass spectrometry (LC–MS/MS) at the Harvard Proteomic Core (A3B AP–MS data are in Supplementary Table 1). CRAPome repository was used to remove likely nonspecific interactions before S9.6 IP overlap analysis[67].

For A3B-mycHis purification, 293T cells grown in RPMI were transfected in 15 cm plates with 20 µg of plasmid using a 3:1 ratio of polyethyleneimine (Polysciences PEI 40k, 24765) to DNA. Twenty-four hours post-transfection, the cells were collected by trypsinization, washed in PBS–EDTA and collected by centrifugation. Cell pellets were frozen at −80 °C. For purification, cells were lysed in 25 mM HEPES (pH 7.4), 300 mM sodium chloride, 20 mM imidazole, 10 mM magnesium chloride, 0.5 mM TCEP, 0.1% Triton X-100, 20% glycerol and Roche complete protease inhibitors. Lysis was performed by 2 min of sonication at a 40% duty cycle. Following sonication, RNase A was added to 100 µg ml[-1] and Benzonase to 5 units per ml followed by incubation at 37 °C for an hour. Cell debris was pelleted by centrifugation at 16,000*g* for 30 min at 25 °C. The supernatant was collected, and sodium chloride was added to a final concentration of 1 M. APOBEC3B-mycHis was allowed to bind to 50 µl nickel-NTA resin per 10 × 15 cm plates for 2 h at 4 °C. The resin was collected in BioRad polyprep columns and washed with 25 mM HEPES (pH 7.4), 300 mM sodium chloride, 0.1% Triton X-100, 40 mM imidazole and 20% glycerol. Protein was eluted in the same buffer with the addition of TCEP to 1 mM and 300 mM imidazole. Purity and concentration were assessed by PAGE with Coomassie stain with gels imaged using a LI-COR Odyssey instrument.

As an alternative procedure for A3B-mycHis purification (Extended Data Fig. 5d), Expi293F cells grown in Expi293 Expression Medium were transfected in 60 ml cultures according to the manufacturer's standard protocol (Thermo Fisher Scientific). Seventy-two hours post-transfection, the cells were collected by centrifugation, washed in PBS–EDTA and pelleted. Cell pellets were frozen at −80 °C. The AP procedure is the same as that described above except RNase A and Benzonase treatment was for 2 h, and APOBEC3B-mycHis was allowed to bind to 50 µl nickel-NTA resin for 2 h at room temperature.

### A3B activity assays

Deamination reactions were performed at 37 °C for 2 h using whole cell lysate, 4 pmol of 3′-fluorescein-labeled oligonucleotide, 0.025 U uracil DNA glycosylase (UDG), 1× UDG buffer (NEB) and 1.75 U RNase A. Reaction mixtures were treated with 100 mM NaOH at 95 °C for 10 min to achieve complete backbone breakage. Reaction mixtures were separated on 15% Tris–borate–EDTA (TBE)-urea gels to separate the substrate from the product. Gels were scanned using a Typhoon FLA-7000 image reader.

A3B activity assays with purified A3B-mycHis or mutants were performed similarly as above in 25 mM HEPES (pH 7.4), 50 mM NaCl, 0.4 U ml$^{-1}$ Roche RNase Inhibitor for the indicated amounts of time at 37 °C. Reactions were stopped at 95 °C for 5 min then UDG was added to 0.4 U per reaction and incubated for 10 min at 37 °C. Sodium hydroxide was added to 100 mM, and reactions were heated to 95 °C for 5 min. An equivalent volume of 80% formamide in 1× TBE with xylene cyanol and bromophenol blue was added, and reactions were heated again to 95 °C for 3 min to ensure the melting of double-stranded regions of DNA/RNA. Products were separated by 15% denaturing PAGE and digitally scanned using a LI-COR Odyssey imager. Quantitation was performed using LI-COR Odyssey software.

### Electrophoretic mobility shift assays

For competition experiments, EMSAs were performed in 25 mM HEPES (pH 7.4), 50 mM sodium chloride and 0.4 U μl$^{-1}$ Roche RNase inhibitor. For R-loop substrate EMSAs, NEB2 buffer (no BSA) was used to promote the annealing of substrates. Oligonucleotide substrates (illustrated in Fig. 7a and full sequences listed in Supplementary Table 1) were annealed by heating the components to 95 °C in a heat block and then permitted to cool to >10 °C below the predicted annealing temperature under the buffer conditions (UNAFold). Reactions were set up with labeled oligo in the tube to which A3B or mutants were added to the appropriate concentration. Reactions were incubated at room temperature for 5 min, and then either run or competitor was added with an additional 10 min incubation at room temperature. To run the gels, an equal volume of agarose gel loading dye (30% polyethylene glycol, 1× TBE and dyes) was added to each reaction mix and half of each reaction was loaded on the gel. Gels were imaged using a LI-COR Odyssey and quantitated with LI-COR Odyssey software.

### Drug treatments

PMA (Sigma-Aldrich, P8139) was added to media at 25 ng ml$^{-1}$ at 37 °C with 5% $CO_2$ for denoted time. JQ1 (Tocris, 4499) was added to media at 0.5 μM at 37 °C with 5% $CO_2$ for 4 h unless denoted otherwise. Triptolide (Tocris, 3253; Selleckchem, S3604) was added to media at 1 μM at 37 °C with 5% $CO_2$ for 4 h unless denoted otherwise. Flavopiridol (Selleckchem, S1230) was added to media at 1 μM at 37 °C with 5% $CO_2$ for 1 h unless denoted otherwise. Dox (MP Biomedicals, 198955) was added to media at 1 μg ml$^{-1}$ at 37 °C with 5% $CO_2$ for 24 h unless denoted otherwise. The splicing inhibitor, Plad B (ref. 54; Tocris, 6070), was added to media at 5 μM at 37 °C with 5% $CO_2$ for 2 h unless noted otherwise.

### Antibodies

Primary antibodies used in these experiments were α-Tubulin (Sigma-Aldrich, T5168; Abcam, ab6046 and ab4074), α-A3B (5210-87-13, custom[68]), α-Flag (Sigma-Aldrich, F1804), α-Topoisomerase I (Abcam, ab109374), α-Lamin B1 (Abcam, ab16048), α-IgG2a (Sigma-Aldrich, M5409), α-HA (Cell Signaling Technology, 3724S), α-GFP (Abcam, ab290, Lot GR3251545 and GR3270983 for ChIP), α-mCherry (Abcam, ab167453) α-HNRNPUL1 (gift from S. Wilson, University of Sheffield, UK), α-rabbit IgG Isotype Control (Invitrogen, 02-6102, lot RI238244), α-RNA/DNA Hybrid S9.6 (Kerafast, ENH001 or obtained in house from a hybridoma cell line[69,70]), α-dsDNA (Abcam, ab27156) and α-gamma-H2AX (Novus, NB100-384). Secondary antibodies used were α-rabbit IRdye 800CW (LI-COR, 827-08365), α-mouse IRdye 680LT (LI-COR, 925-68020), α-rabbit HRP (Cell Signaling Technology, 7074P2 or Sigma-Aldrich, A0545) and α-mouse HRP (Cell Signaling Technology, 7076P2 or Sigma-Aldrich, A8924), Alexa Fluor 488 goat anti-mouse IgG (Invitrogen, A-11029), Alexa Fluor 594 goat anti-mouse IgG (Invitrogen, A-11032), Alexa Fluor 488 goat anti-rabbit IgG (Invitrogen, A-11034), Alexa Fluor 647 goat anti-mouse IgG (Invitrogen, A-21236) and Alexa Fluor 594 goat anti-rabbit IgG (Invitrogen, A-11037).

### Co-IP experiments

Semi-confluent 293T cells were transfected with plasmids using TransIT-LT1 (Mirus) per the manufacturer's protocol. Cells were collected in 1× PBS 48 h post-transfection. Cells were washed two times in 1× PBS followed by lysis (150 mM NaCl, 50 mM Tris−HCl (pH 8.0), 0.5% Tergitol, 1× protease inhibitor (Roche), RNase and DNase). Cells were vortexed vigorously and incubated at 4 °C for 30 min before clearing by centrifugation. Cleared lysates were then added to anti-Flag M2 Magnetic Beads (Sigma, M8823) followed by end-over-end rotation overnight at 4 °C. Beads were then washed three times in lysis buffer followed by elution in elution buffer (lysis buffer + 0.15 mg ml$^{-1}$ Flag Peptide (Sigma-Aldrich)).

### EdU and PI staining

Semi-confluent MCF10A or U2OS cells were treated with 10 μM EdU for 2 h before collection. Click-iT Plus EdU Alexa Fluor 488 Flow Cytometry Assay Kit (Invitrogen, C10632) with the addition of FxCycle PI/RNase Staining Solution (Invitrogen, F10797) was used per manufacturer's protocol, and flow cytometry of a minimum of 10,000 cells per condition was performed on LSRFortessa with subsequent analysis with Flow Jo version 10.8.1 (BD).

### RNA/DNA hybrid slot blots

RNA/DNA hybrid slot-blot experiments were performed based on a standard protocol[42,71]. RNase H sensitivity was carried out by incubation with 2 U of RNase H (NEB, M0297) per microgram of genomic DNA for 18 h at 37 °C. S9.6 and dsDNA samples were run on the same membrane and cut for primary antibody incubation. Images were acquired with LI-COR Odyssey Fc. Exposure settings for each antibody were consistent within experiments. S9.6 signal relative to dsDNA was quantified using Image Studio software (LI-COR Biosciences). Quantification was performed using S9.6 and dsDNA signal within in the linear range and normalized to WT, untreated or control samples.

### mRNA RT−qPCR

Isolation of polyA+ mRNA (High Pure RNA Isolation Kit; Roche Life Science, 11828665001), RT to generate cDNA (Transcriptor RTase; Roche Life Science, 3531317001) and qPCR were done according to manufacturer's protocols. The abundance of various mRNAs was quantified by RT−qPCR relative to the stable housekeeping transcript, *TBP*. Gene-specific primers have been described[72] and are listed in Supplementary Table 1.

### DRIP

DRIP was performed using the S9.6 antibody[29,69,73]. Noncrosslinked nuclei were lysed in nuclear lysis buffer (50 mM Tris−HCl (pH 8.0), 5 mM EDTA, 1% SDS) and subjected to Proteinase K treatment (Sigma-Aldrich) for 3 h at 55 °C. Genomic nucleic acids were precipitated with isopropanol, washed in 75% ethanol and sonicated in IP dilution buffer (16.7 mM Tris−HCl (pH 8.0), 1.2 mM EDTA, 167 mM NaCl, 0.01% SDS, 1.1% Triton X-100) with Diagenode Bioruptor to an average length of 500 base pair (bp). Following addition of protease inhibitors (0.5 mM PMSF, 0.8 μg ml$^{-1}$ pepstatin A, 1 μg ml$^{-1}$ leupeptin), sonicated genomic nucleic acids were precleared with protein A Dynabeads (Invitrogen) blocked with acetylated BSA (Sigma-Aldrich, B8894). A total of 10 μg were subjected to S9.6 or no antibody IP overnight at 4 °C. RNase H sensitivity was carried out by incubation with 1.7 U RNase H (NEB, M0297) per microgram of genomic DNA for 3 h at 37 °C before IP. Retrieval of the immunocomplexes with beads, washes and elution was performed as described for ChIP. Samples were incubated with Proteinase K (Sigma-Aldrich) at 45 °C for 2 h. For qPCR analysis, DNA was purified with QIAquick PCR purification kit (QIAGEN) and analyzed by qPCR with Rotor-Gene Q and QuantiTect SYBR green (QIAGEN). The amount of immunoprecipitated material at a particular gene region was calculated as the percentage of input after subtracting the

background signal (no antibody control). The primers used for DRIP are listed in Supplementary Table 1. For DRIP–seq analysis, multiple S9.6 IPs were pooled. DNA was purified with MinElute PCR purification kit (QIAGEN) and subjected to library preparation and sequencing on a NovaSeq 6000 with 150 bp paired-end reads at Oxford Genomics Center (WTCHG, University of Oxford).

## RNA/DNA hybrid and protein co-IP

DNA/RNA hybrid co-IPs were carried out using S9.6 antibody[29,69,74]. Non-crosslinked nuclei were lysed in RSB buffer (10 mM Tris–HCl (pH 7.5), 200 mM NaCl, 2.5 mM MgCl$_2$) with the addition of 0.2% sodium deoxycholate, 0.1% SDS, 0.05% sodium lauroyl sarcosinate and 0.5% Triton X-100. Nuclear extracts were then sonicated with Diagenode Bioruptor and diluted in RSB with 0.5% Triton X-100 (RSB + T). RNA/DNA hybrids were immunoprecipitated for 2 h at 4 °C with BSA-blocked protein A Dynabeads (Invitrogen) conjugated with the S9.6 antibody in the presence of 1.2 ng of RNase A (PureLink, Invitrogen). Washes of the immunocomplexes were carried out with RSB + T (four times) and RSB (two times). Immunocomplexes were then eluted by incubating at 70 °C with 1× LDS (Invitrogen) and 100 mM DTT for 10 min. Where indicated, IPs were performed in the presence of 1.3 µM DNA/RNA hybrid competitors[70] (Supplementary Table 1). The same procedure was used for protein co-IP, and anti-GFP antibody (Abcam, ab290) was used instead of S9.6 antibody. Proteins were separated by SDS–PAGE and immunoblotted with α-A3B (5210-87-13; ref. 68), α-Topoisomerase I (Abcam, ab109374), α-Lamin B1 (Abcam, ab16048), α-GFP (Abcam, ab290) and α-HNRNPUL1 (gift from S. Wilson, University of Sheffield, UK) antibodies. For RNA/DNA hybrid slot-blot analysis, A3B-eGFP co-IP was performed starting from 350 µg of proteins following the same procedure without the addition of RNase A. Immunocomplexes were eluted in 1% SDS and 0.1 M NaHCO$_3$ for 30 min at room temperature, and nucleic acids were precipitated overnight with isopropanol and glycogen (Roche) after Proteinase K digestion (Sigma-Aldrich) for 2 h at 45 °C. RNase H sensitivity was performed by incubating with 7.5 U of RNase H (NEB, M0297) for 2.5 h at 37 °C.

## ChIP

ChIP experiments were done by crosslinking cells with 1% formaldehyde at 37 °C for 15 min before the reactions were quenched with 0.125 M glycine for 5 min[29,73]. Nuclei were isolated by lysing cells with cell lysis buffer (5 mM PIPES (pH 8.0), 85 mM KCl, 0.5% NP-40 supplemented with 0.5 mM PMSF and 1× complete EDTA-free protease inhibitors; Sigma-Aldrich). Nuclear pellets were then resuspended in nuclear lysis buffer (50 mM Tris–HCl (pH 8.0), 5 mM EDTA, 1% SDS supplemented with 0.5 mM PMSF and 1× complete EDTA-free protease inhibitors; Sigma-Aldrich) before sonication (Diagenode Bioruptor). Insoluble chromatin was removed by centrifugation. Soluble chromatin was then diluted in ChIP IP buffer (16.7 mM Tris–HCl (pH 8.0), 1.2 mM EDTA (pH 8.0), 167 mM NaCl, 0.01% SDS, 1.1% Triton X-100 supplemented with 0.5 mM PMSF and 1× complete EDTA-free protease inhibitors; Sigma-Aldrich) and precleared by incubation with protein A Dynabeads (Invitrogen) blocked with acetylated BSA (Sigma-Aldrich, B8894). Precleared chromatin was then incubated with α-GFP antibody (Abcam, ab290, lot GR3251545 and GR3270983). BSA-blocked protein A Dynabeads were then added to collect immunocomplexes and washed once with buffer A (20 mM Tris–HCl (pH 8.0), 2 mM EDTA, 0.1% SDS, 1% Triton X-100 and 0.150 M NaCl), once with buffer B (20 mM Tris–HCl (pH 8.0), 2 mM EDTA, 0.1% SDS, 1% Triton X-100 and 0.5 M NaCl), once with buffer C (10 mM Tris–HCl (pH 8.0), 1 mM EDTA, 1% NP-40, 1% sodium deoxycholate and 0.25 M LiCl) and then twice with buffer D (10 mM Tris–HCl (pH 8.0) and 1 mM EDTA). Chromatin complexes were eluted in 1% SDS and 0.1 M NaHCO$_3$. Samples were decrosslinked by incubating at 65 °C for at least 4 h in the presence of RNase A (PureLink, Invitrogen) and NaCl (0.3 M) and digested with proteinase K (Sigma-Aldrich) for 2 h at 45 °C. DNA purification and qPCR analysis were performed as described for DRIP. The primers used for ChIP are listed in Supplementary Table 1.

For ChIP–seq analysis, multiple ChIP IPs were pooled. DNA was purified with MinElute PCR purification kit (QIAGEN) and subjected to library preparation and sequencing on a NovaSeq 6000 with 150 bp paired-end reads at Oxford Genomics Center (WTCHG, University of Oxford).

## IF for R-loop analysis

Experiments were performed similar to reported procedures[33,66] with details as follows.

**S9.6 IF analysis.** U2OS or MCF10A cells either WT or deficient for A3B were analyzed for S9.6 IF as indicated. Untreated cells were analyzed or treatments were performed as described. Treatment with the transcription initiation inhibitor (triptolide, final concentration 1 µM) was performed for 4 h, or cells were transfected with indicated constructs and either treated with JQ1 (final concentration 0.5 µM in DMSO) or equivalent DMSO concentration only control for 4 h or Plad B (final concentration 5 µM) for 2 h. After each indicated treatment, cells were fixed with 100% ice-cold methanol at 4 °C for 10 min, a common fixation method for S9.6 and R-loops[33,75–77], followed by washing three times with PBS at room temperature. For in vitro RNase H treatment, fixed cells were washed with nuclease-free water to remove PBS and treated with 150 U ml$^{-1}$ RNase H in 1× RNase H reaction buffer (NEB, M0297). Cells were incubated for 2 h at 37 °C followed by two 5 min washes with 1× PBS. Untreated samples were similarly treated except using 1× RNase H reaction buffer without enzyme. To detect S9.6, cells were then blocked with 3% BSA/PBS at room temperature for 1 h and incubated with S9.6 antibody (Kerafast, ENH001; 1:200) at 4 °C for 18 h. Some samples were costained with the DNA damage marker γH2AX (Novus, NB100-384; 1:500). Following primary antibody incubation, cells were washed with PBS three times for 5 min and incubated with appropriate secondary antibody for each primary antibody in 3% BSA/PBS blocking buffer at room temperature for 1 h. Cells were then washed in PBS three times for 5 min, and each coverslip was mounted on a 12 mm glass slide using Vectashield mounting medium containing DAPI (Vector Laboratories, H-1200). Samples were analyzed using a Fluoview FV 3000 confocal microscope (Olympus; Miller Laboratory) or Nikon AR1 (University of Minnesota Imaging Center), and nucleoplasmic S9.6 signal was quantified using Image J (v 1.48) as described in Quantification and statistical analysis subsection below. All constructs were expressed to similar levels.

**mCherry-RNaseH1-mutant IF analysis.** WT or *A3B* KO U2OS cells were transfected with mCherry-RNaseH1-D10R-E48R catalytic mutant (mCherry-RNaseH1 mut; refs. 33,66,75) and allowed to incubate for 48 h before treatment. Cells expressing mCherry-RNaseH1 mut were either untreated, treated with JQ1 (final concentration 0.5 µM in DMSO) or treated with the equivalent DMSO concentration as a control for 4 h. Following treatment, cells were fixed with 100% ice-cold methanol at 4 °C for 10 min followed by washing three times with PBS at room temperature. Cells on individual coverslips from each condition were mounted on a 12 mm glass slide using Vectashield mounting medium containing DAPI (Vector Laboratories, H-1200). Samples were then analyzed using a Fluoview FV 3000 confocal microscope (Olympus; Miller Laboratory), and mCherry-RNaseH1 mut signal was detected with a 561 nm diode laser and appropriate filter with high-sensitivity Peltier-cooled GaAsP spectral confocal detector. For experiments performed in U2OS *A3B* KO cells, WT A3B-eGFP or catalytic mutant A3B-E255A-eGFP was cotransfected with mCherry-RNaseH1 mut and cells expressing both constructs were analyzed. For GFP signal of ectopically expressed A3B, samples were analyzed with a 488 nm diode laser and appropriate filter with high-sensitivity Peltier-cooled GaAsP spectral confocal detector. DAPI signals were detected using a 405 nm diode laser and appropriate filter with high-sensitivity Peltier-cooled GaAsP spectral confocal detector. Equal expression between samples was determined by quantification of the total nuclear fluorescence signal for mCherry using Image J and western blotting for

both mCherry-RNaseH1 mut and A3B WT and E255A. Quantification of nucleoplasmic mCherry-RNaseH1 mut was performed as described in the Quantification and statistical analysis subsection below.

## Immunoblot analysis

For immunoblotting assays, the samples were combined with 2.5× SDS–PAGE loading buffer. Samples were separated by a 4–20% gradient SDS–PAGE gel and transferred to PVDF-FL membranes (Millipore). Membranes were blocked in blocking solution (5% milk + PBS supplemented with 0.1% Tween 20) and then incubated with primary antibody diluted in blocking solution. Secondary antibodies were diluted in blocking solution + 0.02% SDS. Membranes were imaged with a LI-COR Odyssey instrument or film.

## ChIP–seq and DRIP–seq data processing

Adapters were trimmed with Cutadapt version 1.13 (ref. [78]) in paired-end mode with the following parameters: -q 15, 10 –minimum-length 10 -A AGATCGGAAGAGCGTCGTGTAGGGAAAGAGTGT -a AGATCGGAA-GAGCACACGTCTGAACTCCAGTCA. Obtained sequences were mapped to the human hg38 reference genome with STAR version 2.6.1d (ref. [79]) and the parameters --runThreadN 16 --readFilesCommand gunzip -c –k --alignIntronMax 1 --limitBAMsortRAM 20000000000 --outSAMtype BAM SortedByCoordinate. Properly paired and mapped reads (-f 3) were retained with SAMtools version 1.3.1 (ref. [80]). PCR duplicates were removed with Picard MarkDuplicates tool. Reads mapping to the DAC Exclusion List Regions (accession: ENCSR636HFF) were removed with Bedtools version 2.29.2 (ref. [81]). FPKM-normalized bigwig files were created with deepTools version 2.5.0.1 (ref. [82]) bamCoverage tool with the parameters –bs 10 –p max -e --normalizeUsing RPKM. ChIP–seq and DRIP–seq peaks were called with MACS2 version 2.1.1.20160309 (ref. [83]) and the following parameters: callpeak -f BAMPE -g 2.9e9 -B -q 0.01 –call-summits. Each IP and its respective input were used as treatment and control, respectively. DRIP–seq differential peak calling was performed with MACS2 bdgdiff tool.

## Transcription unit annotation

Gencode V31 annotation, based on the hg38 version of the human genome, was used to extract the location of the transcription units. All genes were taken from the most 5′ transcription start site to the most 3′ poly(A) site/transcription end site. The eRNAs annotation based on the hg38 version of the human genome was taken from the FANTOM5 database.

## Metagene profiles

Metagene profiles were generated from FPKM-normalized bigwig files with Deeptools2 computeMatrix tool with a bin size of 10 bp, and the plotting data were obtained with plotProfile –outFileNameData tool. Graphs were then created with GraphPad Prism 8.3.1.

## RNA-seq data processing

RNA-seq data from ref. [30] were processed as follows: adapters were trimmed with Cutadapt in single-end mode with the following parameters: -q 15, 10 –minimum-length 10 -a AGATCGGAAGAGCACACGTCT GAACTCCAGTCA –max-n 1. The trimmed reads were mapped to the human hg38 reference genome with STAR and the parameters --runThreadN 16 --readFilesCommand gunzip -c –k --limitBAMsortRAM 20000000000 --outSAMtype BAM SortedByCoordinate. SAMtools was used to retain only properly mapped reads (-F 4). Gene expression level (transcripts per million) was calculated with Salmon version 0.13.1 (ref. [84]) and the Gencode V31 annotation. For each gene, only the highest expressed transcript was retained.

## APOBEC mutation and gene expression

Whole-exome sequencing and RNA-seq datasets for all primary breast tumor specimens ($n = 977$) and normal breast tissues ($n = 111$) in TCGA were downloaded from the Broad Institute analysis pipeline through the Firehose GDAC resource (http://gdac.broadinstitute.org/). Similarly, whole-genome sequencing datasets for all primary breast tumor samples ($n = 794$) in the ICGC were downloaded from the ICGC data portal (https://dcc.icgc.org/). Because ICGC tumors lack corresponding RNA-seq data, expression values for genes in normal breast tissues were obtained by averaging available GTEx data ($n = 29,589$ genes from 396 normal breast tissue samples; https://gtexportal.org/home/).

SBS mutations from TCGA and ICGC breast cancers were used for analyses here (that is, INDELs and other more complex somatic variations were filtered out)[55,85]. Tumor datasets were ranked initially by APOBEC mutation enrichment scores using established methods[49]. Enrichment score significance was assessed using a Fisher's exact test with Benjamini–Hochberg false discovery rate correction ($q < 0.05$). TCGA breast tumors with significant APOBEC mutational signature enrichments ($n = 154$ tumors) were used to test whether mutation load per megabase associates with differential gene expression (tumor versus normal tissue). Mean normal expression values for each gene from 111 normal breast tissues from the TCGA breast cancer dataset were used to generate a baseline for determining fold changes in gene expression in tumor tissues. For each of the 154 APOBEC signature-enriched tumors, we first generated the following seven gene expression groups: (1) tumor genes with expression values of 0 (gene number range = 722–3,903 and median = 2,144); (2) tumor genes with expression values less than 0.8-fold of the normals (first quartile of all genes in all tumors; gene number range = 787–4,369 and median = 2,688); (3) tumor genes with fold changes between 0.8- and 1.2-fold of the normals (covers from first quartile to third quartile of all genes; gene number range = 8,530–14,135 and median = 11,556); (4) tumor genes with fold changes between 1.2-fold and 4-fold above the normals (gene number range = 1,297–3,248 and median = 2,018); (5) tumor genes with fold changes between fourfold and eightfold above the normals (gene number range = 57–415 and median = 150); (6) tumor genes with fold changes between 8-fold and 16-fold above the normals (gene number range = 18–304 and median = 67); (7) tumor genes with fold changes greater than 16-fold above the normal (gene number range = 11–698 and median = 53). Finally, we calculated the fraction of APOBEC signature mutations (TCW to TTW or TGW) per tumor per megabase using the exon lengths of the genes in each group.

A similar analysis was done for ICGC tumor mutation versus GTEx expression values. Five expression groups were created—nonexpressed genes in (Exp = 0) and all other genes divided into expression quartiles. Only C-to-G and C-to-T mutations in TCW trinucleotide motifs were used in these analyses and were plotted for each expression group as (1) total number of T(C>G/T)W mutations, (2) total number of T(C>G/T) W mutations divided by the total number of all SBSs in a tumor and (3) total number of T(C>G/T)W mutations as a fraction of the total nucleotide size of genes' (exons and introns) in that expression group (mutations per megabase per tumor). Gene size information was downloaded from the UCSC table browser resource (https://genome.ucsc.edu/cgi-bin/hgTables), and correspond to the 'UCSC Genes, knownGene' reference set. All mutation calls and gene sizes/positions are relative to the hg19 human reference genome.

## Splice factor and APOBEC mutation analysis

TCGA mutation data were downloaded from Broad GDAC Firehose as above. In total, 119 splicing factor genes with recurring mutations in 33 cancers were used as the analysis gene set[86]. In total, 107 of the 119 genes had deleterious mutations in the TCGA BRCA dataset. These deleterious mutations included stop codon mutations, splice site mutations and insertion and deletion frameshift mutations. Trinucleotide contexts were calculated using the deconstructSigs package[87]. The APOBEC mutation signature in this analysis included all COSMIC SBS2 and/or SBS13 mutations[8,88]. Statistical analyses were done with Fisher's exact tests (with $a = 0.05$) and Student's $t$-tests as indicated.

## Housekeeping gene set analysis

We performed 100,000 random selections of 119 housekeeping genes from a previously defined set of 3,804 (ref. 89). In each iteration, we asked whether the selected 119 genes contained one or more deleterious mutations (that is, frameshift, stop codon or splice site) in each tumor of the TCGA breast cancer dataset ($n = 841$). From these iterations, the median number of mutated housekeeping genes was 35/119, the minimum was 0/119 and the maximum was 79/119. Similarly, from these iterations, the median number of tumors containing mutations in housekeeping genes was 15/841, the minimum was 0/841 and the maximum was 38/841. In contrast, from the 119 splice factor genes reported to be mutated across cancer, 107 of these were found to contain deleterious mutations in the TCGA breast cancer dataset; these 107 mutated splice factor genes are distributed across 81 breast tumors (that is, 81/922 TCGA tumors). For each of the 100,000 iterations, a Fisher's exact test was done for APOBEC3 signature enrichment, and, in all instances after correcting for multiple hypothesis testing, no significant enrichment was found for the housekeeping gene sets (Benjamini–Hochberg corrected $Q = 1.0$).

## APOBEC kataegis analysis using PCAWG WGS datasets

To analyze APOBEC-associated kataegis, the set of WGS breast adenocarcinomas was downloaded from the official PCAWG release (https://dcc.icgc.org/releases/PCAWG; $n = 198$). Kataegic events were detected using a sample-dependent intermutational distance (IMD) cutoff, which is unlikely to occur by chance given the mutational burden and mutational pattern of each sample[21,90]. SigProfilerSimulator (v 1.1.2) was used to generate a random distribution of the mutational spectra while maintaining the ±2 bp sequence context and the strand coordination within genic regions of each mutation[91]. This background model was used to determine the cutoff for the sample-dependent IMD by ensuring that 90% of clustered mutations occur within the original sample compared to the expected distribution ($Q < 0.01$). The heterogeneity of mutation rates across the genome and the confounding effects of copy number alterations and clonality were addressed by performing a 10 Mbp regional mutation density correction and by using a cutoff for the difference in variant allele frequencies between adjacent mutations in a clustered event (variant allele frequency difference <0.10) (ref. 21). Clustered events consisting of ≥3 or ≥5 mutations were classified as kataegis. Events that did not fall within 10 kbp of a detected structural variant breakpoint were used for nonstructural variation associated downstream analysis. All breakpoints were determined based on the official PCAWG release. Only base substitution mutations with TCW context were considered associated with APOBEC3 mutagenesis. A 1,000 bp window was included upstream and downstream of each DRIP–seq R-loop region to determine overlap with kataegic events. Mutation enrichment analysis was performed for each mutation by normalizing for the availability of a given motif (RTCA or YTCA) and the number of cytosines within ±20 bp (ref. 49). Additional analyses were conducted using R, Prism (v8.0), and the ggplot2 R package. Statistical significance between the tetranucleotide enrichments of kataegis and dispersed APOBEC3 mutation datasets was determined using a nonparametric Fisher's exact test, using an α of 0.05 ($P$ values reported in the text). Statistical significance for tetranucleotide mutation biases within samples containing overlaps of R-loop and kataegic events compared to dispersed mutations was assessed using a Mann–Whitney $U$ test ($Q$ values shown in each dot plot). The Cohen's $D$ effect size was calculated across all pairwise region comparisons to assess the skew of the distributions within R-loop-associated kataegis in comparison to all genome-wide kataegis.

## Quantification and statistical analysis

S9.6 and mCherry-RNaseH1-mut IF quantification was done as described in refs. 33,66. Specifically, mCherry-RNaseH1 mut or S9.6 images obtained on the confocal microscope were opened in Image J (v 1.48). For each image, nuclei of individual cells (≥60 cells per sample) were outlined using the selection tools function. Fluorescence intensity per area of each selection (entire nucleus) was measured using the measure function. Nucleoli for each nucleus were identified by importing DAPI overlayed channels for each image. The fluorescence intensity of nucleoli was measured by selecting DNA-free regions and using the measure function. Nucleoli-only intensity was subtracted from the total nuclear fluorescence signal to obtain the nucleoplasmic fluorescence intensity for either S9.6 or mCherry-RNaseH1 mut. These readings were normalized to control samples to obtain the 'relative fluorescence intensity'. For statistical analysis, one-way analysis of variance was used when comparing more than two groups followed by a Dunnett's multiple comparison test, a Mann–Whitney test or a two-tailed Student's $t$-test as indicated. Statistical analyses for bioinformatic studies are described above.

## Reporting summary

Further information on research design is available in the Nature Portfolio Reporting Summary linked to this article.

## Data availability

The Gene Expression Omnibus accession number for the ChIP–seq and DRIP–seq datasets reported in this paper is GSE148581. Questions regarding these sequencing data can be addressed to N.G. or R.S.H. The A3B AP–MS datasets are in Supplementary Table 1. Questions regarding these proteomic results can be addressed to R.S.H. Requests for materials and/or questions regarding any of the constructs, cell lines, microscopy results or other data described here can be addressed to R.S.H. Source data are provided with this paper.

## Code availability

No custom code or software was generated as part of the study. Details of all software packages used for data processing and/or analysis may be found in the Methods.

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

## Acknowledgements

We thank H. Gupta (UT Health San Antonio) and N.J. Proudfoot (Oxford University) for critically reading the manuscript, J. Becker and J. Duda (University of Minnesota) for corroborative localization data with A3B mutants, A. Taylor (UT Health San Antonio) for suggesting alternative purification procedures, the University of Minnesota Imaging Center for access to instrumentation and the Oxford Genomics Center at the Wellcome Center for Human Genetics (funded by Wellcome Trust grant 203141/Z/16/Z) for the generation and initial processing of the sequencing data. Studies in the Harris Lab were supported by NCI P01 CA234228 (to R.S.H.), NIAID R37 AI064046 (to R.S.H.) and a Recruitment of Established Investigators Award from the Cancer Prevention and Research Institute of Texas (CPRIT RR220053 to R.S.H.). NG Lab is supported by the Royal Society University Research Fellowship (BVD07340), Royal Society Enhancement Award (RGF\EA\180023) and EPA Research Fund (Sir William Dunn School of Pathology, University of Oxford) to N.G. and CRUK development fund (CRUK DF-0119) to A.C. and N.G. M.T. and S.M. are supported by the Wellcome Trust Investigator Award (WT210641/Z/18/Z to S.M.). KMM Lab was supported by NCI (RO1 CA198279, CA201268 and CA250905), Cancer Prevention and Research Institute of Texas (RP220330) and a postdoctoral fellowship (PF-22-092-01-DMC) from the American Cancer Society to A.S. LBA Lab was supported by US National Institutes of Health (R01 ES030993 and R01 ES032547). Salary support for J.L.M. was provided by an NSF Graduate Research Fellowship (00039202) and by HHMI. Salary support for M.C.J. was provided by T32 CA009138 and NCI F31 CA243306. Salary support for B.S. was provided by HHMI and the Ovarian Cancer Research Alliance (Mentored Investigator Grant 812337). Salary support for D.J.S. was provided by NIAID K99 AI147811. R.S.H. is the Ewing Halsell President's Council Distinguished Chair, a CPRIT Scholar and an investigator of the Howard Hughes Medical Institute at the University of Texas Health San Antonio.

## Author contributions

R.S.H., J.L.M., A.C. and N.G. conceived and designed these studies. J.L.M. and A.C. performed experiments unless otherwise noted. E.K.L. and S.L. made equal secondary contributions. E.K.L. generated U2OS knockdown and complement cell lines and assisted in tissue culture and genomic DNA isolations for dot-blot experiments. S.L., J.K., A.S. and K.M.M. designed, performed and quantified IF experiments. M.T. and S.M. conducted DRIP–seq/ChIP–seq data analysis. C.B. performed DRIP–qPCR validations and HeLa R-loop IP. B.S. assisted with cell culture experiments and R-loop quantification. M.R.B. assisted with cell culture studies. M.C.J., N.A.T., D.J.S., E.N.B. and L.B.A. performed bioinformatic analyses. M.A.C. contributed to biochemical experiments. R.S.H. and J.L.M. drafted the manuscript with input from all other authors.

## Competing interests

The authors declare no competing interests.

## Additional information

**Extended data** is available for this paper at https://doi.org/10.1038/s41588-023-01504-w.

**Correspondence and requests for materials** should be addressed to Kyle M. Miller, Natalia Gromak or Reuben S. Harris.

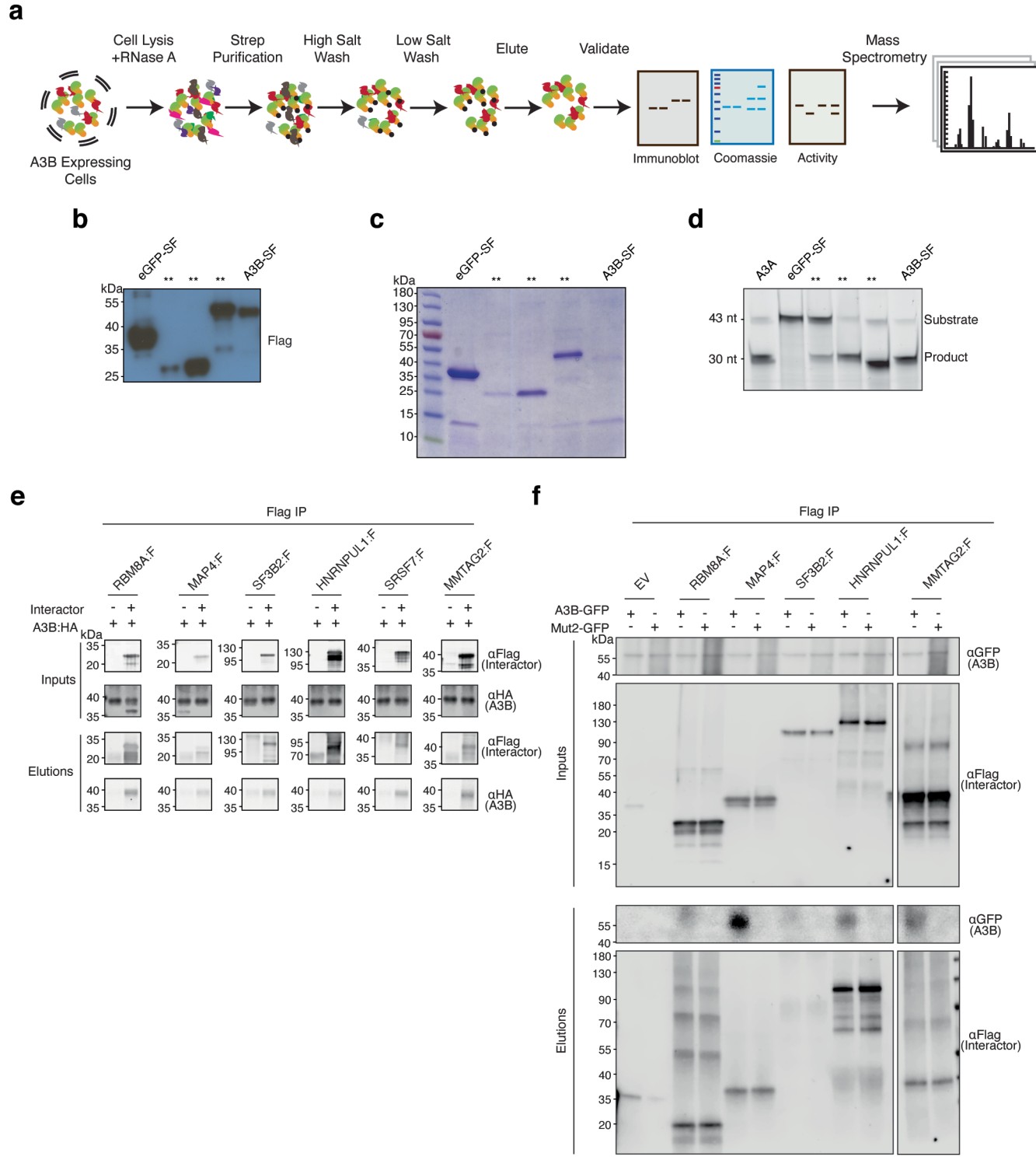

**Extended Data Fig. 1 | Controls for AP-MS experiments. a**, Schematic of the AP-MS workflow used to identify the cellular A3B interactome. A3B is shaded orange/green and cellular proteins are indicated by different shapes/colors. **b-c**, Anti-Flag immunoblot and Coomassie gel analysis of eGFP-SF and A3B-SF following affinity purification and prior to analysis by mass spectrometry (\*\*, samples not pertaining to this manuscript; representative images; n = 6 independent experiments). **d**, DNA deaminase activity of eGFP-SF and A3B-SF following affinity purification (purified A3A was used as a positive control; \*\*, samples not pertaining to this manuscript; representative images; n = 6 independent experiments). **e**, co-IP of indicated Flag-tagged interactors and HA-tagged A3B in 293 T cells (representative data from n = 2 independent

experiments). Upper immunoblots show the indicated proteins in whole cell lysates (input), and lower immunoblots show the Flag-immunoprecipitated samples (elution). kDa markers are shown the left of each blot and the primary antibody used for detection is shown to the right. **f**, co-IP of indicated Flag-tagged interactors and eGFP-tagged A3B or eGFP-tagged Mut2 from 293 T cells (representative data from n = 2 independent experiments). Upper immunoblots show the indicated proteins in whole cell lysates (inputs), and lower immunoblots show the anti-Flag immunoprecipitated samples (elutions). kDa markers are shown to the left of each blot and the primary antibody used for detection is shown to the right.

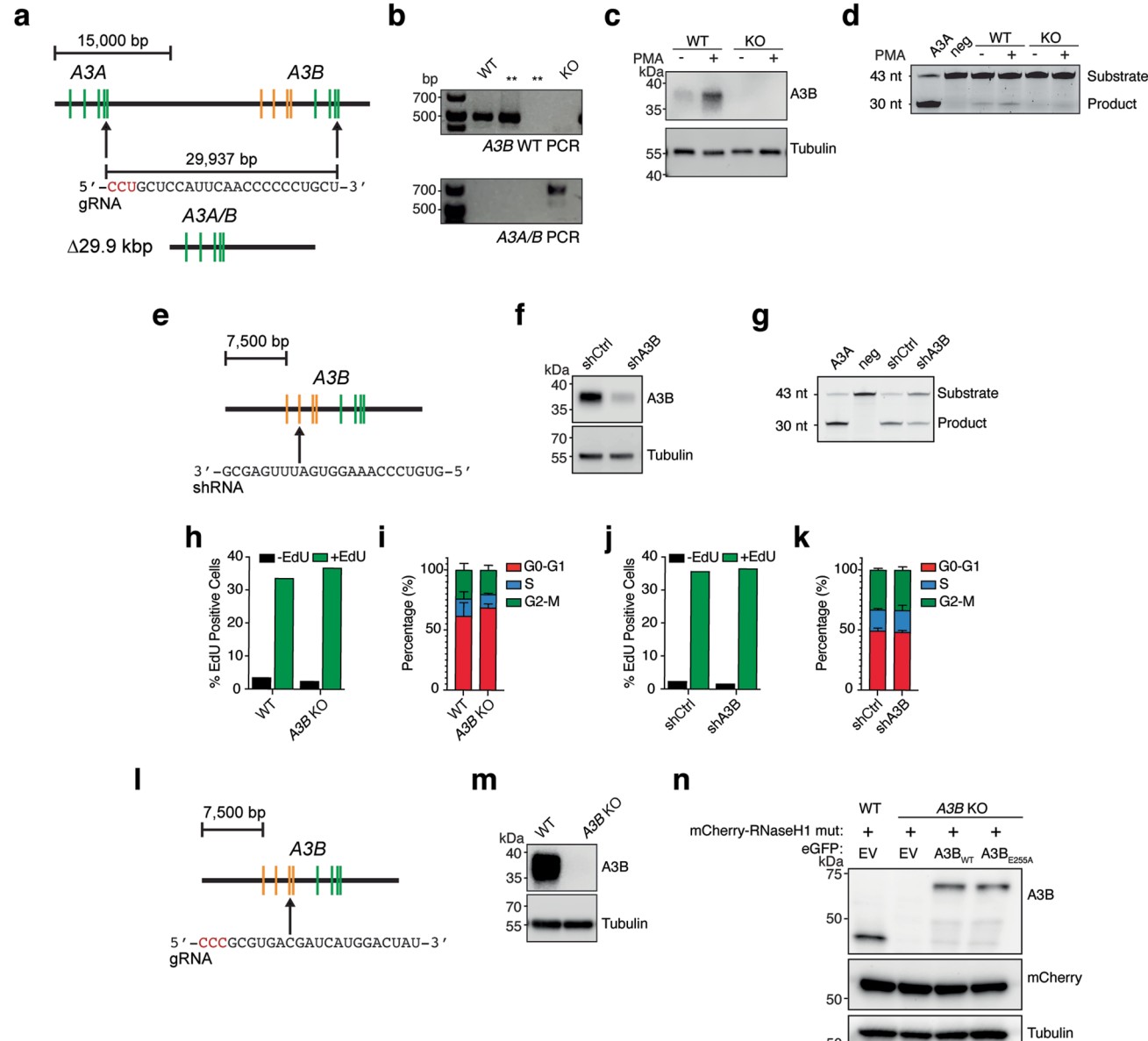

**Extended Data Fig. 2 | Construction and validation of cell lines. a**, Schematic of the *A3B* knock-out strategy resulting in an *A3A/B* fusion. CRISPR cleavage sites are indicated by arrows and the homologous gRNA-targeted region is shown below with PAM (red). Exons are indicated by colored boxes. **b**, Diagnostic PCR products distinguishing WT *A3B* and 29.9 kbp *A3B* deletion allele (**, clones not pertaining to this manuscript; sequence verified). **c**, Immunoblot of MCF10A WT and *A3B* KO derivative treated with DMSO or PMA (25 ng/ml, 24 hrs) and probed with the indicated antibodies (n = 3 independent experiments). **d**, DNA deaminase activity assay using extracts from MCF10A WT and *A3B* KO derivative treated with DMSO or PMA (25 ng/ml, 24 hrs; purified A3A positive control; reaction buffer negative control; n = 3 independent experiments). **e**, *A3B* gene schematic with an arrow indicating the exon 2 mRNA region targeted by an *A3B*-specific shRNA in depletion experiments (target sequence shown below). **f**, Immunoblot of U2OS shCtrl and shA3B cell lines probed with the indicated

antibodies; (n = 3 independent experiments). **g**, DNA deaminase activity assay of extracts from U2OS shCtrl and shA3B cell lines (purified A3A was used as a positive control and reaction buffer as a negative control; n = 3 independent experiments). **h**, EdU staining of MCF10A WT and *A3B* KO cell lines (n = 1 with a minimum of 10,000 cells per condition). **i**, PI staining of MCF10A WT and *A3B* KO cell lines (n = 3 experiments with 10,000 cells per condition; mean ± SD). **j**, EdU staining of U2OS shCtrl and shA3B cell lines (n = 1 with 10,000 cells per condition). **k**, PI staining of U2OS shCtrl and shA3B cell lines (n = 3 experiments with 10,000 cells per condition; mean ± SD). **l**, *A3B* gene schematic with an arrow indicating the exon 3 gRNA targeting region (target sequence shown below). **m**, Immunoblot of whole cell extracts from U2OS WT and *A3B* KO cell lines probed with the indicated antibodies (n = 3 independent experiments). **n**, Immunoblot of whole cell extracts from U2OS WT and *A3B* KO cell lines transfected as shown and probed with the indicated antibodies (n = 2 independent experiments).

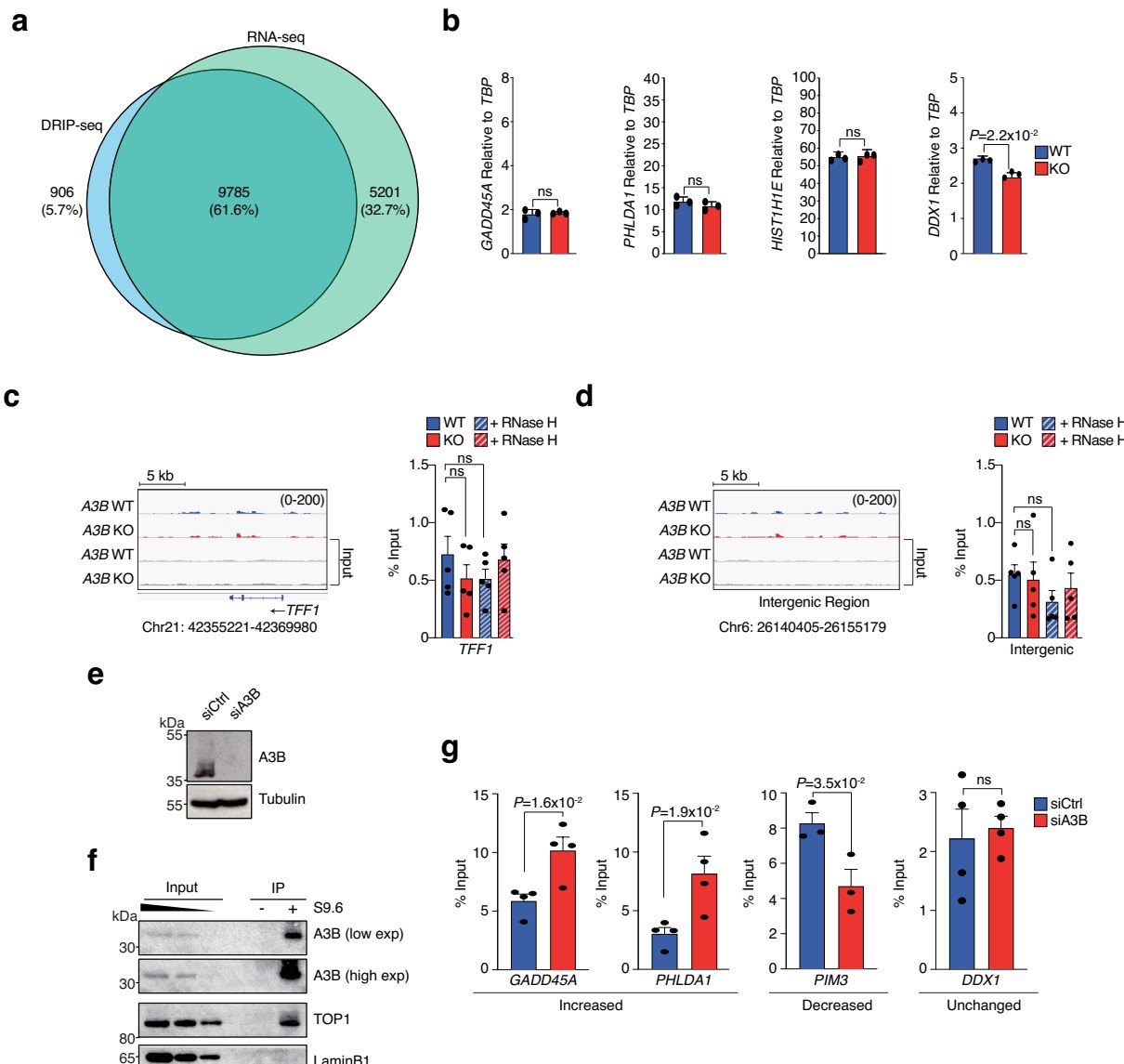

**Extended Data Fig. 3 | Supporting data for DRIP-seq experiments. a**, Venn diagram depicting the overlap between DRIP-seq positive genes and expressed genes (RNA-seq) in MCF10A. **b**, RT-qPCR analysis of mRNA levels in MCF10A (WT) and *A3B* knockout MCF10A (KO) cells. Values for the indicated genes are expressed relative to the housekeeping gene, *TBP* (n = 3 independent experiments; mean ± SEM; *P*-value by two-tailed unpaired *t*-test). **c-d**, DRIP-seq profiles for a non-expressed gene, *TFF1*, and an intergenic region in MCF10A (WT and *A3B* KO) cells. DRIP-qPCR ± exogenous RNase H (RNH; striped bars) is shown in histograms to the right (n = 5 biologically independent experiments;

means ± SEM expressed as percentage of input; ns by two-tailed unpaired *t*-test). **e**, Immunoblot of HeLa cells transfected with either an siRNA against Luciferase (siCtrl) or A3B (siA3B) and probed with the indicated antibodies (n = 2 independent experiments). **f**, Immunoblots of indicated proteins in S9.6 IP reactions from HeLa cells (n = 2 independent experiments). Lamin B1 is a negative control. **g**, DRIP-qPCR of genes from the subgroups listed in Fig. 5c–e in HeLa cells (n = 4 for each gene, except n = 3 for *PIM3*, in biologically independent experiments; means ± SEM expressed as percentage of input; *P*-value by two-tailed unpaired *t*-test).

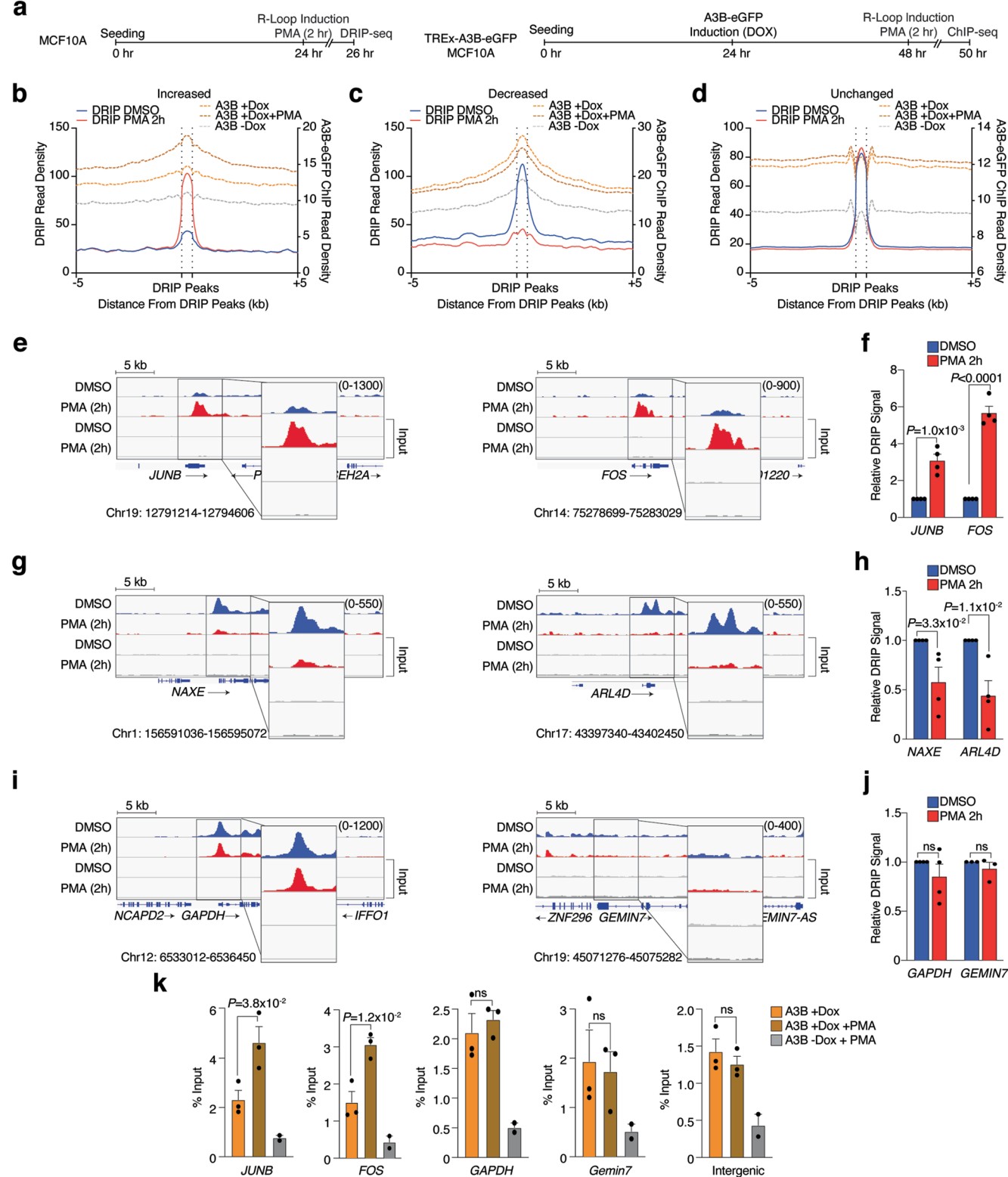

**Extended Data Fig. 4 | See next page for caption.**

**Extended Data Fig. 4 | Kinetics of R-loop induction and resolution.**
**a**, Schematic of the DRIP-seq (left) and A3B-eGFP ChIP-seq (right) workflows
used for panels b-j. **b**–**d**, Meta-analysis of read density (FPKM) for DRIP-seq
results from DMSO (blue) or PMA-treated (25 ng/ml) MCF10A (red) partitioned
into 3 groups (increased, decreased, and unchanged) as described in the text.
A3B-eGFP ChIP-seq data (Dox-, Dox+, and Dox+PMA in gray, orange, and brown
dashed lines, respectively) superimposed on DRIP peaks ± 5 kb (right y-axis).
**e, f**, DRIP-seq profiles for *JUNB* and *FOS* from the increased data set in panel b.
*JUNB* DRIP-seq profile is the same as Fig. 6d PMA 2 h. DRIP-qPCR is shown in the
histogram to the right (n = 4 independent experiments; means ± SEM normalized
to DMSO; *P*-value by two-tailed unpaired *t*-test). **g, h**, DRIP-seq profiles for *NAXE*

and *ARL4D* from the decreased data set in panel c. DRIP-qPCR is shown in the
histogram to the right (n = 4 independent experiments; means ± SEM normalized
to DMSO; *P*-value by two-tailed unpaired *t*-test). **i, j**, DRIP-seq profiles for *GAPDH*
and *GEMIN7* from the unchanged data set in panel d. DRIP-qPCR is shown in
the histogram to the right (n = 4 for *GAPDH* and n = 3 for *GEMIN7* independent
experiments; means ± SEM normalized to DMSO; ns by two-tailed unpaired
*t*-test). **k**, ChIP-qPCR is shown in the histogram for PMA-responsive (*JUNB*, *FOS*)
and PMA non-responsive (*GAPDH*, *GEMIN7*) genes as well as an intergenic control
(n = 3 independent experiments for all conditions except n = 2 for -DOX + PMA;
means ± SEM expressed as percentage of input; *P*-value by two-tailed unpaired
*t*-test).

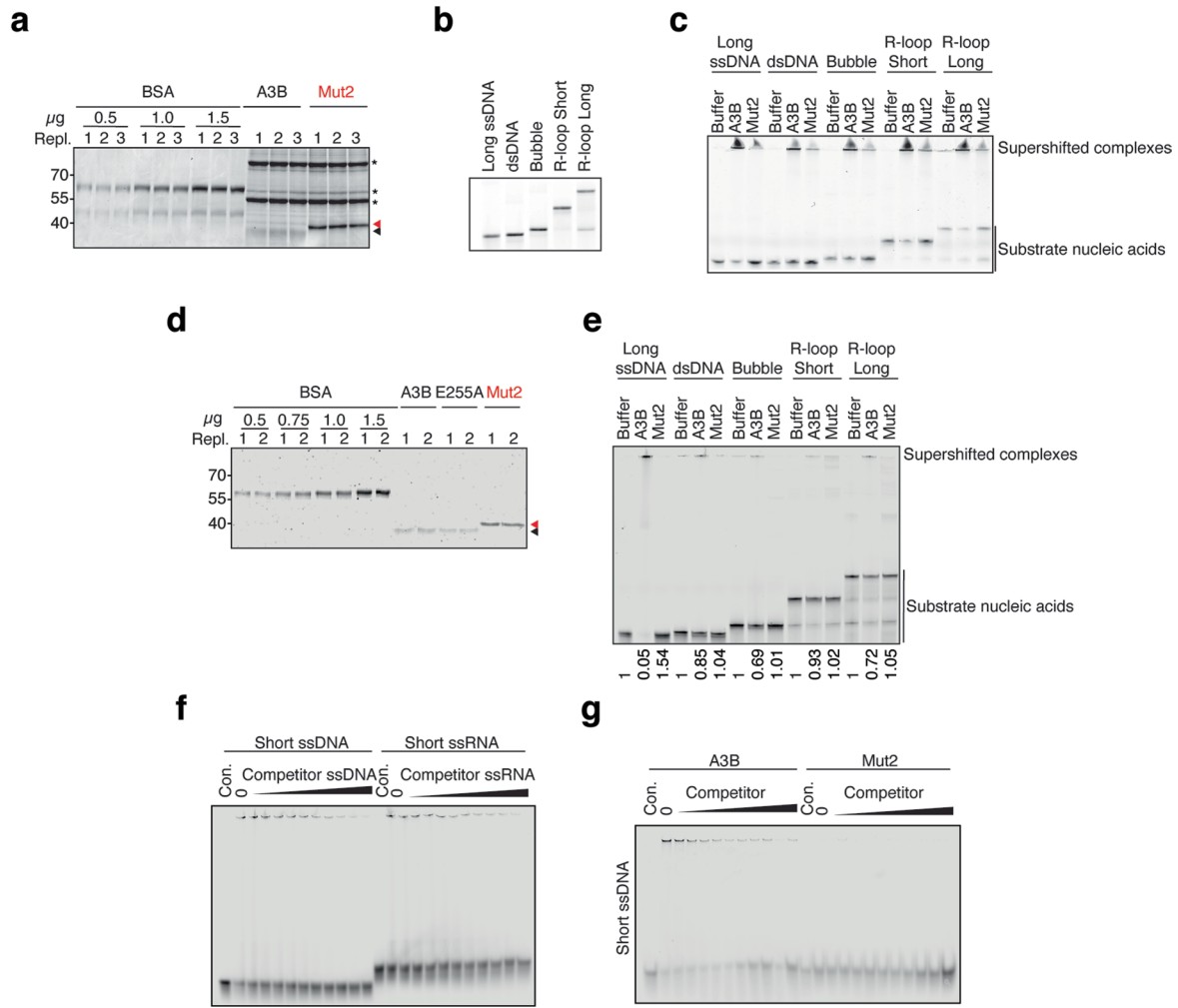

**Extended Data Fig. 5 | Purifications of A3B and Mut2 including additional EMSA results. a**, Coomassie-stained gel of Ni-NTA affinity purified A3B and Mut2 proteins from 293 T cells (3 replicate loadings for quantification). Black and red arrow heads indicate WT A3B-mycHis and Mut2-mycHis, respectively. Co-purifying proteins (*) are similar for WT and Mut2 (n = 3 independent experiments). **b**, Native TBE-PAGE of the 5' fluorescently labeled substrates depicted in Fig. 7a (size standards not applicable due to native conditions; n = 3 independent experiments). **c**, Native EMSA comparing WT and Mut2 binding to the indicated nucleic acid substrates. Stronger WT binding is indicated by more supershifted substrates, more intense staining of complexes retained in the wells, and larger diffusion 'tails' within each well (an unavoidable issue if some complexes fail to enter the gel; size standards not applicable due to native conditions; n = 3 independent experiments). **d**, Coomassie-stained gel of purified A3B-, A3B-E72A-, and Mut2-mycHis proteins from Expi293 cells (2 replicate loadings for quantification; n = 1 independent experiments). Black

and red arrow heads indicate purified A3B, A3B-E72A, and Mut2 proteins (>85% pure). **e**, Native EMSA comparing WT A3B and Mut2 binding to the indicated nucleic acid substrates. Stronger WT binding is indicated by a larger proportion of supershifted substrates, more intense staining of complexes retained in the wells, and a diminution of unbound substrate at the expected mobility (this experiment used proteins shown in panel **d**). The numbers below represent quantification of the substrate band relative to that of the buffer control; n = 3 independent experiments. **f**, Native EMSAs of WT binding to short 15mer ssDNA or RNA in the presence of increasing concentrations of otherwise identical unlabeled competitor (this experiment used proteins shown in panel **d**; n = 3 independent experiments). **g**, EMSAs comparing WT and Mut2 binding to short 15mer ssDNA and RNA in the presence of increasing concentrations of otherwise identical unlabeled competitor ssDNA or RNA (this experiment used proteins shown in panel **d**; n = 3 independent experiments).

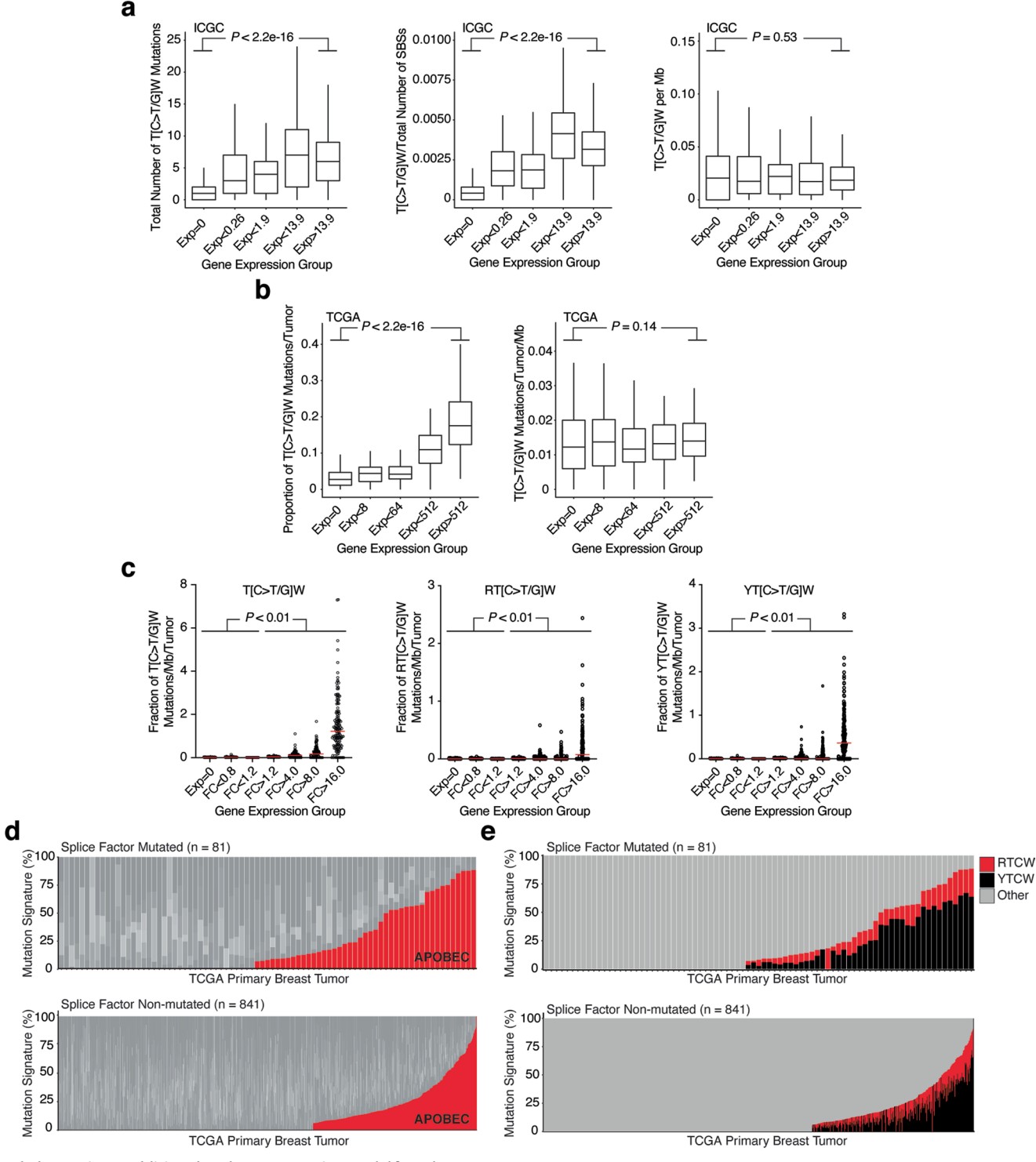

**Extended Data Fig. 6 | Additional analyses supporting model for R-loop mutation. a, b,** Positive correlations between gene expression levels and APOBEC signature T(C > T/G)W mutation number and frequency in ICGC and TCGA breast cancer data sets flatten upon normalization for gene size (*P*-value by Pearson's correlation). ICGC expression groups are based on gene expression levels in normal breast tissue from the Genotype-Tissue Expression (GTEx) project. TCGA expression groups are 0 and quartiles for anything >0 and based on average expression levels for each gene using TCGA RNA-seq values from primary breast tumors. **c,** Dot plot representations of the relationship between APOBEC signature mutations (per mb per tumor) and the indicated TCGA breast cancer gene expression groups (FC, fold-change relative to mean normal expression value in the TCGA normal breast tissue RNA-seq data). Left is identical to main Fig. 8b and the center and right panels show breakdowns into RTCW

and YTCW subsets, respectively. Pairwise comparisons are significant for all combinations of the lowest 3 and the highest 4 FC expression groups (*P*-value by Welsh's *t*-test). **d,** Data here are identical those in Fig. 8c to facilitate comparison with tetranucleotide breakdowns in panel **e. e,** An alternative representation of the data in panel **d,** with RTCW mutation proportions shown in red, YTCW mutation proportions in black, and other signatures in gray. This analysis revealed a significant trend with only 1/43 (2.3%) of the APOBEC3 signature-enriched splice factor mutant breast tumors lacking mutations in A3B-associated RTCW motifs in comparison to 52/326 (15.9%) of the APOBEC3 signature-enriched non-splice factor mutant tumors (*that is*, the A3B-associated tetranucleotide preference is enriched in the splice factor mutant group and/or depleted from the non-splice factor mutant group; *P* = 0.028 by Fisher's exact test).

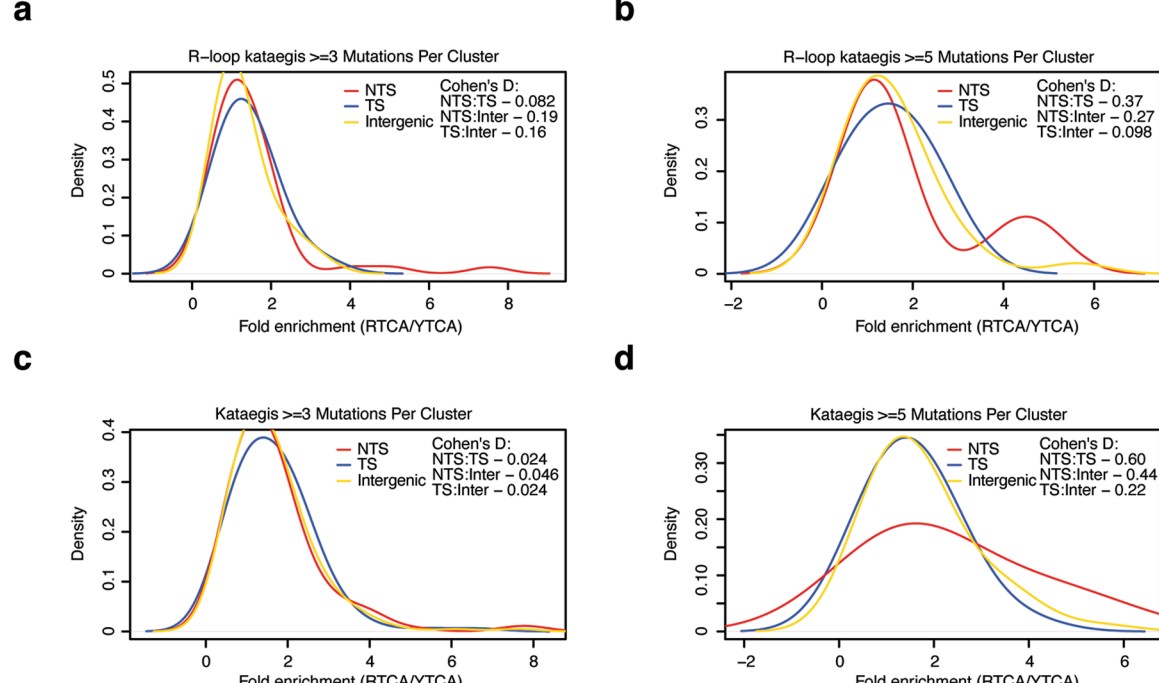

**Extended Data Fig. 7 | Enrichments of R-loop *kataegis* across RTCA versus YTCA contexts. a, b,** Distributions of the fold-enrichment of RTCA versus YTCA sequence contexts within non-transcribed, transcribed, and intergenic regions (red, blue, and yellow lines, respectively). The Cohen's D effect size was calculated for all pairwise region comparisons within R-loop *kataegic* events that include smaller clustered events (panel **a**) versus only larger *kataegic* events with ≥5 mutations per cluster (panel **b**). **c, d,** The same comparisons were performed for all *kataegic* events genome-wide that include smaller clustered events (panel **c**) and only larger *kataegic* events ≥5 mutations per cluster (panel **d**).

# Reporting Summary

Nature Research wishes to improve the reproducibility of the work that we publish. This form provides structure for consistency and transparency in reporting. For further information on Nature Research policies, see our Editorial Policies and the Editorial Policy Checklist.

## Statistics

For all statistical analyses, confirm that the following items are present in the figure legend, table legend, main text, or Methods section.

| n/a | Confirmed | |
|---|---|---|
| ☐ | ☒ | The exact sample size (*n*) for each experimental group/condition, given as a discrete number and unit of measurement |
| ☐ | ☒ | A statement on whether measurements were taken from distinct samples or whether the same sample was measured repeatedly |
| ☐ | ☒ | The statistical test(s) used AND whether they are one- or two-sided<br>*Only common tests should be described solely by name; describe more complex techniques in the Methods section.* |
| ☐ | ☒ | A description of all covariates tested |
| ☐ | ☒ | A description of any assumptions or corrections, such as tests of normality and adjustment for multiple comparisons |
| ☐ | ☒ | A full description of the statistical parameters including central tendency (e.g. means) or other basic estimates (e.g. regression coefficient) AND variation (e.g. standard deviation) or associated estimates of uncertainty (e.g. confidence intervals) |
| ☐ | ☒ | For null hypothesis testing, the test statistic (e.g. *F*, *t*, *r*) with confidence intervals, effect sizes, degrees of freedom and *P* value noted<br>*Give P values as exact values whenever suitable.* |
| ☒ | ☐ | For Bayesian analysis, information on the choice of priors and Markov chain Monte Carlo settings |
| ☒ | ☐ | For hierarchical and complex designs, identification of the appropriate level for tests and full reporting of outcomes |
| ☒ | ☐ | Estimates of effect sizes (e.g. Cohen's *d*, Pearson's *r*), indicating how they were calculated |

*Our web collection on statistics for biologists contains articles on many of the points above.*

## Software and code

Policy information about availability of computer code

| Data collection | ImageJ (v1.48) NIH https://imagej.nih.gov/ij/<br>FV-10-ASW3.1 Olympus https://www.olympus-ims.com/en/<br>GraphPad Prism 6 N/A http://www.graphpad.com<br>Image Studio Li-COR Biosciences https://www.licor.com/bio/image-studio/<br>Typhoon FLA-7000 Image Reader GE Life Sciences<br>Li-COR Odyssey Fc Li-COR<br>NovaSeq 6000<br>Fluoview 3000 Confocal  Olympus<br>Nikon AR1<br>BDLSRFortessa<br>Rotor Gene Q Thermocycler (QIAGEN) software v2.3.1 |
|---|---|

| Data analysis | Image J, FV-10-ASW3.1 Olympus and Fluoview 3000 Confocal Olympus and Nikon AR1were used for collection and quantification of immunofluorescence. GraphPad Prism v6, 8.0 and 8.3.1 was used for data plotting and statistical calculations. Image Studio Li-COR Biosciences was used for western blotting and dot blot quantifications. Cutadapt 1.13 (N/A https://cutadapt.readthedocs.io/en/stable/index.html), STAR 2.6.1d (N/A https://github.com/alexdobin/STAR), SAMtools 1.3.1 (N/A http://www.htslib.org/), Picard tools (N/A https://broadinstitute.github.io/picard/), Deeptools 2.5.0.1 (N/A https://deeptools.readthedocs.io/en/latest/index.html), MACS2 2.1.1.20160309 (N/A https://github.com/taoliu/MACS) and Bedtools version 2.29.2 were used for ChIP-seq and DRIP-seq data processing. Salmon 0.13.1 (N/A https://salmon.readthedocs.io/en/latest/index.html) was used for RNA-seq differential expression analysis. Typhoon FLA-7000 Image Reader GE Life Sciences and Li-COR Odyssey Fc Li-COR were used for western blot and dot blot imaging. Real-time quantitative PCRs (qPCRs) were analysed with the software v2.3.1 of Rotor Gene Q Thermocycler (QIAGEN). Li-COR Odyssey and Odyssey software were used for EMSA and A3B activity assay imaging and quantification. FlowJo was used for analysis of flow cytometry data (PI and EdU staining). |
|---|---|

For manuscripts utilizing custom algorithms or software that are central to the research but not yet described in published literature, software must be made available to editors and reviewers. We strongly encourage code deposition in a community repository (e.g. GitHub). See the Nature Research guidelines for submitting code & software for further information.

## Data

Policy information about availability of data

All manuscripts must include a data availability statement. This statement should provide the following information, where applicable:
- Accession codes, unique identifiers, or web links for publicly available datasets
- A list of figures that have associated raw data
- A description of any restrictions on data availability

The accession number for the ChIP-seq and DRIP-seq reported in this paper (Fig 5-6, S3 and S4) is GEO: GSE148581. The GEO accession number for the ChIP-seq, and DRIP-seq reported in this paper is: GSE148581. Protein-coding and non-coding genes annotation was taken from Gencode V31 and enhancers from FANTOM5, both based on hg38 version of the human genome. ChIP-seq and DRIP-seq peak calling files are available in the GEO submission GSE148581.

# Field-specific reporting

Please select the one below that is the best fit for your research. If you are not sure, read the appropriate sections before making your selection.

☒ Life sciences ☐ Behavioural & social sciences ☐ Ecological, evolutionary & environmental sciences

For a reference copy of the document with all sections, see nature.com/documents/nr-reporting-summary-flat.pdf

# Life sciences study design

All studies must disclose on these points even when the disclosure is negative.

| Sample size | No calculation or statistics were used to determine sample size. Sample size was determined according to established practice and applicable standard at affordable costs. Samples sizes are indicated separately for different experiments. Statistical analysis was used to determine statistical significance of obtained results (as indicated in figure legends). P-values are reported in figures and/or figure legends. |
|---|---|
| Data exclusions | No samples were excluded. |
| Replication | The exact sample size (n) for each experimental group/condition are provided in the figure legends. |
| Randomization | Randomization was not feasible for this type of molecular biology, biochemistry and genomics experiments. The culture cell plates were randomly assigned to each group for respective treatment and samples in the same experiments were treated in the same manner. |
| Blinding | Blinding was not possible for our experiments as data collection and analysis were performed by the same person. |

# Reporting for specific materials, systems and methods

We require information from authors about some types of materials, experimental systems and methods used in many studies. Here, indicate whether each material, system or method listed is relevant to your study. If you are not sure if a list item applies to your research, read the appropriate section before selecting a response.

## Materials & experimental systems

| n/a | Involved in the study |
|-----|----------------------|
| ☐ | ☒ Antibodies |
| ☐ | ☒ Eukaryotic cell lines |
| ☒ | ☐ Palaeontology and archaeology |
| ☒ | ☐ Animals and other organisms |
| ☒ | ☐ Human research participants |
| ☒ | ☐ Clinical data |
| ☒ | ☐ Dual use research of concern |

## Methods

| n/a | Involved in the study |
|-----|----------------------|
| ☐ | ☒ ChIP-seq |
| ☐ | ☐ Flow cytometry |
| ☒ | ☐ MRI-based neuroimaging |

# Antibodies

| Antibodies used | Primary antibodies used in these experiments were:<br>Tubulin (Abcam, Cat# ab4074, RRID:AB_2288001, Clone EPR13478(B)). WB 1:5000<br>Tubulin (Abcam, Cat# ab6046, RRID:AB_2210370, lot#GR77827-1). WB 1:5000<br>Tubulin (Sigma-Aldrich Cat# T5168, RRID:AB_477579, lot#039M4769V). WB 1:10000<br>Flag (Sigma-Aldrich Cat# F1804, RRID:AB_262044, lot#SLCF9337). WB 1:10000<br>A3B (5210-87-13, inhouse13). WB 1:2000<br>Topoisomerase I (Abcam, Cat# ab109374, RRID:AB_10861978, clone EPR5375, lot #GR49853-20) WB 1:1000 or 1:500.<br>Lamin B1 (Abcam Cat# ab16048, RRID:AB_443298). WB 1:2000<br>anti-IgG2a (Sigma-Aldrich Cat# M5409, RRID:AB_1163691)<br>anti-HA (Cell Signaling Technology Cat# 3724, RRID:AB_1549585). WB: 1:5000<br>HNRNPUL1 (gift from Prof. Stuart Wilson, University of Sheffield, UK). WB 1:1000.<br>GFP (Abcam Cat# ab290, RRID:AB_303395, lot# GR3251545 and GR3270983). WB 1:3000. Co-IP 2.5 µg. ChIP 2.5 µg.<br>mCherry (Abcam Cat# ab167453, RRID:AB_2571870, lot# GR3209879-3). WB 1:1000.<br>IgG Isotype Control (Thermo Fisher Scientific Cat# 02-6102, RRID:AB_2532938, lot#RI238244). Co-IP 2.5 µg.<br>RNA/DNA hybrids clone S9.6 (Gromak Lab, University of Oxford Cat# Gromak_1, RRID:AB_2810829). Slot Blot 1:1000. RNA/DNA hybrid and protein co-immunoprecipitation 100 µl. DRIP 30 µl.<br>RNA/DNA hybrids clone S9.6 (Kerafast Cat# ENH001, RRID:AB_2687463, lot# 032119_2 and 032119_4). IF 1:200. Dot blot 1:200000<br>dsDNA (Abcam Cat# ab27156, RRID:AB_470907) Dot blot 1:300000<br>gamma-H2AX (Novus Cat# NB100-384, RRID:AB_10002815, lot# A22). IF 1:1000<br><br>Secondary antibodies used were:<br>Rabbit IRdye 800CW (LI-COR Biosciences Cat# 827-08365, RRID:AB_10796098) WB 1:10000<br>Mouse IRdye 680LT (LI-COR Biosciences Cat# 925-68020, RRID:AB_2687826) WB 1:10000<br>Rabbit HRP (Cell Signaling Technology Cat# 7074, RRID:AB_2099233) WB 1:10000<br>Rabbit IgG (whole molecule)-Peroxidase antibody produced in goat (Sigma-Aldrich Cat# A0545, RRID:AB_257896). WB 1:8000.<br>Mouse HRP (Cell Signaling Technology Cat# 7076, RRID:AB_330924) WB 1:10000<br>Mouse IgG (whole molecule)-Peroxidase antibody produced in goat (Sigma-Aldrich Cat# A8924, RRID:AB_258426). Slot Blot 1:3333.<br>Alexa Fluor 488 goat anti-mouse IgG (Molecular Probes Cat# A-11029, RRID:AB_2534088). IF 1:1000<br>Alexa Fluor 594 goat anti-mouse IgG (Molecular Probes Cat# A-11032, RRID:AB_2534091). IF 1:1000<br>Alexa Fluor 488 goat anti-rabbit IgG (Thermo Fisher Scientific Cat# A-11034, RRID:AB_2576217) IF 1:1000<br>Alexa Fluor 594 goat anti-rabbit IgG (Thermo Fisher Scientific Cat# A-11037, RRID:AB_2534095) IF 1:1000<br>Alexa Fluor 647 goat anti-mouse IgG (Thermo Fisher Scientific Cat# A-21236, RRID:AB_2535805). IF 1:1000 |
|-----------------|---|
| Validation | Rabbit polyclonal alpha Tubulin (ab4074) was validated by Abcam using WB (1 µg/ml dilution) in several human (including HeLa, NIH 3T3 and PC12) and mouse whole cell extracts. The specific WB band can be prevented by incubation with human alpha-tubulin peptide. https://www.abcam.com/alpha-tubulin-antibody-loading-control-ab4074.html.<br><br>Rabbit IgG Isotype control (02-6102) was used in literature as IP control (e.g. Ge Zhy, Mol Cell, 2019 PMID: 30846317; Cristini et al, Nat Commun, 2022 PMID:35618715). We further showed the absence of protein binding tested by WB following co-IP with this antibody (Fig 1d).<br><br>Rabbit polyclonal Lamin B1 antibody was validated by Abcam using WB (0.1-1 µg/ml dilution) in HeLa and A431 whole cell extracts. No band was detected upon LMNB1 (lamin B1) knock-out in HAP1 cells. https://www.abcam.com/lamin-b1-antibody-nuclear-envelope-marker-ab16048.html.<br><br>Rabbit monoclonal (EPR5675) anti-Topoisomerase I (ab109374) antibody was validated by Abcam using WB (1/10000 dilution) in several human whole cell lysates (e.g. MCF7, Jurkat, HepG2, K562). https://www.abcam.com/products/primary-antibodies/topoisomerase-i-antibody-epr5375-ab109374.html. Authors have previously validated the specificity of this ab for WB applications by showing a decreased band at 90 KDa in presence of siRNA targeting TOP1 in human U2OS and WI38 lysates (Mouly et al, Cell Death Dis, 2018 PMID: 30209297) and for usage in RNA/DNA and protein co-IP (Cristini et al, Cell Rep, 2018 PMID: 29742442; Abakir et al, Nat Genet, 2020 PMID: 31844323).<br><br>Mouse monoclonal RNA/DNA hybrid clone S9.6 antibody used in this study was previously validated for DRIP application by showing that RNase H digestion significantly removed S9.6 signal, indicating its specificity for RNA/DNA hybrids (Groh et al, PLoS Genet, 2014 PMID: 24787137; Cristini et al, Cell Rep, 2018 PMID:29742442; Cristini et al, Cell Rep, 2019 PMID: 31533039; Cristini et al, Nat Commun, 2022 PMID:35618715). We further validated the specificity of this antibody in our DRIP-qPCR experiments by using RNase H digestion in vitro control (Fig. 5g,i,k and 6d,e). S9.6 antibody was validated in Immunofluorescence experiments by expression of RNaseH, which processes R-loops and removes the specific RNA/DNA hybrid signal (see Kim et., 2019, Genes Dev, PMID: 31753913; |

Bayona-Feliu et al., 2021, Nature Genetics, PMID: 33986538; Abakir et al., 2020, Nature Genetics 2020, PMID:31844323; Jurga M et al, 2021, Nat Commun, PMID: 34526504; Ramachandran et al, 2021, Nat Commun, PMID: 34140498; Pérez-Calero C et al, 2020, Genes Dev, PMID: 32439635).

Rabbit polyclonal anti-gamma H2AX (NB100-384) was validated by Abcam using a genetic strategy using Western blotting with extracts from H2AX WT cells and H2AX KO cells (from human HEK293, human melanoma (G361), mouse wildtype embryonic fibroblasts (+/+) or mouse H2AX knockout embryonic fibroblasts (-/-), either untreated or treated with the Neocarzinostatin, a DNA damaging agent and probed with antibody at 0.1 ug/ml.

Rabbit polyclonal anti-GFP (ab290) antibody was validated by Abcam using WB to detect the GFP fraction from cell extracts expressing recombinant GFP fusion proteins, notably in COS7 and LNCaP whole cell lysate transfected with GFP-Eml4. It is routinely used in immunoprecipitation and it has been validated in IP in HEK293 nuclear lysate expressing GFP versus lysates from cells with no GFP. https://www.abcam.com/products/primary-anti4bodies/gfp-antibody-ab290.html. It is also used for ChIP applications (https://www.abcam.com/content/anti-gfp) in human cells (Mei Tan-Wong et al, Mol Cell, 2019 PMID: 31679819).

Rabbit polyclonal anti-mCherry (ab167453) antibody was validated by Abcam using WB to detect the mCherry fraction from cell extracts expressing recombinant mCherry fusion proteins in HEK293 whole cell lysate transfected with pFin-EF1-mCherry vector. IF validation was performed similarly by Immunofluorescent analysis of HEK293 cells transfected with pFin-EF1-mCherry vector labeling mCherry with ab167453 at 1/500 dilution.

Rabbit polyclonal anti-hNRNPUL1 used in this study was a gift from Prof. Stuart Wilson, University of Sheffield, UK. The specificity of this antibody has been validated in the Gromak laboratory for WB applications by showing the decreased band at around 100 KDa in presence of siRNA targeting hNRNPUL1 in human cell lysates. This antibody has further been validated by IP.

Validation for anti-A3B (5210-87-13, in house) is published in Brown, W.L. et al. A rabbit monoclonal antibody against the antiviral and cancer genomic DNA mutating enzyme APOBEC3B. Antibodies (Basel) 8(2019).

# Eukaryotic cell lines

Policy information about cell lines

**Cell line source(s)**

U2OS cells were obtained from ATCC (ATCC HTB-96).
U2OS shCtrl and shA3B cell lines were made from U2OS cells (ATCC HTB-96) using previously described shCtrl and shA3B lentiviral constructs, viral production and transduction methods and puromycin selection 1 μg/mL1. U2OS pcDNA3.1-A3-3xHA stable lines were made via linear (NruI digested) transfection and selection using 800 μg/mL G418.
HEK 293T cells were obtained from ATCC (#CRL-3216).
MCF10A cells were obtained from ATCC (ATCC CRL-10317)
MCF10A-TREx-A3B-eGFP were maintained in the same MCF10A media described above with the addition of 100 μg/mL Normocin.
S9.6 Hybridoma cells were obtained from ATCC (ATCC HB-8730).
MCF10A A3B KO cell line was engineered by transduction of MCF10A cells (ATCC CRL-10317) with pLentiCRISPR, expressing the gRNA sequence GCTCCATTCAACCCCCCTGCT targeting both the A3A and A3B genes. Cells were selected with puromycin and seeded for single cell cloning. Deletion mutant lines were identified by PCR using primers amplifying unique sequences within the A3B gene and/or the A3A/B junction (primers in ref.2) and confirmed by qPCR and immunoblots.
HeLa cells were obtained from Nicholas Proudfoot (University of Oxford, UK) and are originally from ATCC.
U2OS A3B KO cell line was engineered from U2OS cells (ATCC HTB-96) by transduction with pLentiCRISPR, expressing the gRNA sequence GCGTGACGATCATGGACTAT targeting exon 3 of A3B. Cells were selected with puromycin and seeded for single cell cloning. Biallelic A3B knockout was confirmed by PCR using primers spanning the gRNA target region and subsequent sequencing in addition immunoblotting.

**Authentication**

STR profiling is used to authenticate the cell lines that were purchased from ATCC.
U2OS shCtrl and shA3B cell lines were validated using RT-qPCR and Western Blot for knockdown efficiency.
MCF10A-TREx-A3B-eGFP was validated using IF, Western blotting and RT-qPCR for inducible expression of A3B-eGFP.
MCF10A A3B KO was validated by PCR using primers amplifying unique sequences within the A3B gene and/or the A3A/B junction (primers in ref.2) and confirmed by qPCR and immunoblots.
U2OS A3B KO cell line was confirmed by PCR using primers spanning the gRNA target region and subsequent sequencing in addition immunoblotting.

**Mycoplasma contamination**

All cell lines were confirmed mycoplasma negative. MCF10A-TREx-A3B-eGFP had been cured of mycoplasma prior to initial studies.

**Commonly misidentified lines**
(See ICLAC register)

No commonly misidentified lines were used.

# ChIP-seq

## Data deposition

☒ Confirm that both raw and final processed data have been deposited in a public database such as GEO.

☒ Confirm that you have deposited or provided access to graph files (e.g. BED files) for the called peaks.

| Data access links | The accession number for the ChIP-seq and DRIP-seq reported in this paper is GEO: GSE148581. |
| --- | --- |
| *May remain private before publication.* | |
| Files in database submission | ChIP-seq: Input and GFP IP -DOX, Input and GFP IP +DOX, Input and GFP IP +DOX +PMA in MCF10A-TREx-A3B-eGFP. DRIP-seq: Input and IP DMSO, PMA 2h, PMA 6h in MCF10A A3B WT or KO. |
| Genome browser session (e.g. UCSC) | https://genome.ucsc.edu/s/TMichael2/082021Submission_APOBEC_ChIPseq_DRIPseq_hg38 |

## Methodology

| Replicates | 1. DRIP-seq and ChIP-seq results were validated by DRIP- and ChIP-qPCR. |
| --- | --- |
| Sequencing depth | For ChIP-seq: between 67 and 89 million unique and properly mapped paired-end reads per sample. For DRIP-seq: between 50 and 149 million unique and properly mapped paired-end reads per sample. Length of the reads: 150 bp. |
| Antibodies | anti-GFP (Abcam, ab290, lot# GR3251545 and GR3270983) for ChIP-seq and RNA/DNA hybrids (S9.6, Natalia Gromak Lab, University of Oxford Cat# Gromak_1, RRID:AB_2810829) for DRIP-seq. |
| Peak calling parameters | Adapters were trimmed with Cutadapt version 1.13 in paired-end mode with the following parameters: -q 15, 10 –minimum-length 10 -A AGATCGGAAGAGCGTCGTGTAGGGAAAGAGTGT -a AGATCGGAAGAGCACACGTCTGAACTCCAGTCA. Obtained sequences were mapped to the human hg38 reference genome with STAR version 2.6.1d and the parameters --runThreadN 16 --readFilesCommand gunzip -c –k --alignIntronMax 1 --limitBAMsortRAM 20000000000 --outSAMtype BAM SortedByCoordinate. Properly paired and mapped reads (-f 3) were retained with SAMtools version 1.3.1. PCR duplicates were removed with Picard MarkDuplicates tool. Reads mapping to the DAC Exclusion List Regions (accession: ENCSR636HFF) were removed with Bedtools version 2.29.2. FPKM-normalized bigwig files were created with deepTools version 2.5.0.1 bamCoverage tool with the parameters –bs 10 –p max -e --normalizeUsing RPKM.

ChIP-seq and DRIPseq peaks were called with MACS2 version 2.2.6. (Zhang et al., 2008) and the parameters: callpeak -f BAMPE -g 2.9e9 -q 0.01 --bdg --call-summits --nomodel --extsize 200. Each IP and its respective input were used as treatment and control, respectively. DRIP-seq differential peak calling was performed with MACS2 bdgdiff tool. The lists of called peaks are presented as the summit -/+ 250 bp (based on the --call-summits option of MACS2). |
| Data quality | Number of peaks with a fold enrichment > 5 and a q-value < 0.01 for ChIP-seq: 1,301 peaks for GFP IP DOX- DMSO, 610 peaks for GFP IP DOX+ DMSO, and 771 peaks for GFP IP DOX+ PMA 2h.

Number of peaks with a fold enrichment > 5 and a q-value < 0.01 for DRIP-seq: 30,920 peaks for WT DMSO, 48,537 peaks for WT +PMA 2h, 89,578 peaks for WT+PMA 6h, 33,794 peaks for KO DMSO, 60,404 peaks for KO+PMA 2h, 73,070 peaks for KO+PMA 6h. |
| Software | Adapters were trimmed with Cutadapt version 1.13 in paired-end mode. Obtained sequences were mapped to the human hg38 reference genome with STAR version 2.6.1d a. Properly paired and mapped reads (-f 3) were retained with SAMtools version 1.3.1. PCR duplicates were removed with Picard MarkDuplicates tool. eads mapping to the DAC Exclusion List Regions (accession: ENCSR636HFF) were removed with Bedtools version 2.29.2. FPKM-normalized bigwig files were created with deepTools version 2.5.0.1 b. |

# Flow Cytometry

## Plots

Confirm that:

☒ The axis labels state the marker and fluorochrome used (e.g. CD4-FITC).

☒ The axis scales are clearly visible. Include numbers along axes only for bottom left plot of group (a 'group' is an analysis of identical markers).

☒ All plots are contour plots with outliers or pseudocolor plots.

☒ A numerical value for number of cells or percentage (with statistics) is provided.

## Methodology

| Sample preparation | Semi-confluent MCF10A or U2OS cells were treated with 10uM EdU for 2 hrs prior to harvesting. Click-iT Plus EdU Alexa Fluor 488 Flow Cytometry Assay Kit (Invitrogen C10632) with the addition of FxCycle PI/RNase Staining Solution (Invitrogen F10797) was used per manufacturer's protocol and flow cytometry of a minimum of 10000 cells per condition was performed on LSRFortessa with subsequent analysis with Flow Jo version 10.8.1 (Flow Jo, BD). |
| --- | --- |
| Instrument | BD LSRFortessa |
| Software | Flow Jo version 10.8.1 |
| Cell population abundance | 10000 cells analyzed for each condition (no events except dead cells excluded) |
| Gating strategy | See above |

☐ Tick this box to confirm that a figure exemplifying the gating strategy is provided in the Supplementary Information.

