## [Peer Review File · Nature Genetics]

Peer Review Information

Manuscript Title: APOBEC3B regulates R-loops and promotes transcription-associated mutagenesis in cancer

Corresponding author name(s): Professor Kyle Miller, Dr Natalia Gromak, Professor Reuben (S) Harris

Reviewer Comments & Decisions:

Decision Letter, initial version:

4th Sep 2020

Dear Professor Harris,

First of all, please accept my sincere apologies for the delay in returning this decision to you. Your patience has been much appreciated.

Your Article entitled "R-loop resolution and mutagenic outcomes promoted by the DNA cytosine deaminase APOBEC3B" has now been seen by 3 referees, whose comments are attached. While they find your work of potential interest, they have raised serious concerns which in our view are sufficiently important that they preclude publication of the work in Nature Genetics, at least in its present form.

While the referees find your work of some interest, they raise concerns about the strength of the novel conclusions that can be drawn at this stage. You will see that all three reviewers recognise the potential impact of the work. However, all three are unified in their call for more mechanistic depth to underpin your observations. In particular, they ask for experiments to delineate how A3B is resolving R-loops and also for more evidence that these sites are indeed targets for A3B-mediated mutagenesis. Without these data, we agree that publication in Nature Genetics would, at this stage, be premature.

Should further experimental data allow you to fully address these criticisms we would be willing to consider an appeal of our decision (unless, of course, something similar has by then been accepted at Nature Genetics or appeared elsewhere). This includes submission or publication of a portion of this work someplace else.

The required new experiments and data include, but are not limited to those detailed here. We hope you understand that until we have read the revised manuscript in its entirety we cannot promise that it will be sent back for peer review.

If you are interested in attempting to revise this manuscript for submission to Nature Genetics in the future, please contact me to discuss a potential appeal. Otherwise, we hope that you find our referees' comments helpful when preparing your manuscript for resubmission elsewhere.

Sincerely,

Safia Danovi
Editor
Nature Genetics

Referee expertise:

Referee #1: APOBEC biology

Referee #2: DNA damage and R-loop biology

Referee #3: transcription-coupled repair and APOBEC biology

Reviewers' Comments:

Reviewer #1:

Remarks to the Author:

Many have proposed for years that the single-stranded DNA present in R-loop structures are potential targets of APOBEC proteins but without any data so far to support this claim. The manuscript by McCann et al. presents the first strong evidence that indeed APOBEC3B targets R-loop structures in cells and promotes mutations at R-loop sites. In addition, McCann et al. found that A3B not only targets R-loop structures but is also important in the resolution of R-loops. This result is both surprising and novel because it implies that A3B may have an important function for the protection against genome instabilities caused by R-loops during transcription. The results shown in this manuscript are convincing but still preliminary. Whereas data to support the increase of R-loops in the absence of A3B are strong and well presented, there are no results to explain how A3B prevents or resolves R-loops. Moreover, there is a lack of strong evidence to demonstrate that APOBEC3B generates mutations at R-loop sites.

Major comments

Alas, only a speculative mechanism was proposed in the discussion to explain why A3B is important to suppress R-loop in cells. The authors should provide a mechanism to explain why there is an accumulation of R-loops in the absence of A3B and moreover, how A3B protects cells against the accumulation of R-loops. Some models proposed in the discussion can be tested, such as the recruitment of DNA repair factors at R-loop sites in an A3B activity-dependent manner.

A3B is expressed at different levels in cancer cells as shown in several previous publications from the lead author (e.g., Burns et al., Nature 2013). Does the level of A3B correlate with R-loop level in cancer cells? From the results presented in this manuscript, I would anticipate that cancer cell lines

with a high level of A3B may be less prone to accumulate R-loop? Moreover, the A3B gene is often absent amongst many human genomes. Do A3B null cancer cell lines contain a higher level of R-loops?

I further find curious why the authors use the MCF10A cell line for the DRIP-seq experiments. The authors showed previously that MCF10A does not express A3B and reported no deaminase activity in this cell line (Burns et al., Nature 2013). Similar results have been reported by other groups (Kanu et al., Genome Biology 2016). I would expect that U2OS is a better cell line model because it will have a stronger variation of A3B level between WT and A3B KO cells and thus accordingly should lead to a stronger change of R-loop levels. It is surprising that the increase of R-loop levels monitored by IF in the absence of A3B looks similar in U2OS cells (high A3B expressing cell line) and MCF10A cells (very low to no A3B).

There is no comparison of the APOBEC mutation level at R-loop sites vis-a-vis outside R-loop sites. Do R-loop sites have more APOBEC mutations than other regions, in particular, untranscribed regions?

Recently, several groups demonstrated that the APOBEC signature arose mainly as a result of A3A activity and not A3B, especially in breast tumors that are used in this study (Cortez et al., PLoS Genetics 2019). For this reason, it does not seem justified a priori to separate tumors with high versus low APOBEC signature to study A3B mutation at R-loop. In addition, whilst the authors showed that A3A does not resolve R-loops, they cannot exclude that A3A does not cause mutations at R-loop sites too. It will be essential to demonstrate that the mutations quantified at R-loop sites are the consequence of A3B activity and not A3A. A3A is known to favor the YTC site whereas A3B prefers the RTC site (Chen et al., Nature Genetics 2015). Are APOBEC mutations quantified at R-loop sites mainly present in YTC or RTC sites? A subpopulation of breast cancer cells accumulates numerous mutations at RTC sites (Jalili et al., Nature Communications 2020) suggesting that A3B mutations are prominent in some breast tumors. The analysis shown in Figure 7 should focus on this population of tumor samples. Moreover, are tumor samples with a high APOBEC signature but A3B null depleted of mutations at R-loop sites?

Other comments

Figure 3a-d: There is no control experiment to demonstrate that JQ1 induces R-loop level.

Figure 3g: There is only a difference of the R-loop level at one concentration of genomic DNA. It is surprising to not observe a dose-dependent effect. Especially perplexing is that the low concentration of gDNA of WT versus E255A showed no difference.

Figure 4c-d: The quantification does not seem to reflect the dot blot experiment. The quantification showed a very strong decrease of S9.6 level after FLV and TRP treatment whilst the dot blot membrane only showed a very weak decrease.

Figure 5: The global quantification of the R-loop level by DRIP-seq showed a strong difference between WT and A3B KO cells (about 4 to 5-fold increase) in Figure 5b. However, this quantification does not seem to correlate with the 2 examples shown in Figure 5e with a modest increase of 2-fold. How do the other propose to explain such a discrepancy? Moreover, there are no examples to illustrate the "decreased" group.

Figure 5: There is no complementation experiment to demonstrate that restoring the A3B level also restores the R-loop level by DRIP-seq (same comments for IF and dot blot results). Does A3B overexpression after DOX induction lead to the opposite effect compared with the A3B KO for the 2 described groups (enriched and decreased)? This is an important experiment to demonstrate that the change is not the result of a clonal selection difference especially with such a minute change in R-loop level shown in Figure 5e.

Figure 5/6e-f: Why do the input samples have such a high background signal that they appear almost identical to the A3B+DOX Chip-seq samples? I would expect a flat signal for the input, which unfortunately raises concerns about the significance of the enrichment detected in the A3B+DOX Chip-seq samples. I echo these same comments for the A3B-Dox Chip-seq samples. Why is the signal almost identical to +DOX samples?

Figure 6: Confirmation of the kinetics of R-loop induction and resolution after PMA treatment by IF and dot blot is important in demonstrating that all these methods are indeed monitoring the same type of structures.

Figure 7f: The bottom part of the model includes only speculation about how A3B could resolve R-loop structures, and has not been demonstrated in this manuscript. To avoid confusion and future overinterpretation amongst the literature, I believe it important to not conclude this manuscript by proposing a speculative model that has not been validated. The bottom half should be removed or clearly stated that it presents a speculative, unproven model.

Reviewer #2:

Remarks to the Author:

In the manuscript "R-loop resolution and mutagenic outcomes promoted by the DNA cytosine deaminase APOBEC3B", the authors investigate the functional interplay between the APOBEC3B (A3B) and R-loops as a potential substrate of this cytosine deaminase enzyme. In an AP-MS approach, the authors first show significant overlap of the A3B interactome with a previously published R-loop proteomic dataset (Figure 1). Loss of A3B leads to increased nuclear R-loop accumulation (Figure 2), whereas overexpression of a catalytically active enzyme reduces R-loops in a transcription-dependent manner (Figures 3-4).

This part of the paper is solid and shows a clear connection between A3B and global R-loop levels in human cancer cell lines, although the authors do not really address how direct these effects are in terms of A3B acting on the ssDNA portion of the R-loop as suggested in their model (see more specific comments below).

The authors then perform DRIP-Seq in A3B WT and knockout cells as well as A3B-ChIP-Seq to correlate changes in R-loop distribution and A3B binding genome-wide under normal and PMA-mediated transcription-induced conditions (Figure 5-6). These results are consistent with the idea that A3B can target R-loops in vivo to prevent their accumulation or help to resolve induced R-loops at around 3800 genes in a timely manner. Finally, these genes characterized by A3B-mediated R-loop resolution show an increased APOBEC3 mutation signature in TCGA and ICGC breast cancer tumor samples (Figure 7).

The second part of the paper is rather correlative and some of the effects are rather mild and overstated in the manuscript. For example, it is not clear to me how comparable the DRIP-Seq datasets in A3B WT and knockout cells are with A3B ChIP-Seq binding profiles in an overexpression

situation. It seems difficult to make correlative conclusions from these two experimentally distinct conditions. The authors should first solidify the evidence of a direct physical connection between R-loops and APOBEC3 under comparable experimental conditions. Therefore, although I believe this study has potential and addresses an interesting and cancer-relevant connection of A3B and R-loops, it seems too preliminary in its current state for publication in Nature Genetics.

Major Comments:

- 1) The quality of the AP-MS data is difficult to assess for reviewers as the only data provided is the final table with specific A3B interactors. How many proteins were identified in each of the individual six biological replicates and what was the overlap with controls? How many proteins were filtered out by comparison with the CRAPome. The authors should provide a full list and better overview of the proteomic dataset.
- 2) Related to this point, the authors validate physical interaction of A3B with TOP1 and hnRNPUL1 by Co-IP experiments (Figure 1d/e). However, TOP1 is not part of the list of 34 A3B interactors identified in the study? Was this protein identified in their AP-MS datasets? This raises the question how many other bona fide interaction partners are not included in this list and what is the real overlap of factors with the S9.6 interactome?
- 3) The S9.6 IF experiments presented in Figures 2-4 show globally a nice correlation between the the level of catalytically active A3B and R-loop levels. However, this could also be mediated via an indirect effect of A3B protein affecting the expression, protein level or activity of another important R-loop regulator. Can the authors provide more direct evidence that A3B is directly bound to R-loops (e.g. via a Proximity ligation assay between A3B and S9.6)? Can this interaction be reversed via RNaseH overexpression?
- 4) The authors conclude from the DRIP-Seq data in Figure 4 that A3B dramatically alters the genome-wide distribution of R-loops but this seems overstated in light of the fact that the vast majority of R-loop peaks remain unchanged despite apparent A3B binding as determined by A3B overexpression. This surprising result may also be explained by the two very distinct experimental conditions used for this correlative analysis.
- 5) Another concern with the A3B ChIP-Seq data is that the Input lanes show a similar profile as the IP's (e.g Figure 5e). The authors should at least provide better insights into these ChIP-seq datasets. Is the majority of peaks coinciding with R-loop peaks (as only this is shown in the metaplots) or are there also other A3B peaks outside of R-loop regions. Can the authors confirm the relative A3B enrichments at example genes by ChIP-qPCR?
- 6) One conceptual question that remains is whether the targeting of R-loops by A3B is now beneficial (due to faster R-loop resolution) or more detrimental (due to cytosine deamination and arising mutations). The model in Figure 7 is quite vague as it seems to support both faithful and mutagenic repair. This should at least be better discussed.

Minor Comments:

- 1) The cartoon in Figure 1A for validation of the AP-MS experiment is not really meaningful and should be replaced with real data from the supplemental Figure S1.
- 2) The authors state in the Materials and Methods section that "S9.6 signal was quantified by removing both cytoplasmic and nucleolar signal" and nucleoli were defined as "overlying DAPI-less regions, which correspond to nucleoli". This is not clear to me how this analysis was done as some of the example images do not show any DAPI-less regions despite very bright S9.6 nucleolus signals. Other labs have overcome this issue by co-staining with a nucleolar marker (e.g. Sollier et al., Mol Cell 2014) but I cannot find a similar experimental setup here. This seems also critical to exclude the possibility that A3B loss triggers nucleolus disruption and the "increase in nucleoplasmic signal" are simply scattered nucleolar fragments which have recently also been implicated with Pol II transcription

(see Abraham et al., Nature 2020)

Reviewer #3:

Remarks to the Author:

The manuscript "R-loop resolution and mutagenic outcomes promoted by the DNA cytosine deaminase APOBEC3B" by McCann and colleagues proposes a role of APOBEC3B in processing R-loops in human cells and tumors. The authors show that APOBEC3B interacts with multiple R-loop interacting proteins by mass spectrometry and that loss of APOBEC3B results in increased R-loop presence as measured by immunofluorescence and dot blot. The elevation of R-loops appears to be partially dependent on the enzymatic activity of APOBEC3B. To determine which specific R-loops APOBEC3B helps resolve, the authors conduct DRIP-seq experiments and determine ~ 20,000 R-loop sites that either have increased or decreased signal in the absence of APOBEC3B. These changed sites tend to be enriched in protein coding sequences in the genome. Finally, the authors assess the enrichment of APOBEC-induced mutation at R-loop sites in breast cancers and argue that the R-loops that are enhanced in the absence of APOBEC3B contain more APOBEC signature mutations, suggesting that recruitment of APOBEC3B to these sites may make them prone to mutation by the enzyme. The identification of a role for APOBEC3B in R-loop resolution would establish a novel normal function in the enzyme and therefore could be of significant importance. However, I have some concerns about the analysis linking APOBEC3B activity at R-loop sites to cancer mutagenesis and the lack of a mechanism describing how APOBEC3B mediates R-loop resolution. My major concerns follow.

1) The authors state that their DRIP-seq experiments indicate large genome-wide changes in R-loop formation in the APOBEC3B deficient cells. However, ~20,000 sites are changed and ~130,000 are unchanged. This is less than 15% of sites measured which seems to be either a limited effect or targeted to specific sites.

2) Based on the immunofluorescence quantifications presented, APOBEC3B KO cells have 2-3 fold more S9.6 signal. This would suggest that APOBEC3B is helping resolve 50-66% of the R-loops in the cell. Is there a reason for the discrepancy between the immunofluorescence data and the DRIP-seq data?

3) If only 15% of R-loops are processed by APOBEC3B, are these sites related in any way that would indicate a possible reason that this set is specifically targeted by the enzyme and that APOBEC3B is not acting more generally in R-loop resolution? Does Gene Ontology analysis of the genes impacted by R-loop formation indicate that a specific class of genes is prone to R-loop formation in the absence of APOBEC3B?

4) In Fig. 7, the authors indicate that APOBEC-induced mutations are enriched in DRIP-seq peaks that are increased with APOBEC3B deficiency. However, in the high APOBEC Signature tumors, there is very little difference in mutation abundance between the R-loop classes. This would indicate that APOBEC3B does not specifically target any R-loop class. The effect the authors report is driven largely by the standardization of APOBEC-induced mutations in high APOBEC-Signature tumors by the amount of putative APOBEC-induced mutations in low APOBEC-Signature tumors. This standardization is misleading since it will compare whether there is a difference in the distribution of mutations between the tumor classes, not if the mutations across a single class are unevenly distributed across the genome. It's particularly problematic since the confidence that any mutations in the low APOBEC signature mutations are actually caused by APOBEC3B and not another mutational process that happens to have an overlapping signature. These other mutational processes are likely to have a different distribution across the genome as APOBEC signature mutations have an uniquely uniform distribution across genomic features like replication timing and chromatin accessibility [1-3]. DRIP-seq

peaks enriched in the absence of APOBEC3B are in protein coding regions that have lower mutation densities for non-APOBEC-induced mutations [4], which may explain, why mutations in this R-loop class is depleted in the low APOBEC-signature tumors.

5) APOBEC3B cytidine deaminase activity on ssDNA is strongly inhibited by RNA [5]. It is unclear how it would induce damage on DNA in areas with high RNA density. This effect would seem to argue against the mechanism for R-loop resolution proposed in Fig. 7f. An in vitro assay showing APOBEC3B activity on a R-loop-like substrate would significantly help support the claim that recruitment of APOBEC3B to R-loops could enhance DNA mutagenesis.

6) The differences in the S9.6 immunofluorescence signal in Fig. 2A does not appear to match the quantifications in Fig. 2B. There seems to be very little difference in Fig. 2A between treatments. This is likely due to the very intense nucleolar staining, which appears to be RNaseH resistant in this panel. The lack of RNaseH sensitivity of the S9.6 signal indicates that either there are some artefactual staining with the S9.6 antibody [6] under the conditions used or the RNaseH treatment was incomplete. The supplemental material indicates that nucleolar staining is subtracted from all immunofluorescence analyses. The authors should indicate in the main text that the intense staining is the nucleoli, explain if its RNaseH resistance is an artefact, and that the signal has been removed from analysis. While it is somewhat standard to exclude nucleolar staining from S9.6 immunofluorescence analysis, I'm not sure it should be excluded. Other R-loop processing factors, like RNaseH and Top1 localize to nucleoli to process R-loops formed during rDNA transcription [7]. Additionally, the dot blot analysis would presumably have nucleolar R-loops included.

7) In Fig. 3c, it appears that the GFP-tagged APOBEC3B is excluded from nucleolar S9.6 signal. Is there a reason for this and would it suggest that APOBEC3B does not co-localize with the R-loops in the rDNA? As mentioned above, RNaseH and Top1 are reported to co-localize with the nucleoli.

8) In Fig. 3, the requirement for APOBEC3B enzymatic activity on R-loop resolution is tested. In panel d, the data indicates that over-expression of the catalytically dead APOBEC3B reduces S9.6 signal, just not to the extent of the wild type APOBEC3B. This may indicate that the catalytic activity is only partially required. However, no control is included to prove the wild type and mutant are expressed at the same level. This is critical for the interpretation of the result.

9) Very little data is presented to support the mechanism presented in Fig. 7f as to how APOBEC3B may be mediating R-loop resolution. Only that APOBEC3B catalytic activity may be partially required. Is Ung2 or mismatch repair machinery also needed?

10) Improper regulation of R-loop stability is often associated with DNA double strand breaks and difficulty with cell cycle progression [8]. Are either of these phenotypes observed in APOBEC3B-deficient cells? If so, it would strengthen the argument that APOBEC3B is playing an important role in R-loop resolution that is biologically important for a cell.

References:

1. Haradhvala, N., et al., Mutational strand asymmetries across cancer reveal mechanisms of DNA damage and repair *Cell*, 2016.
2. Kazanov, M.D., et al., APOBEC-Induced Cancer Mutations Are Uniquely Enriched in Early-Replicating, Gene-Dense, and Active Chromatin Regions. *Cell Rep*, 2015. 13(6): p. 1103-9.
3. Morganella, S., et al., The topography of mutational processes in breast cancer genomes. *Nat Commun*, 2016. 7: p. 11383.
4. Lawrence, M.S., et al., Mutational heterogeneity in cancer and the search for new cancer-associated genes. *Nature*, 2013. 499(7457): p. 214-218.
5. Cortez, L.M., et al., APOBEC3A is a prominent cytidine deaminase in breast cancer. *PLoS Genet*, 2019. 15(12): p. e1008545.
6. Vanoosthuyse, V., Strengths and Weaknesses of the Current Strategies to Map and Characterize R-Loops. *Noncoding RNA*, 2018. 4(2).

7. Shen, W., et al., Dynamic nucleoplasmic and nucleolar localization of mammalian RNase H1 in response to RNAP I transcriptional R-loops. *Nucleic Acids Res*, 2017. 45(18): p. 10672-10692.
8. Abakir, A., et al., N(6)-methyladenosine regulates the stability of RNA:DNA hybrids in human cells. *Nat Genet*, 2020. 52(1): p. 48-55.

Decision Letter, First Appeal

29th Apr 2021

Dear Professor Harris,

I'm so sorry that it's taken so long to return this decision to you. Thank you so much for bearing with me.

Your Article, "R-loop resolution and mutagenic outcomes promoted by the DNA cytosine deaminase APOBEC3B" has now been seen by 3 referees. You will see from their comments copied below that while they find your work of considerable potential interest, they have raised quite substantial concerns that must be addressed. In light of these comments, we cannot accept the manuscript for publication, but would be very interested in considering a revised version that addresses these serious concerns.

We hope you will find the referees' comments useful as you decide how to proceed. If you wish to submit a substantially revised manuscript, please bear in mind that we will be reluctant to approach the referees again in the absence of major revisions.

You'll note that your reviewers consider the manuscript to be improved. However, Reviewers #1 and #3 remain unconvinced by the assertion that APOBEC3B (A3B) suppresses R-loops and that A3B-mediated R-loop resolution causes enhanced A3B-dependent mutagenesis. Editorially, we remain intrigued by this idea and recognise that it has the potential to be field-forwarding. However, as I think I said to you during our last conversation, the burden of proof with novel concepts such as this one is high, and for us to consider moving forward with the paper, we will need both Reviewers #1 and #3 to be satisfied that your proposed mechanism is sufficiently supported by your data. So, please address their comments in full. Reviewer #2 has made minor requests which should also be addressed.

If you choose to revise your manuscript taking into account all reviewer and editor comments, please highlight all changes in the manuscript text file. At this stage we will need you to upload a copy of the manuscript in MS Word .docx or similar editable format.

*1) Include a "Response to referees" document detailing, point-by-point, how you addressed each referee comment. If no action was taken to address a point, you must provide a compelling argument.

This response will be sent back to the referees along with the revised manuscript.

*2) If you have not done so already please begin to revise your manuscript so that it conforms to our Article format instructions, available [here](http://www.nature.com/ng/authors/article_types/index.html). Refer also to any guidelines provided in this letter.

[redacted]

If you wish to submit a suitably revised manuscript we would hope to receive it within 6 months. If you cannot send it within this time, please let us know. We will be happy to consider your revision so long as nothing similar has been accepted for publication at Nature Genetics or published elsewhere. Should your manuscript be substantially delayed without notifying us in advance and your article is eventually published, the received date would be that of the revised, not the original, version.

Thank you for the opportunity to review your work.

Sincerely,

Safia Danovi
Editor

Nature Genetics

Reviewers' Comments:

Reviewer #1:

Remarks to the Author:

The authors have provided some new information and addressed some of my concerns. However, I still find several major issues with the results: in particular, the new analyses on Kataegis and mutation in splicing factors provide purely speculative models based on correlations and are not supported by any empirical evidence. While the findings presented in this manuscript are novel and interesting, the results shown are preliminary and there remains a lack of a clear mechanism explaining how A3B mediates R-loop resolution.

- The authors provided new data to show that A3B DNA binding is important to prevent R-loop. While these experiments are nice controls to further confirm the implication of A3B, these results do not directly address or explain how A3B prevents the increase of R-loop in cells. A3B-mut2 comprises in fact 6 point mutations across the NTD domain of A3B that together have been shown to affect the overall structure of this domain (PMID: 28575276). Does the Mut2 mutant affect the interaction with other R-loop-related proteins identified in the MS experiment? It would be an interesting model to explore that A3B is important for the recruitment of these factors to R-loop structures.

- The new results showing a high level of Kataegis in transcribed regions driven by A3B are very interesting. However, the authors make a sizable leap to then claim that this increase of Kataegis events in transcribed regions is the direct consequence of R-loop and A3B targeting R-loop structures, especially that there are almost as many Kataegis events in intergenic regions. There are currently no data to support this speculative model and the authors must provide some experimental evidence in support of their claims.

- I have a similar comment in reference to the result showing a correlation between A3B mutation levels and mutation in splicing factors. Whereas defects in splicing factors have been shown to increase R-loop, there is no direct evidence that the increase of A3B mutations is caused by an increase of R-loop levels in these tumors. The proposed model is interesting but fairly speculative.

- The complementation experiments are not convincing. Why do the authors only show complementation experiments for dot plots and not IF or ChIP? The phenotype monitored by IF is much stronger than the one monitored by dot plot (the quantification shows a middle change between conditions and only at one concentration). The authors must complement A3B KO cells with A3B WT or E255A and compare the level of S9.6 to the WT cells by IF.

- In Figure 8f,g, the authors analyse the enrichment of mutations in RTCA motifs that have been previously associated with A3B. The authors should focus on RTCA mutations in the other panels of this figure as well since the authors are only interested in A3B mutations. Figure 8c-e should be compared to the level of A3B mutations intergenic regions as well.

- Figure 8i-j: there is no distinction in the analysis between transcribed and untranscribed regions.

Other comments:

- Why are the new dot-blot experiments (in Figures 2, 3, and 4) performed with WT and A3B KO conditions presented in separate panels? Provided the goal of these panels is to directly compare WT and A3B KO, it is essential to show the results in the same panel for a fair comparison. If presented separately, how can the reader determine whether the experiments have been done on the same membrane with the same exposure time?
- There is no coomassie gel to assess the quality of A3B purification and the mutants.
- There is no control experiment in Figure 3b 3d, and 7f to show the level of S9.6 in untreated cells. It is important to show the basal level of S9.6 staining in cells without treatment to assess the level of S9.6 decrease when A3B is overexpressed compared to untreated cells. I am not sure I understand why the authors express A3B-WT, E255A, or A3A in WT U2OS treated with JQ1 and not directly in A3B KO cells. Since JQ1 induces S9.6 in the presence of A3B, it is unclear why the authors hypothesise that overexpressing A3B should suppress S9.6 in this situation? Does the JQ1 treatment affect A3B recruitment to R-loop?

Reviewer #2:

Remarks to the Author:

The authors have properly addressed most of the issues raised by the three referees and have greatly improved their manuscript. Importantly, this revised version includes new biochemical and bioinformatic results that strengthen the conclusion that A3B can directly interact and deaminate sequences in the context of the ssDNA of an R-loop. My concerns from the previous version have been adequately addressed. In my opinion, this manuscript in its current version represents an important advance in the field and should be published in Nature Genetics. However, four minor issues need to be addressed prior to publication:

1. Figure 3a,b: The authors show specific reduction in R-loop signal upon A3B expression in JQ1 treated cells as a mean of R-loop induction. Could the authors repeat this experiment with another R-loop inducing agent (e.g. splicing inhibitor) to exclude that this effect is specific to JQ1 induced R-loops?
2. Page 11: "A global comparison of DRIP-seq peaks between KO and WT MCF10A revealed large changes in the overall R-loop landscape with 8,296 peaks 'increased', 13,761 peaks 'decreased', and 154,036 peaks 'unchanged'. Even though I agree with the authors response that the level of changes is in agreement with previous R-loop perturbation studies, I would rather choose the word "moderate" than "large" changes here, as the large majority of peaks is simply unchanged.
3. Similar comment in Page 12: "As anticipated, PMA caused large changes in the overall R-loop landscape with 13,422 peaks increased, 16,432 peaks decreased, and 171,322 unchanged"
4. The new biochemical data shows that A3B can bind and deaminate R-loops is an important addition to the manuscript. Even though, "native EMSAs were hard to quantify due to accumulation of large protein/nucleic acid complexes in the wells", I would find these gels informative and should be added at least to the supplement instead of only showing competitor experiments.
5. Regarding the different nucleic acid/R-loop structures used in Figure 7, the authors should provide more information on the sequences that were used, how these structures were annealed and what

kind of competitor DNA was used for the individual constructs. Otherwise, it's very difficult to assess the differences in affinities between the structures, as the authors claim.

Reviewer #3:

Remarks to the Author:

The authors have adequately addressed most of my criticisms. They should be commended for the amount of additional work put into the revision. However, they did not sufficiently address my previous point 4, which stated that the reported enrichment of APOBEC-induced mutations in regions that contain increased R-loops upon APOBEC3B deletion is driven by the standardization of these enrichments by tumors containing very little APOBEC-induced mutation. While the authors agreed that these critiques were fair, they maintained the same analysis in the manuscript and added two additional analyses to strengthen their argument.

Unfortunately, these additional analyses also have issues that limit the conclusions that can be drawn. In new figures 8 f and g, they authors assess the enrichment of APOBEC3A-like signatures and APOBEC3B-like signatures in kataegis events that occur on the non-transcribed strand, transcribed strand, or intergenic regions. They then compare this to APOBEC3A and APOBEC3B enrichments among Non-clustered mutations. In the rebuttal, the authors claim that APOBEC3B-induced kataegis are enriched in transcribed regions, primarily on the non-transcribed strand. From the data shown, it is impossible to determine the number of the events on the transcribed strand, non-transcribed strand, and intergenic regions. The p-values shown claim to be for a comparison between the plots of kataegis mutations versions scattered mutations in each genomic region class, which would assess whether kataegis events are more likely to be enriched in the APOBEC3B signature as compared to the APOBEC3A signature. This does not evaluate whether there are differences in APOBEC3B signature enrichment in R-loop forming regions compared to non-R-loop forming regions.

In new figures 8h and j, the authors then assess the percentage of total mutations that are composed of the APOBEC mutation signature in 81 tumors containing somatically acquired mutations in Splicing factors compared to 81 randomly chosen tumors without a mutation in a splicing factor. No statistics were included in this figure. Furthermore, this analysis needs additional controls to be informative. The authors need to evaluate whether the tumors have similar numbers of total mutations in them. This is important because tumors with higher mutational burdens will be more likely to have a mutation in the splicing factors. Because APOBEC-induced mutations are frequently the major component of the total mutation burden, there may be an artifactual association between tumors with splicing factor mutations and the amount of APOBEC-induced mutation. I would also recommend that the authors perform the same analysis for several other housekeeping genes to ensure that they do not produce the same effect.

The analyses in Figure 8 panels b, c, d, e, h, and j also have an issue in that they are evaluating total numbers of APOBEC-induced mutations in each of these analyses, despite the fact that multiple APOBECs contribute to these totals. APOBEC3A mutated tumors often have much higher amounts of APOBEC-induced mutation compared to APOBEC3B mutated tumors which would seem likely to overwhelm the signal for the APOBEC3B-induced mutations. If the authors are observing an effect on total APOBEC mutations, this would suggest that the effect is not specific to APOBEC3B. The authors may want to consider only evaluating mutations in tumors enriched for RTC-type APOBEC mutations.

I would recommend that either this figure be removed from the manuscript or that the analyses be corrected. The role of APOBEC3B in R-loop processing may be interesting enough on its own for publication without extension to contributing to APOBEC-induced mutations. For panels b-e to show that APOBEC3B-induced mutations are enriched in the regions that R-loops are increased upon APOBEC3B depletion, the authors should identify tumors that are likely to be specifically mutated by APOBEC3B (i.e. the tumors are enriched in RTC mutations compared to YTC mutations) and then compare the density of mutations in the increased R-loop regions (where APOBEC3B should be recruited) compared to the unchanged regions (where APOBEC3B is not specifically recruited). They should not standardize this data by mutations from tumors with little APOBEC3B activity.

Minor Point:

I also recommend the authors re-assess the quantification of the deaminase activity assays in new figure 7. While the experiments clearly show APOBEC3B activity on R-loop structures in vitro, the activity of APOBEC3B on the R-loop appears to be significantly reduced compared to the activity on ssDNA in the gel images in Figures 7b and 7e. The quantifications indicate that their activity on the two substrates is similar.

Author Rebuttal to Initial comments

Detailed Responses to the Reviewers' Comments

Reviewer #1:

The authors have provided some new information and addressed some of my concerns. However, I still find several major issues with the results: in particular, the new analyses on Kataegis and mutation in splicing factors provide purely speculative models based on correlations and are not supported by any empirical evidence. While the findings presented in this manuscript are novel and interesting, the results shown are preliminary and there remains a lack of a clear mechanism explaining how A3B mediates R-loop resolution.

Response: We would like to thank you for your critique and recognizing that “the findings presented in this manuscript are novel and interesting.” We have done our best to address your specific comments below including several fortifications of the global bioinformatics analyses (**Fig. 8** and Fig. S6) and additional molecular experiments (**Fig. 8d** and Fig. S1f), providing insights into molecular mechanisms of A3B-mediated R-loop resolution. In addition, to help readers appreciate the bulk of our results, we include a working model in **Fig. 8a** in which A3B (in addition to other factors) contributes to R-loop resolution through

a deamination-dependent mechanism. This working model is supported by our data sets, as well as by well-known precedents (AID in immunoglobulin gene class switch recombination and guide RNA-directed APOBEC-Cas9 complexes in base editing). In addition to conveying a visual summary of our results, we also recognize that models are starting points for future investigation and have revised our discussion to suggest additional ways forward as well as potential alternatives.

Previously you said that “Many have proposed for years that the single-stranded DNA present in R-loop structures are potential targets of APOBEC proteins but without any data so far to support this claim. The manuscript by McCann *et al.* presents the first strong evidence that indeed APOBEC3B targets R-loop structures in cells and promotes mutations at R-loop sites.

...”. We truly appreciate this assessment and have received similar reactions from other experts in our research areas. We are hopeful that our fully revised manuscript has addressed your comments and will become the first of many to explore the interconnections between A3B, R-loop resolution, and R-loop mutagenesis.

1) The authors provided new data to show that A3B DNA binding is important to prevent R-loop. While these experiments are nice controls to further confirm the implication of A3B, these results do not directly address or explain how A3B prevents the increase of R-loop in cells. A3B- mut2 comprises in fact 6 point mutations across the NTD domain of A3B that together have been shown to affect the overall structure of this domain (PMID: 28575276). Does the Mut2 mutant affect the interaction with other R-loop-related proteins identified in the MS experiment? It would be an interesting model to explore that A3B is important for the recruitment of these factors to R-loop structures.

Response: As a requested, we have now included new co-IP data in **Fig. S1f** comparing interactor binding with wildtype A3B and Mut2. Five representative interactions were analyzed in co-IP experiments and, in all instances, binding was reduced or abolished for Mut2, suggesting that one or more of these proteins may co-operate with A3B in maintaining R-loop homeostasis. In addition, we are confident that the 6 amino acid substitutions in Mut2, as reported by Dr. Xiaojiang Chen and co-workers (2017, *Nucleic Acids Research*, PMID: 28575276), do not alter the core structure and/or integrity of the A3B N-terminal domain. These

residues are all located on the surface of the protein, and they result in higher (not lower) yields of protein from 293T cell purifications (**Fig. S5a**). We are confident that this protein is folded properly and functional because, like wildtype A3B, it retains the capacity to localize to the nuclear compartment (**Fig. 7e**), which is a hallmark property of A3B governed by the N-terminal domain, and it retains DNA deaminase activity (**Fig. 7g**).

Moreover, we have also revised the **Discussion** to better explain our working model and potential alternatives, as well as include roles for other interactors that may be part of our

existing AP-MS data sets (or may very well await future identification). We note in our revised **Discussion** that the mechanism(s) responsible for the observed R-loop decreases in the absence of A3B are less clear and may be due to altered R-loop accessibility and/or altered interactions affecting overall R-loop homeostasis. We feel that answers to all of these additional questions are beyond the scope of this paper and more appropriate for future studies.

2) The new results showing a high level of Kataegis in transcribed regions driven by A3B are very interesting. However, the authors make a sizable leap to then claim that this increase of Kataegis events in transcribed regions is the direct consequence of R-loop and A3B targeting R-loop structures, especially that there are almost as many Kataegis events in intergenic regions. There are currently no data to support this speculative model and the authors must provide some experimental evidence in support of their claims.

Response: We agree that these results are indeed very interesting. In revised **Fig. 8** and Fig. S6, we now provide three independent tests of the working model that A3B contributes to R-loop resolution and at least a proportion of these encounters lead to mutagenic outcomes (error-free outcomes are not detectable by bioinformatic analyses). First, we show that gene *over-expression* (and not simply gene expression) levels associate positively with the overall fraction of APOBEC3 signature mutations observed across breast cancer (**Fig. 8b** and Fig. S6a-c). Second, we show splice factor mutant breast tumors are more likely to harbor APOBEC3-attributed mutations than non-SF mutant breast tumors (**Fig. 8c** and Fig. S6d-e). This finding is underscored by new data showing that the splice inhibitor, Plad B, causes a general increase in R-loops that can be suppressed by overexpressing A3B (**Fig. 8d**). These results are really interesting because prior literature has associated both dysregulated gene expression and splice-factor defects with R-loop formation, and we are now also able to add additional associations with APOBEC3 mutations. Third, we fortify the previously presented kataegis analyses showing an RTCW mutation bias, particularly on the non-transcribed strand of expressed genes (**Fig. 8f-g**), with a new analysis showing a striking bimodal distribution of *kataegic* events in breast cancer and that the majority of APOBEC3 *kataegic* events are located far away from sites of structural variation (and therefore also likely far away from DNA double-strand breaks and recombination repair; **Fig. 8e**). Taken together with all of our other studies, these additional results provide additional support for an A3B-facilitated R-loop resolution and mutation model and are not easily explained by any other mechanism.

3) I have a similar comment in reference to the result showing a correlation between A3B mutation levels and mutation in splicing factors. Whereas defects in splicing factors have been shown to increase R-loop, there is no direct evidence that the increase of A3B mutations is caused by an increase of R-loop levels in these tumors. The proposed model

is interesting but fairly speculative.

Response: As above for kataegis, our analysis of the APOBEC mutation signature in the splice factor mutant vs non-mutant breast cancers is motivated by prior literature (PMID 29617667; PMID 32076118; PMID 30054334; PMID 29395063; bioRxiv <https://doi.org/10.1101/2020.06.08.130583>) and included as one of several tests of the working model in **Fig. 8a**. We also now break-down these comparisons into A3B-associated RTCW and A3A-associated YTCW mutation groups, which shows large proportions of mutations in both of these tetranucleotide motifs in most tumors (Fig. S6d-e). In addition, we found that the A3B-associated tetranucleotide preference is enriched in the splice factor mutant group and/or depleted from the non-splice factor mutant group ($P = 0.028$ by Fisher's exact test).

Second, please see response #2 to Reviewer 3, where we perform several additional analyses including random sampling of housekeeping genes and we were unable to find another mutated gene that could associate with APOBEC3-attributed mutations in breast cancer.

Last, because the separation-of-function A3B Mut2 protein is unable to promote R-loop resolution (**Fig. 7h**) and it has a reduced capacity to interact with at least five A3B WT interactors (including some splice factors; Fig. S1f), it is possible that the accessibility of R-loops to A3B may be influenced by these interactions and/or by the overall homeostasis of splice factors in cells. This connection to splicing is further strengthened by new data showing that the splice inhibitor, Plad B, causes a general increase in R-loops that can be suppressed by overexpressing A3B (**Fig. 8d**).

4) The complementation experiments are not convincing. Why do the authors only show complementation experiments for dot plots and not IF or ChIP? The phenotype monitored by IF is much stronger than the one monitored by dot plot (the quantification shows a middle change between conditions and only at one concentration). The authors must complement A3B KO cells with A3B WT or E255A and compare the level of S9.6 to the WT cells by IF.

Response: The answer to this question is complicated by technical and logistical constraints. First, as described in our previously revised manuscript, we generated an A3B-depleted MCF10A cell population and performed complementation experiments with wildtype and catalytic mutant enzymes (WT and E255A, respectively; **Fig. 3e-g**). Quantification of R-loop levels by dot blotting showed that the overall suppressive effect caused by WT A3B requires catalytic activity. These experiments were done multiple times in the Harris lab, which had generated and validated both the shRNA and overexpression constructs. The technical and logistical issues arise because these experiments need to be done with freshly transduced cell populations. This is due to our observation that A3B-overexpressing cells do not revive from cryo-storage, which we suspect is due to the fact that endogenous A3B gene expression is cell-cycle regulated (ex. PMID 32985974), and we suspect that at least one part of the cell cycle is particularly susceptible to A3B-catalyzed deamination. We admit that we do not fully

understand why happily growing cultures can tolerate wildtype A3B expression driven from a heterologous promoter, yet the same culture has major issues reviving from cryostorage (despite dedicated efforts by Harris lab members to generate constructs that as-close-as-possible recapitulate endogenous A3B expression levels). Thus, complementary IF and ChIP experiments have not been done is because we could not ship these complemented lines for revival in the Miller (UT- Austin) and Gromak (Oxford) labs, which have the required expertise for the other methodologies. Moreover, COVID-19 restrictions as well as a recent fire (<https://med.umn.edu/news-events/fire-damage-uic-ccrb-1-220-facility>) have prevented the Harris lab from accessing fluorescent microscopes in institutional core facilities, which would have been required to optimize an R-loop IF protocol and conduct complementary experiments.

5) In Figure 8f,g, the authors analyse the enrichment of mutations in RTCA motifs that have been previously associated with A3B. The authors should focus on RTCA mutations in the other panels of this figure as well since the authors are only interested in A3B mutations. Figure 8c-e should be compared to the level of A3B mutations intergenic regions as well.

Response: We have done the requested analyses for APOBEC3 *kataegic* events (**Fig. 8f-g**), for the APOBEC3 signature/splice factor mutant analyses (**Fig. 8c** and Fig. S6d-e), and for associations with gene over-expression (**Fig. 8b** and Fig. S6c). These combined data sets provide strong evidence for A3B contributing to R-loop mutagenesis. However, please note that analyses of mutations in RTCW motifs alone may underestimate the impact of A3B because **1)** A3B can inflict mutations in both RTCW and YTCW motifs (our original biochemical studies in PMID 23389445 and PMID 24154874 as well as work by others in PMID 26258849; PMID 23599896; PMID 31249028; and PMID: 32719516) and **2)** the bias reported originally in *yeast* for mutation of RTCW vs YTCA motifs by human A3B is a modest 1.2- to 1.3-fold (PMID 26258849).

6) Figure 8i-j: there is no distinction in the analysis between transcribed and untranscribed regions.

Response: This comment refers to our analyses of APOBEC3-attributed mutations in splice factor mutant versus non-mutant breast cancers, which are now **Fig. 8c** and Fig. S6d-e. As requested, we have re-analyzed these data from the perspective of non-transcribed strand (NTS) and transcribed strand (TS) and have found significant enrichments for APOBEC3-associated mutations on the NTS in both the splice factor mutant and non-mutant tumor groups, consistent with our proposed model in which the NTS has a highly likelihood of being single-stranded and exposed to APOBEC3 binding and deamination activities. This additional result is now included as text in the results section (page 18, bottom): “We also noted that APOBEC3-attributed mutations accumulate preferentially on the non-transcribed strand (NTS)

over the transcribed strand (TS) in both splice factor mutant and non-mutant tumor groups with a statistical difference that may relate to the underlying mechanism ($P = 0.0284$ and $P = 4.3 \times 10^{-12}$, respectively, by student's t-test).”

Other comments:

7) Why are the new dot-blot experiments (in Figures 2, 3, and 4) performed with WT and A3B KO conditions presented in separate panels? Provided the goal of these panels is to directly compare WT and A3B KO, it is essential to show the results in the same panel for a fair comparison. If presented separately, how can the reader determine whether the experiments have been done on the same membrane with the same exposure time?

Response: We apologize for not being clearer here. Of course, all dot blots for A3B WT and KO cells for each independent experiment were done using one continuous membrane. The images were cropped (and not otherwise adjusted) for assembly of publication quality figures. A representative raw image for the experiment in **Fig. 2c** is provided below for your consideration as **Response Fig. R1** and, per journal policy, we will provide all source data files/images before publication.

8) There is no coomassie gel to assess the quality of A3B purification and the mutants.

Response: These data are now shown in **Fig. S5a** for A3B WT versus Mut2. A3B is a *very* difficult protein to purify and our many attempts to remove co-purifying proteins from 293T cells invariably result in the protein crashing out of solution. It is therefore important to note that the co-purifying proteins can be regarded as solubility chaperones and, most importantly, that they are *similar* for A3B WT and Mut2 and the only ssDNA deaminase activity in our preparations comes from A3B WT. Moreover, as far as we are aware, no other lab has

published a purification protocol and Coomassie gel image for WT human A3B purified from human cells (bacterial and baculovirus preparations are less active for unknown reasons).

9) There is no control experiment in Figure 3b 3d, and 7f to show the level of S9.6 in untreated cells. It is important to show the basal level of S9.6 staining in cells without treatment to assess the level of S9.6 decrease when A3B is overexpressed compared to untreated cells. I am not sure I understand why the authors express A3B-WT, E255A, or A3A in WT U2OS treated with JQ1 and not directly in A3B KO cells. Since JQ1 induces S9.6 in the presence of A3B, it is unclear why the authors hypothesise that overexpressing A3B should suppress S9.6 in this situation? Does the JQ1 treatment affect A3B recruitment to R-loop?

Response: Recent work by the Miller lab demonstrates that JQ1 causes a

global increase in R-loop levels by inhibiting the BET family of bromodomain proteins (Kim *et al.*, 2019, *Genes and Development*; PMID 31753913). We also cite two additional studies showing that JQ1 induces R-loops in human cancer cells (Edwards *et al.*, 2020, *Cell Reports*, PMID 32966794, Lam *et al.*, 2020, *Nature Comm.*, PMID 32796829). This control is shown above as **Response Fig. R2**.

We use JQ1 in several experiments here simply as a strategy to increase R-loop levels for visualization and quantification (particularly in U2OS cells where nucleoplasmic R-loop levels are relatively low). Please also note that the vast majority of experiments were done without JQ1 including the A3B knockout and knockdown studies in multiple cell lines that invariably show a net increase in R-loop levels.

Moreover, we have now also addressed this concern using an orthologous approach by treating cells with the splicing inhibitor pladienolide B (Plad B), showing increased R-loop levels (as reported by several groups previously), and uniquely showing that these elevated R-loop levels can be suppressed by A3B (new **Fig. 8d**). These new results further demonstrate

that the
R-loop resolution activity of A3B is not unique to JQ1 treatment.

Reviewer #2:

The authors have properly addressed most of the issues raised by the three referees and have greatly improved their manuscript. Importantly, this revised version includes new biochemical and bioinformatic results that strengthen the conclusion that A3B can directly interact and deaminate sequences in the context of the ssDNA of an R-loop. My concerns from the previous version have been adequately addressed. In my opinion, this manuscript in its current version represents an important advance in the field and should be published in Nature Genetics.

Response: Thank you for this clear synopsis and positive recommendation.

However, four minor issues need to be addressed prior to publication:

1. Figure 3a,b: The authors show specific reduction in R-loop signal upon A3B expression in JQ1 treated cells as a mean of R-loop induction. Could the authors repeat this experiment with another R-loop inducing agent (e.g. splicing inhibitor) to exclude that this effect is specific to JQ1 induced R-loops?

Response: This very helpful question motivated us to perform experiments with the spliceosome inhibitor pladienolide B (Plad B). This small molecule targets spliceosome-associated 130 (SAP130), inhibits splicing factor 3B subunit (SF3B1), and impairs U2 small nuclear ribonucleoprotein (U2 snRNP) interaction with pre-mRNA. Prior studies have shown that Plad B inhibits splicing and causes a general increase in R-loops (PMID 30341290; PMID 28257700).

We also found that Plad B treatment causes an increase in R-loop levels in U2OS cells, similar to JQ1 treatment, and, importantly, we uniquely show that this increase can be suppressed by A3B expression (new **Fig. 8d**).

2. Page 11: "A global comparison of DRIP-seq peaks between KO and WT MCF10A revealed large changes in the overall R-loop landscape with 8,296 peaks 'increased', 13,761 peaks 'decreased', and 154,036 peaks 'unchanged'. Even though I agree with the authors response that the level of changes is in agreement with previous R-loop perturbation studies, I would rather choose the word "moderate" than "large" changes here, as the large majority of peaks is simply unchanged.

Response: We have deleted the word “large” and stated peak numbers without an additional descriptor.

3. Similar comment in Page 12: “As anticipated, PMA caused large changes in the overall R-loop landscape with 13,422 peaks increased, 16,432 peaks decreased, and 171,322 unchanged”

Response: As above, we have deleted the word “large”.

4. The new biochemical data shows that A3B can bind and deaminate R-loops is an important addition to the manuscript. Even though, “native EMSAs were hard to quantify due to accumulation of large protein/nucleic acid complexes in the wells”, I would find these gels informative and should be added at least to the supplement instead of only showing competitor experiments.

Response: Thank you for recognizing the importance of our biochemical experiments. As requested, we have now included an example of a native EMSA experiment in **Fig. S5c**. These experiments are indeed informative as you anticipated but admittedly not as visually pleasing as we would like (we apologize for this, but these EMSAs are the best we can do until we and/or others work-out better protocols to purify WT A3B and prevent it from aggregating).

5. Regarding the different nucleic acid/R-loop structures used in Figure 7, the authors should provide more information on the sequences that were used, how these structures were annealed and what kind of competitor DNA was used for the individual constructs. Otherwise, it’s very difficult to assess the differences in affinities between the structures, as the authors claim.

Response: Schematics of all of the different nucleic acid substrates/complexes are shown in **Fig. 7a** and described fully in the **Methods** and **Table S2**.

Reviewer #3:

The authors have adequately addressed most of my criticisms. They should be commended for the amount of additional work put into the revision. However, they did not sufficiently address my previous point 4, which stated that the reported enrichment of APOBEC-induced mutations in regions that contain increased R-loops upon APOBEC3B deletion is driven by the standardization of these enrichments by tumors containing very little APOBEC-induced mutation. While the authors agreed that these critiques were fair, they maintained the same analysis in the manuscript and added two additional analyses to strengthen their argument.

Response: Thank you for appreciating our initial responses and the amount of additional work we have done despite seemingly endless challenges imposed by COVID-19. We have now revised **Fig. 8** to remove the analyses in question and include additional bioinformatic analyses as described below.

1) Unfortunately, these additional analyses also have issues that limit the conclusions that can be drawn. In new figures 8 f and g, they authors assess the enrichment of APOBEC3A-like signatures and APOBEC3B-like signatures in kataegis events that occur on the non-transcribed strand, transcribed strand, or intergenic regions. They then compare this to APOBEC3A and APOBEC3B enrichments among Non-clustered mutations. In the rebuttal, the authors claim that APOBEC3B-induced kataegis are enriched in transcribed regions, primarily on the non-transcribed strand. From the data shown, it is impossible to determine the number of the events on the transcribed strand, non-transcribed strand, and intergenic regions. The p-values shown claim to be for a comparison between the plots of kataegis mutations versions scattered mutations in each genomic region class, which would assess whether kataegis events are more likely to be enriched in the APOBEC3B signature as compared to the APOBEC3A signature. This does not evaluate whether there are differences in APOBEC3B signature enrichment in R-loop forming regions compared to non-R-loop forming regions.

Response: We have revised **Fig. 8**, Fig. S6, the corresponding results section, and the Discussion to report the key bioinformatics analyses that we have done to test the R-loop resolution/mutation model in **Fig 8a** suggested by all of our other studies (**Fig. 1-7** and Fig. S1- S5). Our analyses of APOBEC3-attributed *kataegis* events are now shown in **Fig. 8e-h**. The most important new addition here is an analysis of the distribution of all *kataegis* events, as well as APOBEC3 *kataegis* events, relative to sites of structural variation in breast cancer (**Fig. 8e**). The bimodal distribution of *kataegis* in breast cancer is novel and very much unexpected based on prior work. Moreover, many of these distal *kataegis* events overlap significantly with the R-loop regions defined by our DRIP-seq studies, which supports our working model and provides a mechanistic connection. The additional breakdown of APOBEC3-attributed *kataegis* events into those occurring on the NTS, TS, or intergenic regions is the same as before, except we have now color-coded the dot plots to reflect R-loop associations (red if at least one APOBEC3-associated *kataegis* event overlaps with an A3B-affected DRIP-seq region) (**Fig. 8f-g**). In addition, we have provided two different statistical analyses, the first as before to compare the tetranucleotide mutation enrichments of APOBEC3-associated *kataegis* versus dispersed mutation profiles (*P*- values reported in the revised text) and the second to compare tetranucleotide mutation biases within each mutation analysis group (TS, NTS, and intergenic; *Q*-values reported on dot plots).

In addition, we show that both gene *over*-expression and splicing perturbation (Plad B treatment or somatic mutation) associate positively with mutations attributed to APOBEC3 in

breast cancer (**Fig. 8b-d**, Fig. S6). These results are really interesting because prior literature has associated both dysregulated gene expression and splice-factor defects with R-loop formation, and we are now also able to add additional associations with APOBEC3-attributed mutations (as well as with the sub-signatures associated with A3B). Further consistent with our proposed mechanism, we found that the A3B-associated tetranucleotide preference is enriched in the splice factor mutant group; $P = 0.028$ by Fisher's exact test; Fig. S6e).

2) In new figures 8h and j, the authors then assess the percentage of total mutations that are composed of the APOBEC mutation signature in 81 tumors containing somatically acquired mutations in Splicing factors compared to 81 randomly chosen tumors without a mutation in a splicing factor. No statistics were included in this figure. Furthermore, this analysis needs additional controls to be informative. The authors need to evaluate whether the tumors have similar numbers of total mutations in them. This is important because tumors with higher mutational burdens will be more likely to have a mutation in the splicing factors. Because APOBEC-induced mutations are frequently the major component of the total mutation burden, there may be an artifactual association between tumors with splicing factor mutations and the amount of APOBEC-induced mutation. I would also recommend that the authors perform the same analysis for several other housekeeping genes to ensure that they do not produce the same effect.

Response: The requested statistical analysis has been added to support **Fig. 8c**. The corresponding text has been revised as follows (p18, top): "Remarkably, 53% of the breast tumors with mutant splice factor genes (43/81) had significant levels of APOBEC3 signature mutations (Fig. 8c). In contrast, only 35% of breast tumors without mutations in the same splice factor gene set (326/841) showed a detectable APOBEC3 mutation signature (Fig. 8c; $P < 0.017$ by Fisher's exact test)."

We also appreciate your comment about mutation numbers and, despite higher numbers mutations (mostly APOBEC3-associated) in splice-factor mutant breast tumors, we were not able to find associations for any other gene set. For instance, we randomly selected 119 housekeeping genes from a pool of 3,804 housekeeping genes defined previously (Eisenberg & Levanon, 2013, *Trends in Genetics*, PMID 23810203), and we ran 100,000 simulations to find that none of the randomly selected housekeeping gene sets were even capable of accumulating 81 mutations within the entire TCGA breast cancer data set ($n=976$). If there was an artifactual association with mutation load, then we would have expected it to emerge in such an analysis. Thus, as far as we can determine, the significant association between detrimental mutations in splice factor genes and APOBEC3-attributed mutations in tumors shown here in **Fig. 8c** and Fig. S6d-e is unique and cannot be recapitulated with any other gene set.

3) The analyses in Figure 8 panels b, c, d, e, h, and j also have an issue in that they are

evaluating total numbers of APOBEC-induced mutations in each of these analyses, despite the fact that multiple APOBECs contribute to these totals. APOBEC3A mutated tumors often have much higher amounts of APOBEC-induced mutation compared to APOBEC3B mutated tumors which would seem likely to overwhelm the signal for the APOBEC3B-induced mutations. If the authors are observing an effect on total APOBEC mutations, this would suggest that the effect is not specific to APOBEC3B. The authors may want to consider only evaluating mutations in tumors enriched for RTC-type APOBEC mutations.

Response: The majority of our mutation analyses now include the full APOBEC3 mutational pattern (C mutations in TCW motifs) as well as breakdowns comparing C mutations in RTCW vs YTCW (revised **Fig. 8** and new **Fig. S6**). These combined data sets provide strong evidence for A3B contributing to R-loop mutagenesis. However, please note that mutations in RTCW motifs alone may underestimate the impact of A3B because 1) A3B can inflict mutations in both RTCW and YTCW motifs (our original studies in PMID 23389445 and PMID 24154874 as well as work by others in PMID 26258849; PMID 23599896; PMID 31249028; and PMID: 32719516) and 2) the bias reported originally in *yeast* for mutation of RTCW vs YTCA motifs by human A3B is a modest 1.2- to 1.3-fold (PMID 26258849).

4) I would recommend that either this figure be removed from the manuscript or that the analyses be corrected. The role of APOBEC3B in R-loop processing may be interesting enough on its own for publication without extension to contributing to APOBEC-induced mutations. For panels b-e to show that APOBEC3B-induced mutations are enriched in the regions that R-loops are increased upon APOBEC3B depletion, the authors should identify tumors that are likely to be specifically mutated by APOBEC3B (i.e. the tumors are enriched in RTC mutations compared to YTC mutations) and then compare the density of mutations in the increased R-loop regions (where APOBEC3B should be recruited) compared to the unchanged regions (where APOBEC3B is not specifically recruited). They should not standardize this data by mutations from tumors with little APOBEC3B activity.

Response: As suggested, we have removed the standardized analyses.

Minor Point:

5) I also recommend the authors re-assess the quantification of the deaminase activity assays in new figure 7. While the experiments clearly show APOBEC3B activity on R-loop structures in vitro, the activity of APOBEC3B on the R-loop appears to be significantly reduced compared to the activity on ssDNA in the gel images in Figures 7b and 7e. The quantifications indicate that the activity on the two substrates is similar.

Response: The results of the triplicate experiments in **Fig 7d** (formerly Fig. 7b) and **Fig. 7g** (formerly Fig. 7e) show that the deaminase activity of WT A3B on R-loop exposed ssDNA is

~2-fold lower than that observed with the same enzyme preparation on free ssDNA. These results are described in the text on page 15. In addition, to help avoid any potential for confusion, we have split the previous panel b into three separate panels, b-d.

Decision Letter, first revision:

23rd Sep 2021

Dear Reuben,

I'm sorry it's taken so long to return this decision to you.

Your Article entitled "R-loop homeostasis and cancer mutagenesis promoted by the DNA cytosine deaminase APOBEC3B" has now been seen by 3 referees, whose comments are attached. We have discussed the reviews extensively as an editorial team and we have decided that we cannot offer to publish your manuscript in Nature Genetics.

First of all, I appreciate that you had concerns that Reviewer #1 is conflicted. They have - as you will see - provided a critical report of the revision but I should stress that we have no reason to consider that they are conflicted. We consider their technical concerns, particularly with respect to the use of the S9.6 antibody, to be important. As you know from our earlier discussions, we remain enthusiastic about your high-level findings and we believe the links between APOBEC activity with respect to R-loop resolution and mutagenesis to be novel and interesting. However, for findings such as these, the burden of proof needs to be high and unfortunately, our view is that the revisions have not met this threshold.

I know that this will be very disappointing, particularly given the efforts that you and your team have put into the revisions, and the fact that the initial review report submitted was positive. However, I hope that you can understand our editorial position, and our need to ensure high technical standards across the board.

I see that you have opted out of transfer consultations but I'd be very happy to discuss the paper with my colleagues at Nature Communications to see if there'd be a path to publication there. Please let me know if this would be helpful and I could get the ball rolling straightaway. And of course, if you'd like to discuss our decision in more detail, we can do so.

I am sorry that we cannot be more positive on this occasion but hope that you will find our referees' comments helpful when preparing your paper for submission elsewhere.

Sincerely,

Safia Danovi
Editor
Nature Genetics

Reviewers' Comments:

Reviewer #1:

Remarks to the Author:

The authors have addressed some of my concerns. However, there remain a number of major issues that have been ignored and must still be addressed.

First, the manuscript proposes no mechanism based on data to explain why R-loop levels increase in the absence of A3B. Whilst the current results showing R-loop levels increase when A3B is removed is an interesting observation, a mechanism that explains this increase is surely necessary for publication in such a high-impact journal.

Second, I am concerned that the authors have not performed any complementation experiment of the KO A3B cells nor is a control included for the JQ1 experiments (in the same panel) when monitoring R-loop by IF as I previously requested. These experiments are essential considering that all the dot-blot results show little to no difference between WT and KO cells (only one concentration showed a small significant difference whilst the others showed none). Ergo, the authors need to focus their work on the IF experiment and include all the appropriate controls that remain standard in our field. I understand that technical difficulties are inherent to experimental work, but none of these experiments are challenging (similar experiments seem to have been completed in Figure 3). The authors used a DOX inducible method to express A3B in Figure 1 and showed no A3B expression when DOX is not present. This system should limit cell death during freezing as the cells can be frozen absent of any exogenous A3B expression.

Long term experiments with KOs or shRNA can induce profound changes in cells and can ultimately lead to the observed increase of R-loop levels yet not be the direct consequence of the lack of A3B. For this reason, cell complementation with A3B WT or mutants is essential to demonstrate that restoration of A3B level restores R-loop levels. Moreover, complementation experiments are crucial in establishing whether the DNA deaminase activity of A3B is required (or not) to prevent an increase of R-loop levels. Currently, the argument that the DNA deaminase activity of A3B is important seems based on particularly weak differences between cells treated with JQ1 only or complemented with WT or E255A (Figure 3d-g). From the presented data, I am not convinced that A3B-E255A leads to a different outcome than A3B WT. Regardless of the JQ1 experiments, appropriate controls with DMSO only are essential to determine if the complementation with A3B WT suppresses R-loop at a level comparable to the basal level in untreated cells.

Third, recent papers have revealed concerns regarding the use of S9.6 antibody to monitor R-loop levels, i.e., that it "creates pervasive artifacts when imaging RNA:DNA hybrids" by IF (PMID: 33830170). These new reports have stressed the importance of performing appropriate control experiments and the implementation of alternative approaches not based on the S9.6 antibody when studying R-loop (PMID: 34232287). Key results should be confirmed using such alternative methods (e.g., GFP-dRNH1) to comply to the new standards for high-quality science in our field.

Fourth, the authors now show a Coomassie gel image of A3B purified (Figure S5a). However, the

quality of the purification is markedly concerning for the interpretation of the results since A3B represents a minuscule fraction (barely visible) of the total proteins present in the purified samples. I think it considerably misleading to state that an experiment has been performed with purified proteins when it clearly is not, and there is no indication of the identity of these contaminant proteins. It seems reasonable to interpret that this information was intentionally avoided in the previous version of this manuscript (and even now, the gel is still partially cropped). I find this practice disturbing. Moreover, I disagree with the authors' statement that "no other lab has published a purification protocol and Coomassie gel image for WT human A3B purified from human cells". A detailed purification protocol has been published recently using human A3B in HEK293T cells and a Coomassie gel image was shown with minimal co-purifying proteins (PMID: 31841499 Figure 4B). A RNA and DNA binding assay must be performed with the high-quality purified protein or the results should be removed from the manuscript as preliminary.

Finally, the authors continue to make a sizable leap to their claim that R-loops are associated with an increase of APOBEC mutations. As I stated previously, it appears the authors have overinterpreted their results by using correlation events as proof of the proposed causal mechanism. There seem to be no any significant differences between Kataegis-associated mutations in the transcribed region (TS and NTS) and the intergenic regions in Figure 8F-g, and I do not see any significant difference between Kataegis with or without an R-loop-associated event. Moreover, the association among the increases of mutations in splicing factors, gene expression, and APOBEC mutations is only correlative rather than causative. No information is presented regarding how the authors selected mutated splicing factors. Can splicing defects be detected in this tumor too? More importantly, the authors did not look at A3B expression in breast cancers with mutated splicing factors or with high gene expression. Do these tumors also overexpress A3B too? If yes, such an observation will explain why there are more APOBEC mutations and not necessarily the result of an increase of R-loop levels. In a nutshell, the authors have provided no strong data to support their model that APOBEC mutations at Kataegis events are the result of increased R-loop levels.

Reviewer #2:

Remarks to the Author:

The authors have sufficiently addressed my minor concerns from the second round of revisions. In my opinion, this manuscript in its current version represents an important advance in the field and should be published in Nature Genetics.

Reviewer #3:

Remarks to the Author:

The authors have mostly addressed my previous set of critiques. However, I still have some recommendations concerning the description of the data now in figure 8.

- 1) Please add the simulations of APOBEC signature mutations in tumors with somatic mutations in 119 housekeeping genes as a supplementary figure to support the statement in the manuscript.
- 2) The text states that "APOBEC3-attributed mutations accumulate preferentially on the NTS over the TS in splice factor mutant and non-mutant tumor groups." However, the data is not shown and previously published analyses from other groups indicate only a small transcriptional asymmetry for APOBEC-induced mutations (Haradhvala et al., Cell, 2016; Morganello, et al., Nature Communications

2016). Presenting the data and a potential explanation for any differences with other analyses would be beneficial.

3) The shown association of APOBEC3 mutations with gene expression is stated in the text as "the magnitude of gene overexpression in breast cancer compared to normal breast." It is not obvious how the change in expression level from normal cell to tumor cell fits the model of APOBEC3B-inducing mutations at R-loops. Are the authors suggesting that the more abnormal the transcription at a locus is, the more R-loops are formed and therefore more APOBEC3B-induced mutations occur?

4) The color coding of tumors in Figs. 8 f-g as to whether they contain a kataegis event that occurs in a R-loop prone regions still does not directly address whether R-loops are being targeted by APOBEC3B for mutagenesis. No comparisons are being made based upon this distinction, which results in these panels primarily supporting a role for APOBEC3B in causing kataegis in general. R-loops themselves may or may not play a role in this. Also, the authors indicate that the skew towards RTCW is greatest for kataegis events on the NTS over the TS, however, this appears to be based on the p-value instead of the effect size of the skew. P-value is also dependent on the number of observations, and therefore is not an appropriate measure for this statement.

5) Overall, the analysis in figure 8 supports increased APOBEC3B-induced mutation in tumors with dysregulated transcription but falls short of linking this to a role of APOBEC3B in R-loop processing. This caveat should be made clear in the discussion.

Decision Letter, Second Appeal

IMPORTANT: Please note the reference number: NG-A55322R2-Z Harris. This number must be quoted whenever you communicate with us regarding this paper.

4th Jan 2023

Dear Dr. Harris,

First of all, happy new year. I hope you managed to have a restful break. Second, please accept my apologies for the delay in returning this to you. I was travelling or on annual leave for most of December and so everything was delayed. I am so sorry for keeping you waiting.

Thank you for asking us to reconsider our decision on your manuscript "R-loop homeostasis and cancer mutagenesis promoted by the DNA cytosine deaminase APOBEC3B". I have now discussed the points of your appeal with my colleagues, and we're happy to send the revised manuscript back to Reviewers #1, #3. Unfortunately this means that the next step for you is to resubmit everything (cover letter, revised manuscript, and point-by-point response) - I'm sorry about this.

Thank you for flagging a competing preprint. While this didn't feature in our decision, I am mindful that you will be anxious to expedite the process and I'll do my best to get reports back to you in a timely manner.

When preparing your resubmission, please ensure that it fully complies with our editorial requirements for format and style; details can be found in the Guide to Authors on our website (<http://www.nature.com/ng/>).

Please be sure that your manuscript is accompanied by a separate letter detailing the changes you have made and your response to the points raised. At this stage we will need you to upload:

- 1) a copy of the manuscript in MS Word .docx format.
- 2) The Editorial Policy Checklist:
<https://www.nature.com/documents/nr-editorial-policy-checklist.pdf>
- 3) The Reporting Summary:
<https://www.nature.com/documents/nr-reporting-summary.pdf>
(Here you can read about the role of the Reporting Summary in reproducible science:
<https://www.nature.com/news/announcement-towards-greater-reproducibility-for-life-sciences-research-in-nature-1.22062>)

Please use the link below to be taken directly to the site and view and revise your manuscript:

[redacted]

With kind wishes,

Safia Danovi
Editor
Nature Genetics

Author Rebuttal, first revision:

Reviewers' Comments:

Reviewer #1:

Remarks to the Author:

The authors have addressed some of my concerns. However, there remain a number of major issues that have been ignored and must still be addressed.

First, the manuscript proposes no mechanism based on data to explain why R-loop levels increase in the absence of A3B. Whilst the current results showing R-loop levels increase when A3B is removed is an interesting observation, a mechanism that explains this increase is surely necessary for publication in such a high-impact journal.

We have presented mechanistic analysis by doing extensive biochemical experiments with A3B mutants in vitro and in cellulo, which demonstrate that both nucleic acid-binding and nuclear localization activities of A3B are required for its function in R-loop resolution (Fig. 7). In addition, we have also shown that A3B R-loop-resolution function is dependent on its catalytic deamination activity by slot blot and IF experiments (including new imaging studies

with a mCherry-RNaseH1-mutant construct; **Fig. 3**). Furthermore, using co-IP approaches we show that the nucleic acid binding mutant of A3B has reduced interaction with multiple A3B-interacting factors, including splicing factors, suggesting that one or more of these proteins co-operates with A3B in maintaining R-loop homeostasis (**Fig. S1f**). Overall, our biochemical, genetic, and bioinformatics data sets support a model in which A3B binds to nascent ssRNA and/or to ssDNA exposed in R-loop structures, favoring their deamination and thus promoting their resolution. It is further possible that A3B-interacting factors (i.e., splicing factors and RNA/DNA helicases such as DHX9) co-operate with A3B in this function of R-loop resolution as elaborated in our revised discussion.

Second, I am concerned that the authors have not performed any complementation experiment of the KO A3B cells nor is a control included for the JQ1 experiments (in the same panel) when monitoring R-loop by IF as I previously requested. These experiments are essential considering that all the dot-blot results show little to no difference between WT and KO cells (only one concentration showed a small significant difference whilst the others showed none). Ergo, the authors need to focus their work on the IF experiment and include all the appropriate controls that remain standard in our field. I understand that technical difficulties are inherent to experimental work, but none of these experiments are challenging (similar experiments seem to have been completed in Figure 3). The authors used a DOX inducible method to express A3B in Figure 1 and showed no A3B expression when DOX is not present. This system should limit cell death during freezing as the cells can be frozen absent of any exogenous A3B expression.

We have now included requested controls with DMSO and JQ1 inhibitor as new Fig. 3a-d. As shown in these experiments, R-loops are increased by JQ1 in U2OS cells compared to untreated or DMSO-treated control cells. Moreover, panels a-b use S9.6 mAb and panels c-d use an mCherry-RNaseH1-mutant construct for R-loop detection to fortify this key point. These results are in line with extensive published data with JQ1 and R-loops (Kim et al., 2019, Genes Dev, PMID 31753913; Edwards et al., 2020, Cell Rep, PMID 32966794; Lam et al., 2020, Nature Comm, PMID 32796829).

The complementation experiments presented in our paper have now been done using two independent experimental set ups. First, we used shRNA to deplete A3B in U2OS cells, which resulted in higher R-loop levels, and then we used this system to show that WT A3B but not the E255A catalytic mutant was able to restore R-loop levels (**Fig. 3i-k**). Second, in new IF experiments using both S9.6 mAb and mCherry-RNaseH1-mutant approaches to quantify R-loop formation, we show that the increased R-loop levels caused by CRISPR knock-out of A3B in U2OS cells are restored to original levels by complementation with WT A3B but not with the E255A catalytic mutant (new **Fig. 3l-o**, **Fig. S2l-m**). Thus, using both knockdown and

knockout approaches and both slot blot and IF techniques (S9.6 and mCherry-RNaseH1-mutant), we see that WT but not catalytic A3B mutant restores the A3B activity on R-loops demonstrating the functional importance of the catalytic glutamate and therefore supporting a deamination-dependent mechanism of R-loop resolution by A3B.

We appreciate that slot blot experiments show smaller increases in R-loop levels upon A3B KD (~1.5 fold). However, we would like to point out that this change is comparable with data reported in the literature when the knockdown of other R-loop regulators was analyzed using this assay (e.g., TOP1 Manzo et al., 2018, Genome Biology, PMID 30060749; Promonet et al., 2020, Nature Commun, PMID 32769985; DDX5 Mersaoui et al., 2019, EMBO J, PMID 31267554). We also note that slot blot with S9.6 antibody performed with RNaseH control is considered a reliable technique to detect major changes in the R-loop levels (Vanoosthuysse, 2018, Noncoding RNA, PMID 29657305; Sanz et al., 2021, J Vis Exp, PMID 34515688; Miller et al., 2022, Nucleic Acids Res, PMID 35758606).

Long term experiments with KOs or shRNA can induce profound changes in cells and can ultimately lead to the observed increase of R-loop levels yet not be the direct consequence of the lack of A3B. For this reason, cell complementation with A3B WT or mutants is essential to demonstrate that restoration of A3B level restores R-loop levels.

Moreover, complementation experiments are crucial in establishing whether the DNA deaminase activity of A3B is required (or not) to prevent an increase of R-loop levels. Currently, the argument that the DNA deaminase activity of A3B is important seems based on particularly weak differences between cells treated with JQ1 only or complemented with WT or E255A (Figure 3d-g). From the presented data, I am not convinced that A3B-E255A leads to a different outcome than A3B WT. Regardless of the JQ1 experiments, appropriate controls with DMSO only are essential to determine if the complementation with A3B WT suppresses R-loop at a level comparable to the basal level in untreated cells.

As described in the response above, two independent complementation experiments show that WT but not A3B catalytic mutant restores activity on R-loops. These experiments have been performed in the absence of JQ1 treatment and indeed they show that WT A3B suppresses R-loops in A3B KD/KO cells at a level comparable to the basal level in WT untreated cells.

We appreciate that long-term depletion of A3B could indirectly impact R-loop levels. That is why, as requested during a prior round of revision, we have confirmed the results obtained in A3B KO cells by short-term siRNA depletion (72 hrs) of A3B levels in HeLa cells by DRIP-qPCR (Fig. S3e-f). This experiment argues against the possibility that R-loop changes

observed may be the result of long-term effects in A3B KO cells. This conclusion is further supported by similar results with a new U20S A3B KO line, which was complemented and analyzed by both S9.6 staining and mCherry-RNaseH1-mutant imaging (*Fig. 3l-o, Fig. S2l-m*).

We would also like to point out that A3B is a non-essential human gene that, due to a common deletion allele, is fully lacking in many individuals (Kidd et al., 2007, PLoS Genetics, PMID 17447845). Consistent with its non-essential nature and regulatory roles such as R-loop resolution under stress conditions, no prior reports with A3B to our knowledge have documented that “Long term experiments with KOs or shRNA can induce profound changes in cells”. Indeed, many labs, including ours, have reported A3B shRNA and CRISPR KO studies and none to our knowledge have reported issues (e.g., Burns et al., 2013, Nature, PMID 23389445; Law et al., 2015, Science Advances, PMID 27730215; Petljak et al., 2022, Nature, PMID 35859169).

Third, recent papers have revealed concerns regarding the use of S9.6 antibody to monitor R-loop levels, i.e., that it "creates pervasive artifacts when imaging RNA:DNA hybrids" by IF (PMID: 33830170). These new reports have stressed the importance of performing appropriate control experiments and the implementation of alternative approaches not based on the S9.6 antibody when studying R-loop (PMID: 34232287). Key results should be confirmed using such alternative methods (e.g., GFP-dRNH1) to comply to the new standards for high-quality science in our field.

As requested, we have confirmed our IF data with a new complementation experiment with an mCherry-RNaseH1-mutant construct, which does not rely on the S9.6 mAb, as described above (new Fig. 3c-d, Fig. 3l-o, Fig. S2l-m), as well as with independent approaches presented in the paper, including slot blot, DRIP-seq, and DRIP-qPCR, which also all include corresponding RNaseH controls. Moreover, as highlighted by this reviewer and Smolka et al., 2021, J Cell Biol, PMID 33830170, we are aware of the limitations of the S9.6 Ab and particularly of the fact that it recognizes dsRNA in the cytoplasm and nucleolus. For this reason, we exclusively quantify nucleoplasmic signal (excluding cytoplasm and nucleolus) and demonstrate that this nucleoplasmic signal is RNaseH sensitive in all experiments, indicating that S9.6 detects bona fide RNA/DNA hybrids and is not an artefact.

Fourth, the authors now show a Coomassie gel image of A3B purified (Figure S5a). However, the quality of the purification is markedly concerning for the interpretation of the results since A3B represents a minuscule fraction (barely visible) of the total proteins present in the purified samples. I think it considerably misleading to state that an

experiment has been performed with purified proteins when it clearly is not, and there is no indication of the identity of these contaminant proteins. It seems reasonable to interpret that this information was intentionally avoided in the previous version of this manuscript (and even now, the gel is still partially cropped). I find this practice disturbing. Moreover, I disagree with the authors' statement that "no other lab has published a purification protocol and Coomassie gel image for WT human A3B purified from human cells". A detailed purification protocol has been published recently using human A3B in HEK293T cells and a Coomassie gel image was shown with minimal co-purifying proteins (PMID: 31841499 Figure 4B). A RNA and DNA binding assay must be performed with the high-quality purified protein or the results should be removed from the manuscript as preliminary.

We describe the presence of unavoidable co-purifying contaminants in the A3B preparations in the figure legend (Fig. S5a) and in the Methods section. Despite multiple independent attempts, we have been unable to make A3B preparations as clean as those reported in Cortez et al., 2019, PLoS Genetics, PMID 31841499 (we also apologize for not citing this paper, which was not intentional). However, we note that the Coomassie image reported in the paper by Cortez et al. also shows co-purifying proteins in the same kDa ranges as the dominant co-purifying bands in our preparations. We have tried different purification strategies for full-length A3B and they all result in some level of co-purifying proteins. Our attempts at further purification, such as SEC, have thus far resulted in A3B crashing out of solution and/or indefinitely being retained in columns. Nevertheless, most importantly, the co-purifying proteins for wildtype A3B and Mut2 are the same and therefore unlikely to be responsible for the stark biochemical differences observed for nucleic acid binding. Accordingly, we include the following text in the revised legend to Fig. S5a: "We note that co-purifying proteins () are indistinguishable between A3B WT and Mut2 and therefore any differences in biochemical and cellular studies are most likely due to the amino acid substitution mutations in the latter enzyme [prior biochemical studies with full-length A3B have also reported co-purifying proteins (Cortez et al., 2019, PLoS Genetics, PMID 31841499).]" Moreover, we include the full gel image below as Fig. X, panel b as further assurance that images are not being manipulated beyond the necessary cropping required for journal presentation.*

Finally, the authors continue to make a sizable leap to their claim that R-loops are associated with an increase of APOBEC mutations. As I stated previously, it appears the authors have overinterpreted their results by using correlation events as proof of the proposed causal mechanism. There seem to be no any significant differences between Kataegis-associated mutations in the transcribed region (TS and NTS) and the intergenic regions in Figure 8F-g, and I do not see any significant difference between Kataegis with or without an R-loop-associated event. Moreover, the association among the increases of mutations in splicing factors, gene expression, and APOBEC mutations is only correlative rather than causative.

Our report only claims an association between APOBEC3 mutagenesis and R-loop-associated events. We have been very careful not to overstate this claim in our revised manuscript, and we further added in the discussion that "...although several independent data sets provide strong support for an R-loop mutagenesis model, further experiments including direct cause and effect studies with individual loci will be necessary to demonstrate a role for A3B in R-loop mutagenesis." However, taken all together, our different bioinformatic results combine to support the proposed mechanism of A3B-catalyzed deamination of R-loops, which can result in both non-mutagenic and mutagenic outcomes (model in Fig. 8a).

As regards APOBEC3 signature kataegis mutations, our analyses inform that there are statistically significant skews of RTCA in kataegis events within NTS regions compared to non-clustered mutations. To emphasize the skew of the R-loop associated kataegis events, we have now calculated the Cohen's D effect sizes across all pair-wise region comparisons (Fig. 8f-g and Fig. S7). Specifically, we see an effect size of 0.37 and 0.27 between NTS:TS and NTS:Intergenic regions across R-loop associated kataegis events (≥ 5 mutations per event), respectively, which supports the claim that the RTCA bias is strongest with respect to NTS kataegis events.

*To further fortify the association between APOBEC3 mutagenesis and R-loop-associated events, we have extended our analysis of APOBEC3 signature mutations in overexpressed genes in breast tumors (Fig. 8b/Fig. S6c) by showing a strong bias toward the NTS over the TS in the highest gene overexpression group (>16 FC; Wilcoxon rank sum test $p < 0.038$). This new result is described in our revised **Results** section, and all NTS and TS mutations in this overexpression gene group are listed in a new table (Supplementary Table S2).*

No information is presented regarding how the authors selected mutated splicing factors. Can splicing defects be detected in this tumor too? More importantly, the authors did not look at A3B expression in breast cancers with mutated splicing factors or with high gene expression. Do these tumors also overexpress A3B too? If yes, such an observation will explain why there are more APOBEC mutations and not necessarily the result of an increase of R-loop levels. In a nutshell, the authors have provided no strong data to support their model that APOBEC mutations at Kataegis events are the result of increased R-loop levels.

*We used a previously published set of 119 splicing factor genes with recurrent mutations in 33 cancer types (Seiler et al., 2018, Cell Reports, PMID 29617667). This information is provided in the **Methods** section. We have not analyzed or claimed to detect splicing defects in the datasets. Our focus is not on the splicing defects, rather on recurrent mutations observed in splicing factor genes and the mutation landscapes in the tumors with those recurrent splicing factor mutations. Because splice factor mutations have been well documented previously to result in increased R-loop accumulation (e.g., Chen et al., Mol Cell, 2018, PMID 29395063), it is a reasonable extension to also ask here if this results in increased APOBEC3 mutagenesis.*

We note that A3B mRNA expression levels in the splice factor mutant and non-mutant breast cancers are not significantly different, and therefore the significant associations we report are unlikely to be due to differential expression of this enzyme (A3B/TBP mean \pm SD = 1.47 \pm 0.30 and 1.38 \pm 0.34, respectively).

In addition, we show that increased R-loop formation caused by splicing inhibition with

pladienolide B (Plad B) can be suppressed by A3B (*Fig. 8d*). *This important result further strengthens the connection between splice defects, R-loops, and A3B-mediated resolution which could lead to mutagenic outcomes.*

Reviewer #2:

Remarks to the Author:

The authors have sufficiently addressed my minor concerns from the second round of revisions. In my opinion, this manuscript in its current version represents an important advance in the field and should be published in Nature Genetics.

Thank you very much!

Reviewer #3:

Remarks to the Author:

The authors have mostly addressed my previous set of critiques. However, I still have some recommendations concerning the description of the data now in figure 8.

1) Please add the simulations of APOBEC signature mutations in tumors with somatic mutations in 119 housekeeping genes as a supplementary figure to support the statement in the manuscript.

Although there is no tumor gene set similar to the splice factor mutant gene set analyzed here, we have carried out additional controls to address this point. We have randomly selected and analyzed 119 housekeeping genes (100,000 times), which represents the same “physical size” and number of the splice factor gene set of 119 genes. The median number of mutated housekeeping genes was

35 and the maximum was 79 (as opposed to 107/119 genes in the splice factor set). The median number of tumors containing these mutations was 15 and the maximum was 38 (as opposed to 81 in splice set). Benjamini-Hochberg corrected q-values for APOBEC3 signature enrichment in these 100,000 comparisons of housekeeping gene sets was 1.0. Please also recall that we included a randomly selected set of 81

splice factor non-mutated tumors in a previous revision (essentially the same analysis but a bit smaller; please see **Fig. Y panel b** in adjacent box). This analysis is visually helpful but, as pointed-out by a prior reviewer, not appropriate for statistical tests and thus not included in our main or supplementary display items here.

2) The text states that “APOBEC3-attributed mutations accumulate preferentially on the NTS over the TS in splice factor mutant and non-mutant tumor groups.” However, the data is not shown and previously published analyses from other groups indicate only a small transcriptional asymmetry for APOBEC-induced mutations (Haradhvala et al., Cell, 2016; Morganella, et al., Nature Communications 2016). Presenting the data and a potential explanation for any differences with other analyses would be beneficial.

*We have redone this analysis and we agree with prior reports. We have therefore removed the corresponding text including the line quoted by you here. Moreover, we now mention this point in **Discussion** and cite these papers as well as a recent study showing higher APOBEC3 mutation densities on the NTS of actively expressed genes in multiple cancer types (Chervova et al., 2021, NAR Cancer, PMID 34316712). In addition, we have extended our analysis of APOBEC3 signature mutations in overexpressed genes in breast tumors (**Fig. 8b/ Fig. S6c**) by showing a strong bias toward the NTS over the TS in the highest gene overexpression group (>16 FC; Wilcoxon rank sum test $p < 0.038$). This new result is described in our revised **Results** section, and all NTS and TS mutations in this overexpression gene group are listed in a new table (**Supplementary Table S2**).*

3) The shown association of APOBEC3 mutations with gene expression is stated in the text as “the magnitude of gene overexpression in breast cancer compared to normal breast.” It is not obvious how the change in expression level from normal cell to tumor cell fits the model of APOBEC3B-inducing mutations at R-loops. Are the authors suggesting that the more abnormal the transcription at a locus is, the more R-loops are formed and therefore more APOBEC3B-induced mutations occur?

*Yes, this interpretation is correct. We have revised the text and methods to more clearly explain this analysis. This association is shown in **Fig. 8b/ Fig. S6c left panel** and broken down into RTCW and YTCW motifs in **Fig. S6c middle & right panels**, respectively. This association between gene overexpression and APOBEC3 signature mutation in tumors is*

striking and it contrasts the lack of correlation between genome-wide gene expression levels and APOBEC3 mutations (**Fig. S6a-b**). *In the latter analyses, initially promising associations are wiped-out when one accounts for the physical size of the genes in the expression groups.* Current literature, summarized in a recent review (Bowry et al., 2021, Trends in Cancer, PMID 34052137), suggests that altered transcription (hyper-transcription) in cancer cells promotes R-loop formation. We show that R-loops are substrates for A3B deamination, therefore it is plausible that hyper-transcription in cancer is associated with increased R-loops which are in turn deaminated by A3B and processed into mutations (we have further included this reference and this concept of “altered transcription” in the revised manuscript).

*In addition, based on your interest in this observation, we extended our analysis of APOBEC3 signature mutations in overexpressed genes in breast tumors (**Fig. 8b/Fig. S6c**) by showing a strong bias toward the NTS over the TS in the highest gene overexpression group (>16 FC; Wilcoxon rank sum test $p < 0.038$). This new result is described in our revised **Results** section, and all NTS and TS mutations in this overexpression gene group are listed in a new table (**Supplementary Table S2**).*

4) The color coding of tumors in Figs. 8 f-g as to whether they contain a kataegis event that occurs in a R-loop prone regions still does not directly address whether R-loops are being targeted by APOBEC3B for mutagenesis. No comparisons are being made based upon this distinction, which results in these panels primarily supporting a role for APOBEC3B in causing kataegis in general. R-loops themselves may or may not play a role in this. Also, the authors indicate that the skew towards RTCW is greatest for kataegis events on the NTS over the TS, however, this appears to be based on the p-value instead of the effect size of the skew. P-value is also dependent on the number of observations, and therefore is not an appropriate measure for this statement.

*You are correct that we are unable to make cause-and-effect conclusions from these observations. Together with the results discussed above, our analyses suggest that many kataegis may not be recombination-associated and, rather, are consistent with the prospect of R-loop mutagenesis. With respect to the p-value, you are also correct, and the current set of analyses inform that there are statistically significant skews of RTCA in kataegis events within NTS regions compared to non-clustered mutations. To emphasize the skew of the R-loop associated kataegis events, we have now calculated the Cohen's D effect sizes across all pair-wise region comparisons (**Fig. 8f-g and Fig. S7**). Specifically, we see an effect size of 0.37 and 0.27 between NTS:TS and NTS:Intergenic regions across R-loop associated kataegis events (≥ 5 mutations per event), respectively, which supports the claim that the RTCA bias is strongest with respect to NTS kataegis events. The effect size between TS:Intergenic R-loop associated kataegis events is negligible (0.098). Further, all effect sizes between small R-loop associated kataegis (≥ 3 mutations per cluster) are negligible with an effect size of 0.082, 0.19, and 0.16 comparing NTS:TS, NTS:Intergenic, and TS:Intergenic, respectively.*

5) Overall, the analysis in figure 8 supports increased APOBEC3B-induced mutation in tumors with dysregulated transcription but falls short of linking this to a role of APOBEC3B in R-loop processing. This caveat should be made clear in the discussion.

We have now modified the discussion as recommended and included the following sentence on page 24: "... Nevertheless, although several independent data sets provide strong support for an R-loop mutagenesis model, further experiments including direct cause and effect studies with individual loci will be necessary to demonstrate a role for A3B in R-loop mutagenesis. ...".

Decision Letter, second revision:

1st Feb 2023

Dear Professor Harris,

How are you? I hope you are well.

Your Article, "R-loop homeostasis and cancer mutagenesis promoted by the DNA cytosine deaminase APOBEC3B" has now been seen by 2 referees. You will see from their comments below that while they find your work of interest, some important points are raised. We are interested in the possibility of publishing your study in Nature Genetics, but would like to consider your response to these concerns in the form of a revised manuscript before we make a final decision on publication.

To guide the scope of the revisions, the editors discuss the referee reports in detail within the team, including with the chief editor, with a view to identifying key priorities that should be addressed in revision and sometimes overruling referee requests that are deemed beyond the scope of the current study. Here, the main issues to address are (a) the issue of APOBEC3A versus APOBEC3B and (b) whether APOBEC3 activity is actually targeting R-loops for mutagenesis

We therefore invite you to revise your manuscript taking into account all reviewer and editor comments, experimentally where possible (ideally) and/or textually where appropriate. Please highlight all changes in the manuscript text file. At this stage we will need you to upload a copy of the manuscript in MS Word .docx or similar editable format.

*1) Include a "Response to referees" document detailing, point-by-point, how you addressed each referee comment. If no action was taken to address a point, you must provide a compelling argument.

This response will be sent back to the referees along with the revised manuscript.

*2) If you have not done so already please begin to revise your manuscript so that it conforms to our Article format instructions, available [here](http://www.nature.com/ng/authors/article_types/index.html). Refer also to any guidelines provided in this letter.

[redacted]

We can be flexible about timelines, but please let me know if you anticipate needing more than 6 months for your revision.

Sincerely,

Safia Danovi
Editor
Nature Genetics

Reviewers' Comments:

Reviewer #1:

Remarks to the Author:

The authors have now performed the requested experiments to demonstrate that A3B expression but not A3B catalytic mutant restores R-loop levels in A3B knockout cells. Moreover, the authors confirmed the increase of R-loop levels using a second approach (mCherry-RNase H1-mutants). These particular new results are convincing and strongly improve the quality of the initial part of the manuscript.

However, I remain still unconvinced regarding the EMSA experiments performed with purified A3B and A3B mut2. The attempt to purify A3B led to less than 5% of A3B compared to all co-purified proteins (there are many more co-purified proteins than those indicated by the 3 asterisks). I understand the difficulty in purifying A3B but high-quality purification of A3B has been successfully obtained by other labs (PMID: 31841499 and PMID: 28981865 Fig S1). I disagree with the rationale proposed by the authors "the copurifying proteins for wildtype A3B and Mut2 are the same and therefore unlikely to be responsible for the stark biochemical differences observed for nucleic acid binding." Whereas that could certainly be true, there is as yet no evidence demonstrating that these co-purified proteins have no impact on A3B DNA or RNA binding and the results. The authors cannot simply claim it as fact because they obtained the results they expected a priori. The high level of undetermined proteins present with purified A3B precludes proper interpretation of these results. As I previously conveyed, RNA and DNA binding panels must be performed with high-quality purified proteins or the results should be excised from the manuscript.

The association between APOBEC3 signature and gene expression level is very strong (Fig. 7b and S6c). However, the increase seems to be stronger on YTCW motif than on RTCW motif (Fig. S6c middle versus right panel). How do the authors explain this association with mutations that are associated predominantly with A3A yet not A3B? I maintain a similar comment with respect to Fig. S6d-e.

I find "supporting model for R-loop mutation by APOBEC" would be more appropriate for the title of this figure and the supporting text. There are no data supporting that A3B is involved more than other APOBEC on the basis of these panels.

The association between R-loop and RTCW-enriched Kataegis events is particularly weak (Fig.7f-g). The interpretation of these data should be minimised.

There is an overuse of the modifier "likely" across the manuscript, particularly evident when the authors described their computational analysis. Continual use strengthened my overall impression of the manuscript that the authors overinterpret their computational data to fit their model rather than building their model on strong computational results.

Figure 3n. What are the red and green signals that co-localise outside the nucleus of cells expressing eGFP and E255A?

Reviewer #3:

Remarks to the Author:

The authors have addressed my prior comments regarding their manuscript. My only remaining concerns are that their classification of significant and negligible Cohen's D effect sizes for the RTCA and YTCA signature skew in Figure 8 appears to be arbitrary. Why is a 0.27 effect size significant, but a 0.19 effect size is negligible? In general, I still find the comparison of RTCA to YTCA in the NTS, TS, and intergenic regions to be a non-intuitive assessment of APOBEC3B activity being targeted to R-loops. This comparison is really addressing whether APOBEC3B is more likely to mutate the NTS than APOBEC3A, which seems to be tangential to the question if APOBEC3B is mutating R-loops. Is a more direct assessment possible?

Author Rebuttal, second revision:

Responses to Reviewer's Comments

Reviewer 1

The authors have now performed the requested experiments to demonstrate that A3B expression but not A3B catalytic mutant restores R-loop levels in A3B knockout cells. Moreover, the authors confirmed the increase of R-loop levels using a second approach (mCherry-RNase H1-mutants). These particular new results are convincing and strongly improve the quality of the initial part of the manuscript.

Response: Thank you for appreciating our additional studies.

However, I remain still unconvinced regarding the EMSA experiments performed with purified A3B and A3B mut2. The attempt to purify A3B led to less than 5% of A3B compared to all co-purified proteins (there are many more co-purified proteins than those indicated by the 3 asterisks). I understand the difficulty in purifying A3B but high-quality purification of A3B has been successfully obtained by other labs (PMID: 31841499 and PMID: 28981865 Fig S1). I disagree with the rationale proposed by the authors "the copurifying proteins for wildtype A3B and Mut2 are the same and therefore unlikely to be responsible for the stark biochemical differences observed for nucleic acid binding." Whereas that could certainly be true, there is as yet no evidence demonstrating that these co-purified proteins have no impact on A3B DNA or RNA binding and the results. The authors cannot simply claim it as fact because they obtained the results they expected a priori. The high level of undetermined proteins present with purified A3B precludes proper interpretation of these results. As I previously conveyed, RNA and DNA binding panels must be performed with high-quality purified proteins or the results should be excised from the manuscript.

Response: After many attempts, we have finally established a protocol that enables our group to produce wildtype and mutant human APOBEC3B at >85% purity (please see new

Coomassie gel image in revised **Fig. S5d**). We are happy to say that our new datasets, including new ssDNA deamination and ssDNA/RNA binding experiments (**Fig. S5e-g**), fully corroborate our original findings. In short, human APOBEC3B both binds to and deaminates R-loop substrates.

The association between APOBEC3 signature and gene expression level is very strong (Fig. 7b and S6c). However, the increase seems to be stronger on YTCW motif than on RTCW motif (Fig. S6c middle versus right panel). How do the authors explain this association with mutations that are associated predominantly with A3A yet not A3B? I maintain a similar comment with respect to Fig. S6d-e.

Response: Thank you for appreciating this strong association that can only be explained through the deamination activity of the APOBEC3 enzymes on hyper-transcribed regions. Here, we are focusing on APOBEC3B because it impacts R-loop levels in multiple systems (and also because APOBEC3A does not; *e.g.*, **Fig. 3e-f**). However, we are not discounting the possibility that APOBEC3A may also contribute in a non-specific manner to mutation of R-loop regions in human cancers. This interpretation is reflected in our revised **Discussion** (page 25): “In addition,

we cannot exclude the possibility that other APOBEC3 enzymes, most notably A3A, may also contribute to R-loop mutation. However, a specific role for A3A in R-loop homeostasis is disfavored because its overexpression does not affect R-loop levels (**Fig. 3b**) and, importantly, most APOBEC3 *kataegic* events observed far away from sites of structural variation are enriched for mutations in A3B-associated 5'-RTCW motifs and not in A3A-associated 5'-YTCW motifs (**Fig. 8e-h**).”

We note that YTCW and RTCW are only a proxy for APOBEC3A and APOBEC3B activity, respectively. However, selective involvement of APOBEC3B is supported by our observation that several splice factor-mutated tumors show predominantly RTCW mutations (**Fig. S6e**) and the additional observation that larger kataegic tracks are also RTCW-biased (**Fig. 8g**). We admit that these informatic analyses are not cause-and-effect demonstrations but, taken together with our other bioinformatic studies and wet lab results, they provide strong links between APOBEC3B and R-loop mutagenesis.

We further note that a competing study recently appeared in *bioRxiv*, which provides further evidence that our major conclusions are correct (<https://www.biorxiv.org/content/10.1101/2022.10.21.513235v1>). This study takes a different approach by overexpressing APOBEC3B in a breast cancer cell line and establishing a cause-and-effect relationship between its deaminase activity and mutagenesis of R-loop regions (specifically by mapping APOBEC3B induced mutations to R-loop regions).

I find "supporting model for R-loop mutation by APOBEC" would be more appropriate for the title of this figure and the supporting text. There are no data supporting that A3B is involved

more than other APOBEC on the basis of these panels.

Response: Given the abundance of evidence demonstrating APOBEC3B mutagenesis of R-loop regions of the genome, we feel that this working model is appropriate. Other substrates for APOBEC3 mutagenesis including single-stranded DNA replication and recombination repair intermediates are not depicted for clarity.

The association between R-loop and RTCW-enriched Kataegis events is particularly weak (Fig.7f- g). The interpretation of these data should be minimised.

Response: We have done our best to describe all of our results as clearly as possible and interpret them both individually and collectively.

There is an overuse of the modifier "likely" across the manuscript, particularly evident when the authors described their computational analysis. Continual use strengthened my overall impression of the manuscript that the authors overinterpret their computational data to fit their model rather than building their model on strong computational results.

Response: We have removed some instances of "likely". However, in many instances, we feel that the use of "likely" is an appropriate qualifier, because we would prefer to offer conservative interpretations of individual results and then strengthen our interpretations in the Discussion section when the whole study can be considered.

Figure 3n. What are the red and green signals that co-localise outside the nucleus of cells expressing eGFP and E255A?

Response: We believe this cytoplasmic signal is derived from overexpression of GFP and RNaseH1 proteins, which can cause aggregates or accumulation in vacuoles in the cytoplasm in a subset of cells. We have excluded these cytoplasmic signals from all our IF analysis, focusing exclusively on nuclear signal. We do note however that the presence of RNA:DNA hybrids in the cytosol has also been reported (PMID 36544021), which has the potential to influence the mCherry-RNaseH1 signal in the cytoplasm. We have now replaced these images in the figure with more representative images of eGFP and E255A from our experiments to avoid any confusion for the readers.

Reviewer #3:

The authors have addressed my prior comments regarding their manuscript.

Response: Thank you for appreciating our revisions.

My only remaining concerns are that their classification of significant and negligible Cohen's D effect sizes for the RTCA and YTCA signature skew in Figure 8 appears to be arbitrary. Why is

a

0.27 effect size significant, but a 0.19 effect size is negligible? In general, I still find the comparison of RTCA to YTCA in the NTS, TS, and intergenic regions to be a non-intuitive assessment of APOBEC3B activity being targeted to R-loops. This comparison is really addressing whether APOBEC3B is more likely to mutate the NTS than APOBEC3A, which seems to be tangential to the question if APOBEC3B is mutating R-loops. Is a more direct assessment possible?

Response: We would like to thank the reviewer for providing additional critique of our analysis regarding APOBEC3B-associated R-loop *kataegis* found in **Fig. 8** and **Fig. S7**. To provide further clarity of these results, we have included several changes to the main text (see below). Firstly, we attempt to clarify that the investigation into A3-kataegis is motivated to ultimately provide bioinformatic support of the proposed mechanism found in **Fig. 8a**. Specifically, we expect to find an enrichment of A3B-associated mutation clusters within the NTS regions of genes (**Fig. 8f- g**), and even more specifically, within NTS of putative R-loop regions as classified by the DRIP- Seq experiments (**Fig. S7a-b**). To explore this computationally, we have partitioned the clustered events into NTS and TS events within genic regions and also within intergenic regions. Within each of these regions, the calculated pentanucleotide motif enrichment is calculated and compared to the distributions found in non-clustered events, which tend to be more dispersed across the genomic landscape. We observe only a slight enrichment of A3B-associated *kataegis* in NTS and TS regions for events with ≥ 3 mutations (**Fig. 8f**; q-values calculated using Mann-Whitney U-test; Cohen's $D=0.19$ for NTS: Intergenic, 0.16 for TS: Intergenic, and 0.082 for NTS: TS). These results are consistent when only considering the distributions of putative R-loop associated *kataegis* (**Fig. S7a**). Traditionally, *kataegis* are associated with larger events (≥ 5 mutations per cluster), so we repeated these same analyses revealing a significant skew of RTCA enrichment only within NTS regions (**Fig. 8g**; $q=0.011$), which were consistent across the putative R-loop regions (**Fig. S7b**; Cohen's $D=0.27$ for NTS: Intergenic, 0.098 for TS: Intergenic, and 0.37 for NTS: TS). Further, 70% of these *kataegic* events were more strongly enriched for RTCA compared to YTCA motifs. Collectively, these results support our proposed model reflecting an A3B-mediated R-loop resolution, which we expect to occur on the NTS within genic regions.

Decision Letter, third revision:

Our ref: NG-A55322R4

18th May 2023

Dear Dr. Harris,

Thank you for submitting your revised manuscript "R-loop homeostasis and cancer mutagenesis promoted by the DNA cytosine deaminase APOBEC3B" (NG-A55322R4). It has now been seen by the original referees and their comments are below. The reviewers find that the paper has improved in revision, and therefore we'll be happy in principle to publish it in Nature Genetics, pending minor revisions to comply with our editorial and formatting guidelines. We would also ask that you incorporate the edits that you suggested in your email to me dated 19 May to (textually) address the ongoing concerns raised by Reviewer #1 (I have included your attachment to this email).

Thank you again for your interest in Nature Genetics. We've been on quite a journal with this paper but I'm very pleased with this outcome. I look forward to seeing the paper online and in print.

Sincerely,

Safia

Safia Danovi
Editor
Nature Genetics

Reviewer #1 (Remarks to the Author):

The authors have adequately addressed my concerns regarding the EMSA experiments performed with purified A3B. They have conducted new purifications, resulting in >85% purity of A3B and A3B mut2. These new results support the absence of binding of A3B mut2 to DNA and RNA substrates. Whilst there may be some questions regarding the quality of the biochemical assays presented in this manuscript, the results are sufficient to conclude that A3Bmut2 is unable to bind R-loops.

Unfortunately, my concern regarding Figure 8f has not been satisfactorily addressed. Although the authors have made claims about an association between R-loop and RTCW-enriched Kataegis events based only on statistical results in Figure 8f, the data presented in Figure 8f do not show any visible association between R-loop and RTCW-enriched Kataegis events in the graphs shown in these panels. It remains crucial to remind the authors that statistical differences do not always imply meaningful results. Many scientists have raised concerns about misuse of statical analysis to demonstrate a point that is not relevant scientifically (<https://www.nature.com/articles/d41586-019-00857-9>). Reviewer 3 also raised similar concerns. Therefore, the actual association between R-loop and RTCW-enriched Kataegis events still needs to be demonstrated appropriately to establish a conclusive link between the two or removed from the manuscript entirely.

Reviewer #3 (Remarks to the Author):

I am satisfied with the authors clarification in regards to my final comment.

[inserted email attachment below:]

Reviewer #1

Reviewer comments:

Unfortunately, my concern regarding Figure 8f has not been satisfactorily addressed. Although the authors have made claims about an association between R-loop and RTCW-enriched Kataegis events based only on statistical results in Figure 8f, the data presented in Figure 8f do not show any visible association between R-loop and RTCW-enriched Kataegis events in the graphs shown in these panels. It remains crucial to remind the authors that statistical differences do not always imply meaningful results. Many scientists have raised concerns about misuse of statical analysis to demonstrate a point that is not relevant scientifically (<https://www.nature.com/articles/d41586-019-00857-9>). Reviewer 3 also raised similar concerns. Therefore, the actual association between R-loop and RTCW-enriched Kataegis events still needs to be demonstrated appropriately to establish a conclusive link between the two or removed from the manuscript entirely.

Response to reviewer:

We would like to thank the reviewer for continuing to stive for an optimal presentation of the reported bioinformatic analysis, which were used to support a model reflecting A3B-mediated R-loop resolution and mutagenesis. To address the visualization concerns, we have now chosen an alternative method for displaying the motif enrichment analysis found within R-loop regions (**Fig. 8f-g**). Specifically, we have replaced the RTCA versus YTCA scatter plots with standard boxplots showing the fold enrichments of RTCA/YTCA motifs specifically found within R-loop regions only. These were originally represented as the red dots in the scatter plots and were used to derive the corrected p-values using a Mann-Whitney U-test. We believe this representation more accurately captures the central message of this analysis that may have been ambiguous in prior scatter plots. Notably, larger kataegic events (≥ 5 mutations per cluster) are enriched within non-transcribed regions compared to the distributions observed across all other non-clustered mutations within non-transcribed R-loop regions. We do not see a statistically significant enrichment within transcribed or intergenic regions. Please find revised panels below.

Edits to the main text:

To provide support of a model reflecting A3B-mediated R-loop resolution via mutagenesis on the NTS, we investigated the sequence motifs of mutations across individual kataegic events compared to non-clustered mutations within R-loop regions partitioned into NTS and TS regions of genes and within intergenic regions. Specifically, we investigated the overall enrichments for A3B-associated RTCA and A3A-associated YTCA tetranucleotide motifs for each mutation found in a sample ($R=A$ or G ; $Y=C$ or T ; Fig. 8f-g)²⁶. This analysis indicated that APOBEC3 kataegic mutations overlapping NTS R-loop regions are skewed toward A3B-associated RTCA motifs, in contrast to dispersed APOBEC3 mutations (Fig. 8f-g; Q-values determined using Mann-Whitney U-tests; also see Fig. S7). The overall RTCW skew of kataegic (>3 mutations per cluster) versus dispersed APOBEC3 mutations is elevated for mutations occurring on the NTS and TS but not for mutations in intergenic regions [Fig. 8f; $Q = 0.046$ (NTS), $Q = 0.046$ (TS), and $Q = 0.62$ (intergenic)]. For greater stringency, this latter analysis was repeated for longer APOBEC3 kataegic tracts (>=5 mutations per cluster) and a statistically significant enrichment is only evident for RTCA events on the NTS of genes [Fig. 8g; $Q = 0.011$ (NTS), $Q = 0.27$ (TS), and $Q = 0.082$ (intergenic)]. Specifically, this significant enrichment is due to longer, R-loop associated kataegic events occurring within the NTS, which is not observed for TS or intergenic events (Fig. S7). Further, 70% of the R-loop kataegis occurring within the NTS were enriched for A3B-associated RTCA motifs compared to a minority of events associated with A3A-like YTCA motifs (Fig. 8g; Fig. S7b). ...

Taken together, these different bioinformatic analyses support a model in which at least a subset of R-loop structures is susceptible to C-to-U deamination events that occur on the NTS and are most likely catalyzed by A3B.

Final Decision Letter:

17th Aug 2023

Dear Dr Harris,

I am delighted to say that your manuscript "APOBEC3B regulates R-loops and promotes transcription-associated mutagenesis in cancer" has been accepted for publication in an upcoming issue of Nature Genetics.

Your paper will be published online after we receive your corrections and will appear in print in the next available issue. You can find out your date of online publication by contacting the Nature Press Office (press@nature.com) after sending your e-proof corrections. Now is the time to inform your Public Relations or Press Office about your paper, as they might be interested in promoting its publication. This will allow them time to prepare an accurate and satisfactory press release. Include your manuscript tracking number (NG-A55322R5) and the name of the journal, which they will need when they contact our Press Office.

Acceptance is conditional on the data in the manuscript not being published elsewhere, or announced in the print or electronic media, until the embargo/publication date. These restrictions are not intended to deter you from presenting your data at academic meetings and conferences, but any

enquiries from the media about papers not yet scheduled for publication should be referred to us.

Please note that *Nature Genetics* is a Transformative Journal (TJ). Authors may publish their research with us through the traditional subscription access route or make their paper immediately open access through payment of an article-processing charge (APC). Authors will not be required to make a final decision about access to their article until it has been accepted. [Find out more about Transformative Journals](https://www.springernature.com/gp/open-research/transformative-journals)

Authors may need to take specific actions to achieve [compliance with funder and institutional open access mandates](https://www.springernature.com/gp/open-research/funding/policy-compliance-faqs). If your research is supported by a funder that requires immediate open access (e.g. according to [Plan S principles](https://www.springernature.com/gp/open-research/plan-s-compliance)) then you should select the gold OA route, and we will direct you to the compliant route where possible. For authors selecting the subscription publication route, the journal's standard licensing terms will need to be accepted, including [self-archiving-and-license-to-publish](https://www.nature.com/nature-portfolio/editorial-policies/self-archiving-and-license-to-publish). Those licensing terms will supersede any other terms that the author or any third party may assert apply to any version of the manuscript.

If you have not already done so, we invite you to upload the step-by-step protocols used in this manuscript to the Protocols Exchange, part of our on-line web resource, natureprotocols.com. If you complete the upload by the time you receive your manuscript proofs, we can insert links in your article that lead directly to the protocol details. Your protocol will be made freely available upon publication of

your paper. By participating in natureprotocols.com, you are enabling researchers to more readily reproduce or adapt the methodology you use. Natureprotocols.com is fully searchable, providing your protocols and paper with increased utility and visibility. Please submit your protocol to <https://protocolexchange.researchsquare.com/>. After entering your nature.com username and password you will need to enter your manuscript number (NG-A55322R5). Further information can be found at <https://www.nature.com/nature-portfolio/editorial-policies/reporting-standards#protocols>

We've been on the the journey together with this paper! I'm pleased we've reached this stage and I look forward to seeing the paper online and in print.

Kind regards,

Safia

Safia Danovi
Editor
Nature Genetics